# The fitness cost of spurious phosphorylation

David Bradley [ID] [1,2,3,4,5,8], Alexander Hogrebe [ID] [6,8], Rohan Dandage [ID] [1,2,3,4,5], Alexandre K Dubé[1,2,3,4,5], Mario Leutert [ID] [6,7], Ugo Dionne[1,2,3,4,5], Alexis Chang[6], Judit Villén [ID] [6✉] & Christian R Landry [ID] [1,2,3,4,5✉]

## Abstract

The fidelity of signal transduction requires the binding of regulatory molecules to their cognate targets. However, the crowded cell interior risks off-target interactions between proteins that are functionally unrelated. How such off-target interactions impact fitness is not generally known. Here, we use *Saccharomyces cerevisiae* to inducibly express tyrosine kinases. Because yeast lacks bona fide tyrosine kinases, the resulting tyrosine phosphorylation is biologically spurious. We engineered 44 yeast strains each expressing a tyrosine kinase, and quantitatively analysed their phosphoproteomes. This analysis resulted in ~30,000 phosphosites mapping to ~3500 proteins. The number of spurious pY sites generated correlates strongly with decreased growth, and we predict over 1000 pY events to be deleterious. However, we also find that many of the spurious pY sites have a negligible effect on fitness, possibly because of their low stoichiometry. This result is consistent with our evolutionary analyses demonstrating a lack of phosphotyrosine counter-selection in species with tyrosine kinases. Our results suggest that, alongside the risk for toxicity, the cell can tolerate a large degree of non-functional crosstalk as interaction networks evolve.

**Keywords** Protein Kinases; Evolution; Phosphoproteomics; Structural Bioinformatics; Gene Editing
**Subject Categories** Evolution & Ecology; Post-translational Modifications & Proteolysis; Proteomics

## Introduction

Signalling pathways are often represented as perfectly specific systems, with a linear chain of directed interactions linking the primary signal to an effector enzyme or transcription factor. However, a number of biophysical and evolutionary considerations imply the existence of non-functional interactions in a dense protein network (Levy et al, 2009). Firstly, signalling interactions are often mediated by short linear motifs (SLiMs) that are degenerate in sequence and, therefore, may be created or destroyed by a small number of substitutions (Davey et al, 2015; Kliche et al, 2024). Secondly, members of a protein family may share protein folds and binding interfaces that risk illicit interactions between homologues of the functional binding partners (Nocedal and Laub, 2022; McClune and Laub, 2020). This risk is exacerbated when one considers the thousands of proteins that may interact in the cytoplasm (Levy et al, 2009), and the crowded cellular environment through which signalling occurs (Delarue et al, 2018; Nussinov et al, 2021; Li et al, 2021). While several mechanisms (co-expression, co-localisation, scaffolding, PTMs) can enhance functional specificity (Pawson, 2004; Scott and Pawson, 2009; Good et al, 2011; Miller and Turk, 2018), there is also evidence from PTM data for spurious interactions that contribute biological 'noise' to signalling networks (Levy et al, 2012; Landry et al, 2013; Hornbeck et al, 2015; James et al, 2018a, 2018b).

The optimum that a cell system can achieve is also influenced by evolutionary constraints at the level of the population. That is, whether a deleterious mutation can be removed by purifying selection and a beneficial one can reach fixation. The efficacy of selection is thus linked to the size of the population (higher in large populations), the strength of the selection pressure, and in the context of signalling the nature of the bias for a spontaneous mutation to either gain or lose a physical interaction (Lynch and Hagner, 2015; Lynch et al, 2016). Understanding how these constraints shape the evolution of signalling networks requires the fitness effect of spurious interactions to first be quantified (Levy et al, 2009).

In this context, we consider phosphotyrosine signalling in the budding yeast *Saccharomyces cerevisiae*. Yeast lacks classical tyrosine kinases, SH2 domains, or PTB domains; however, they contain a class of phosphatases called PTPs (protein tyrosine phosphatases) that dephosphorylate tyrosine with high intrinsic efficiency (Pincus et al, 2008; Hunter, 2009; Chen et al, 2017). A small amount of phosphotyrosine is present natively in yeast proteomes, but this is thought to derive from dual-specificity kinases (Manning et al, 2002; Pincus et al, 2008; Lim and Pawson, 2010; Kaneko et al, 2012; Leutert et al, 2023). Therefore, the heterologous expression of tyrosine kinases in this species is expected to generate many phosphotyrosine (pY) sites that are spurious by definition, some of which could have deleterious consequences. Indeed, expression of the hyperactive viral kinase v-SRC leads to toxicity in yeast (Brugge et al, 1987; Kornbluth et al, 1987)—an established result that has been replicated several times

[1]Institut de Biologie Intégrative et des Systèmes (IBIS), Université Laval, Québec, QC, Canada. [2]Department of Biochemistry, Microbiology and Bioinformatics, Université Laval, Québec, QC, Canada. [3]Quebec Network for Research on Protein Function, Engineering, and Applications (PROTEO), Université du Québec à Montréal, Montréal, QC, Canada. [4]Université Laval Big Data Research Center (BDRC_UL), Québec, QC, Canada. [5]Department of Biology, Université Laval, Québec, QC, Canada. [6]Department of Genome Sciences, University of Washington, Seattle, WA, USA. [7]Institute of Molecular Systems Biology, ETH Zürich, Zürich, Switzerland. [8]These authors contributed equally: David Bradley, Alexander Hogrebe. ✉E-mail: jvillen@uw.edu; christian.landry@bio.ulaval.ca

(Xu and Lindquist, 1993; Boschelli et al, 1993; Florio et al, 1994; Trager and Martin, 1997) and demonstrated in the fission yeast *Schizosaccharomyces pombe* for chicken c-Src (Superti-Furga et al, 1993). v-SRC mediated toxicity, in particular, is dependent on the activity of the Hsp90-Cdc37 chaperone complex in *S. cerevisiae* (Xu and Lindquist, 1993; Dey et al, 1996). In more recent years, this relationship between kinase activity and toxicity has been used as the basis for fitness-based deep mutational scanning (DMS) assays of the tyrosine kinase domain (Ahler et al, 2019; Chakraborty et al, 2024).

The spurious physical interaction between the human kinase and yeast substrate leaves a trace in the form of the phosphorylated residue, which serves as a chemical tag that can be assessed and quantified using mass spectrometry-based phosphoproteomics (Olsen et al, 2006; Villén et al, 2007; Rikova et al, 2007; Leutert et al, 2019). This phosphoproteomic approach has been used for the modelling of human tyrosine kinase specificity amid a minimal background of native pY signalling in *S. cerevisiae* (Corwin et al, 2017). More recently, a large panel of human kinases has been tested for fitness in yeast across several conditions and used to assay for phosphorylation-dependent interactions between human proteins (Jehle et al, 2022). Here we express several human tyrosine kinases in yeast alongside their kinase-dead counterparts to specifically focus on the effect of kinase activity. In each case, we assess the differences in fitness and phosphorylation between the WT tyrosine kinase and kinase-dead strains. This comparison allows us to relate spurious interactions to fitness, as each phosphosite is a remnant of the spurious physical interaction between the kinase and substrate. For this purpose, we also take advantage of AlphaFold-derived protein structure models and recent bioinformatic advances to perform proteome-wide variant effect prediction (VEP) for tyrosine phosphorylation on the basis of protein structure and protein conservation. Finally, with this phosphoproteomic data as a baseline of what could be potentially phosphorylated by these kinases in the absence of selection for function, we perform a deep evolutionary analysis of the pY sites and determine whether spurious Y phosphosites were selected against in species with bona fide tyrosine kinases.

## Results

### Expression of human kinases in yeast and their activity

We expressed 24 Y kinases and 7 S/T kinases in yeast by inserting the coding sequence into a landing pad in the yeast genome where the coding sequence is regulated by an inducible promoter (Fig. 1A; Table EV1). The set of tyrosine kinases included non-receptor tyrosine kinases that had been previously studied by (Corwin et al, 2017) and shown to be active (ABL2, SRC, SYK, BMX, FES, FRK, FYN, LYN, SRMS and TNK1), three additional non-receptor tyrosine kinases (ABL1, LCK, TEC), and several variants of v-SRC ($n = 13$) that span a range of kinase activities (Ahler et al, 2019). We also included several kinase domains of receptor tyrosine kinases (RTK) (EPHA1, EPHA2, EPHA3, EPHB3, EPHB1, EPHB4, FGFR2, FGFR3, MERTK and MET) and S/T kinases (IRAK4, NEK6, NEK7, RAF1, TBK1, TLK2 and VRK1). The S/T kinases were selected because they had no orthologs in yeast, their peptide recognition profiles overlapped minimally with the yeast phosphoproteome,

clones were readily available, and their activity can be increased by autophosphorylation (Beenstock et al, 2016). All oligonucleotides used for the construction of these strains are given in Dataset EV1.

The expression and activity of each tyrosine kinase or kinase domain (tagged with GFP) was examined by Western blots, using the protein extracts of yeast cells induced or not with estradiol (Appendix Figs. S1, S2). Detectable expression (anti-GFP) was found for most kinases tested (14/24). The activity of some of the enzymes was validated by monitoring the overall tyrosine phosphorylation of proteins (4G10; pan anti-phosphotyrosine), using kinase-dead (KD) point mutants as controls (10/24). For this control, we mutated the kinase catalytic aspartate to asparagine (D to N) as this is expected to have the strongest effect on activity while minimising kinase destabilisation (Reinhardt and Leonard, 2023). In addition, we analysed the expression and activity of the different v-SRC mutants used in this study. We found wild-type v-SRC to be well-expressed and active in yeast cells. Of the 13 mutants tested, 11 had observable phosphotyrosine signals and nine were sufficiently expressed to detect GFP bands at the correct molecular weights. Finally, for one kinase (EPHB1) we repeated the Western blots in the presence of a phosphatase inhibitor cocktail containing orthovanadate (a PTP inhibitor), demonstrating that pY levels can be increased by the inhibition of endogenous phosphatases (Appendix Fig. S3). Since most kinases were detected by Western blotting, we considered all kinases for downstream experiments even if their expression could be low. Almost all (~95%) of the tested kinases were detected in their phosphorylated form by mass spectrometry (section below), thus providing additional evidence of their expression in this system.

### Effects of human kinase expression on the yeast phosphoproteome

Phosphorylation in all strains was assessed across five biological replicates using data-independent acquisition mass spectrometry (DIA-MS) (Fig. 1A). Yeast proteins were extracted via bead-beating, reduced and alkylated, and digested into peptides by trypsin. Phosphopeptides were enriched by immobilised metal ion affinity chromatography (IMAC) using the automated R2-P2 method (Leutert et al, 2019) and MS-based measurement was performed using an Orbitrap Exploris 480. Across the 389 samples, we identified a total of 42,653 unique phosphopeptides comprising 29,106 unique phosphosites (Appendix Fig. S4). Data quality was high, with a median phosphopeptide enrichment efficiency of 97.3% and a median number of 9,585 identified phosphopeptides per sample. We observed good quantitative reproducibility between each of the five biological replicates per condition with a median Pearson correlation of 0.94 and a median coefficient of variation of 25.3%. The phosphoproteome of each WT kinase strain was compared with that of its kinase-dead mutant strain to determine significantly up- and downregulated phosphosites. In total, we found 4082 upregulated pY sites mapping to 1970 proteins and 9014 up- and downregulated pS/T sites mapping to 2361 proteins. Phosphorylation motifs of the WT kinases generally recapitulate motifs established in the literature (Colicelli 2010; Shah et al, 2018; Johnson et al, 2023) and thus validate the kinase activities (Appendix Figs. S5, S6). Phosphorylation profiles were variable between kinases (Fig. 1B,C) but overall were consistent with the kinase activity assays in Appendix Figs. S1, S2. Highly active

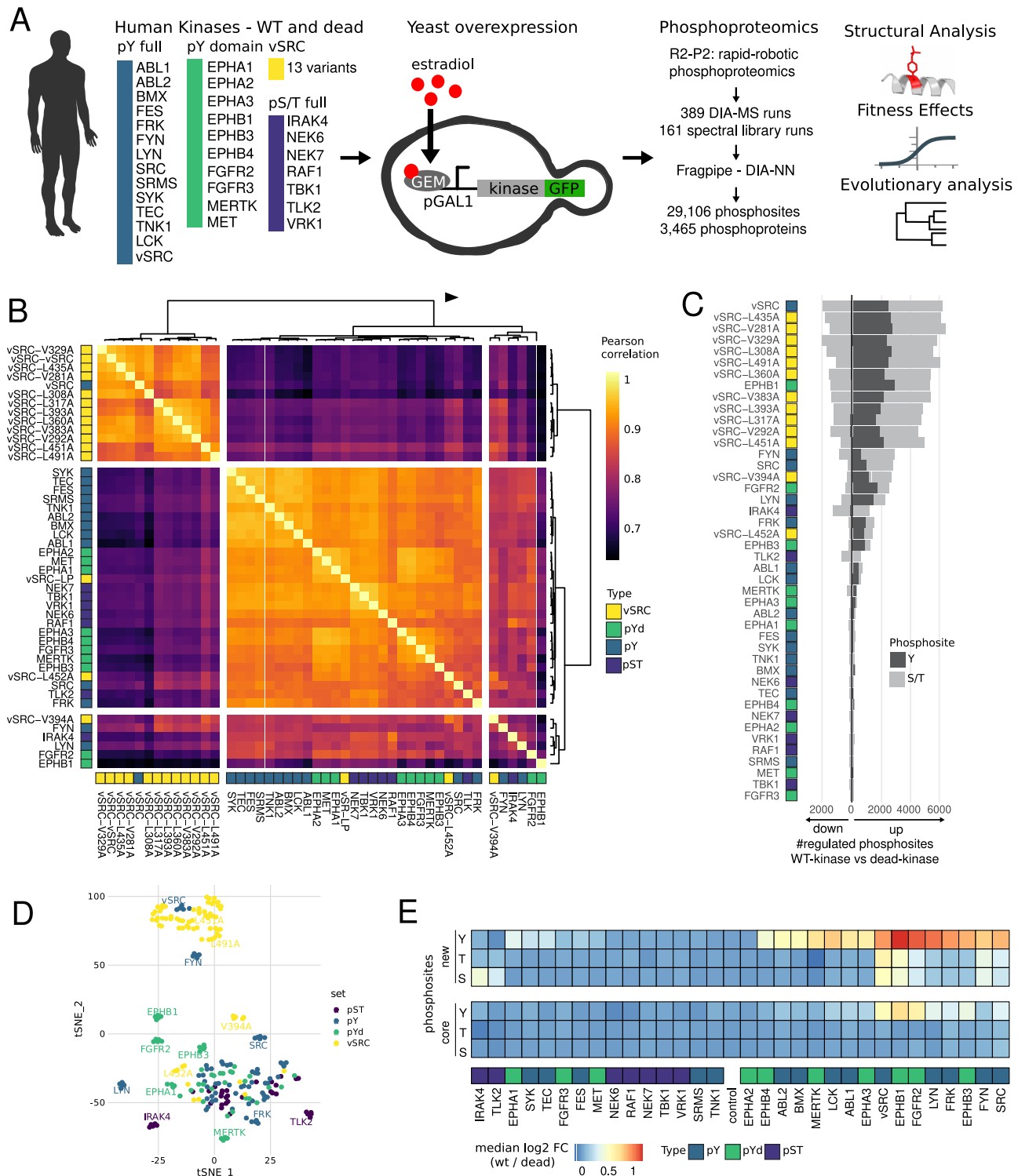

kinases such as v-SRC and EPHB1 generated over 2000 phosphosites, while others were indistinguishable from the dead mutant, suggesting that they failed to become active when expressed in yeast (Corwin et al, 2017; Jehle et al, 2022). The different kinase groups (full-length pY, kinase domain pYd, v-SRC mutants, and pST) can generally be distinguished using dimensionality-reduction methods such as tSNE and kinases with a similar number of targets (e.g. WT

v-SRC and EPHB1) showed distinct phosphorylation profiles (Fig. 1D). However, there is still substantial substrate overlap between groups. For example, around two-thirds of all sites phosphorylated by v-SRC were also phosphorylated by the EPHB1 kinase domain (pYd) (Fig. EV1; Dataset EV2).

The majority of induced pY sites have never been observed before, but there is also some upregulation of endogenous pY sites

**Figure 1.  Expression of human tyrosine kinases in yeast and detection of their substrates using mass spectrometry.**

(A) Inducible expression of human kinases from a genomic landing pad in *S. cerevisiae*, followed by data-independent acquisition (DIA) mass spectrometry. WT and kinase-dead mutants for 31 kinases, as well as 13 v-SRC variant mutants and controls, were grown in five biological replicates each ($n = 390$, one failed, three excluded, see Methods). After phosphoproteomics, the impact of phosphorylation on protein structure, fitness, and evolution is analysed. pY: full-length tyrosine kinase, pYd: tyrosine kinase domain, v-SRC: WT v-SRC and its mutants, pS/pT: full-length serine and threonine kinases. (B) Correlated phosphorylation profiles (Pearson's correlation coefficient) between all kinases tested (WT and v-SRC variants), based upon the median phosphosite intensity (pS/pT/pY) across replicates. (C) Number of up- and downregulated pY (dark grey) and pS/pT (light grey) sites per kinase. Up- and downregulation for each WT kinase is with respect to the kinase-dead mutant. (D) Separation of phosphorylation profile ratios (WT/variant vs kinase-dead, $n = 226$) in two dimensions using the tSNE dimensionality-reduction method. (E) Relative phosphosite abundance log2 (WT/dead) for each kinase, with respect to the phosphoacceptor identity (S, T, Y) and whether or not the phosphosite is a member of the core phosphoproteome in *S. cerevisiae* that is found to be phosphorylated in many conditions (Leutert et al, 2023). Source data are available online for this figure.

with established functions (Fig. 1E), for example, pY192 on the activation loop of the MAPK Slt2. Likewise, the regulated pS/pT sites that we observe upon kinase expression are distributed between those mapping to the 'core' S/T phosphoproteome and those that are condition-specific (Leutert et al, 2023) (Fig. 1E). Fold changes between Y kinases and their dead mutants are generally higher for Y sites than S/T sites, confirming the expectation that direct substrates are more strongly regulated than indirect downstream targets (Fig. 1E) (Kanshin et al, 2017).

## Structural profile of the spurious pY phosphoproteome

The recent availability of AlphaFold2 enables the structural analysis of mutations and PTMs on a proteome-wide scale (Jumper et al, 2021; Varadi et al, 2022). We take advantage of these models to perform a structural analysis across all of our pY protein substrates ($n = 1970$ proteins), with the goal of understanding how spurious phosphorylation may perturb the proteome.

We first calculate the relative solvent accessibility (RSA) for all upregulated pY sites directly from the AF2 structural models (Fig. 2A). Around a third of all pY sites are buried according to the AF2 structures (Fig. 2B), and we find similar results when we calculate buried content for a set of spurious pY sites identified previously (Corwin et al, 2017) (Fig. EV2A,B). We then predict the order/disorder content of our pY sites, using the AF2 structural models and the mean RSA in a ±12 amino acid window surrounding the pY position as a proxy for protein disorder (Akdel et al, 2022). We find that the average accessibility in this window tends to be low and therefore predict the majority of our pY sites (>80%) to map to ordered regions (Figs. 2C and EV2C,D), in agreement with a sequence-based prediction of spurious pY order/disorder content performed by Corwin et al (Corwin et al, 2017). To give context to our findings, we repeated this analysis on a recent reference pS/pT phosphoproteome in yeast (Leutert et al, 2023), revealing that our spurious pY sites are significantly more likely to be buried and ordered than endogenous pS/pT sites (Kolmogorov–Smirnov $p < 2 \times 10^{-16}$, Fisher $p < 2 \times 10^{-16}$, Fig. 2B,C). Interestingly, endogeneous pY sites ($n = 169$) lie between the extremes of spurious pY and endogenous pS/pT in terms of their accessibility and order (see section below 'Limited overlap between the spurious and native phosphoproteomes').

Phosphosites mapping to buried and ordered regions are likely to destabilise the native protein fold. We check this systematically by using FoldX to predict the change in free energy of folding ($\Delta\Delta G$ for Y -> pY) across all unique pY sites mapping to an AF2 structural model. We validate this approach by comparing $\Delta\Delta G$ predicted on a sample of AF2 models with $\Delta\Delta G$ from the

corresponding experimental structures ($n = 231$), revealing a strong correlation of 0.91 (Appendix Fig. S7A). We also find that, as expected, phosphorylation becomes more destabilising as pY sites become less accessible (Fig. EV2E) and that around 20% of all spurious pY are predicted to be destabilising using a standard $\Delta\Delta G$ threshold of 2 kcal/mol (Nishi et al, 2011; Wagih et al, 2018; Høie et al, 2022), a result that is consistent across almost all of the tested kinases (Appendix Fig. S7B). The $\Delta\Delta G$ values we predict are broadly similar to those based on experimental data for amino acid mutations (Nikam et al, 2021; Tsuboyama et al, 2023). However, stabilising phosphosites are rarely predicted here and we also observe a tail (95th percentile) of extreme $\Delta\Delta G$s above 7 kcal/mol (Fig. EV2F). Overall, this results in a large number of predicted destabilising pY across the proteome, especially for highly active kinases such as v-SRC and EPHB1 (Fig. 2D).

The spurious phosphosites generated may also perturb native protein–protein interactions in yeast. We map the spurious pY sites to such protein–protein interfaces, and again use FoldX to predict phosphosites that destabilise the protein interactions at the interface. We use Interactome3D for the structural annotation of protein–protein interfaces in *S. cerevisiae* (Mosca et al, 2013), in addition to a recent AF2-based screen that was performed proteome-wide for the prediction of protein complexes and their structures (Humphreys et al, 2021). We supplement this data with high-confidence machine-learning predictions from InteractomeInsider (Meyer et al, 2018), which predicts residues at the protein–protein interface (Appendix Fig. S7C). Only a small fraction of pY sites (7.2%) map to at least one structural interface although this percentage increases to 13.0% when also considering interfaces predicted by machine learning (Fig. 2E). The total number of interfaces (per pY) predicted by machine learning is generally much higher than what is observed from the structural models (Appendix Fig. S7D,E), indicating missing structural data in the *S. cerevisiae* proteome. However, a significant number of interactions are still predicted to be destabilised ($\Delta\Delta G > 2$ kcal/mol) from these structures due to specific perturbation at the interface (Fig. 2D). We experimentally validated one predicted destabilised interaction between the Rvs167 and Sla1 using the DHFR PCA assay (Fig. EV3). All structural parameters for the computational predictions are given in Dataset EV3 (pY), Dataset EV4 (pYd), Dataset EV5 (v-SRC), and Dataset EV6 (pS/pT).

In Fig. 2F, we present a structural profile of spurious pY phosphorylation covering accessibility, order/disorder, intramolecular destabilisation, and inter-molecular destabilisation (i.e. protein interactions). We also calculate the proximity between spurious sites (pY) and native sites (pS/pT), showing that around ~11% of spurious pY sites are proximal in 1D space to at least one

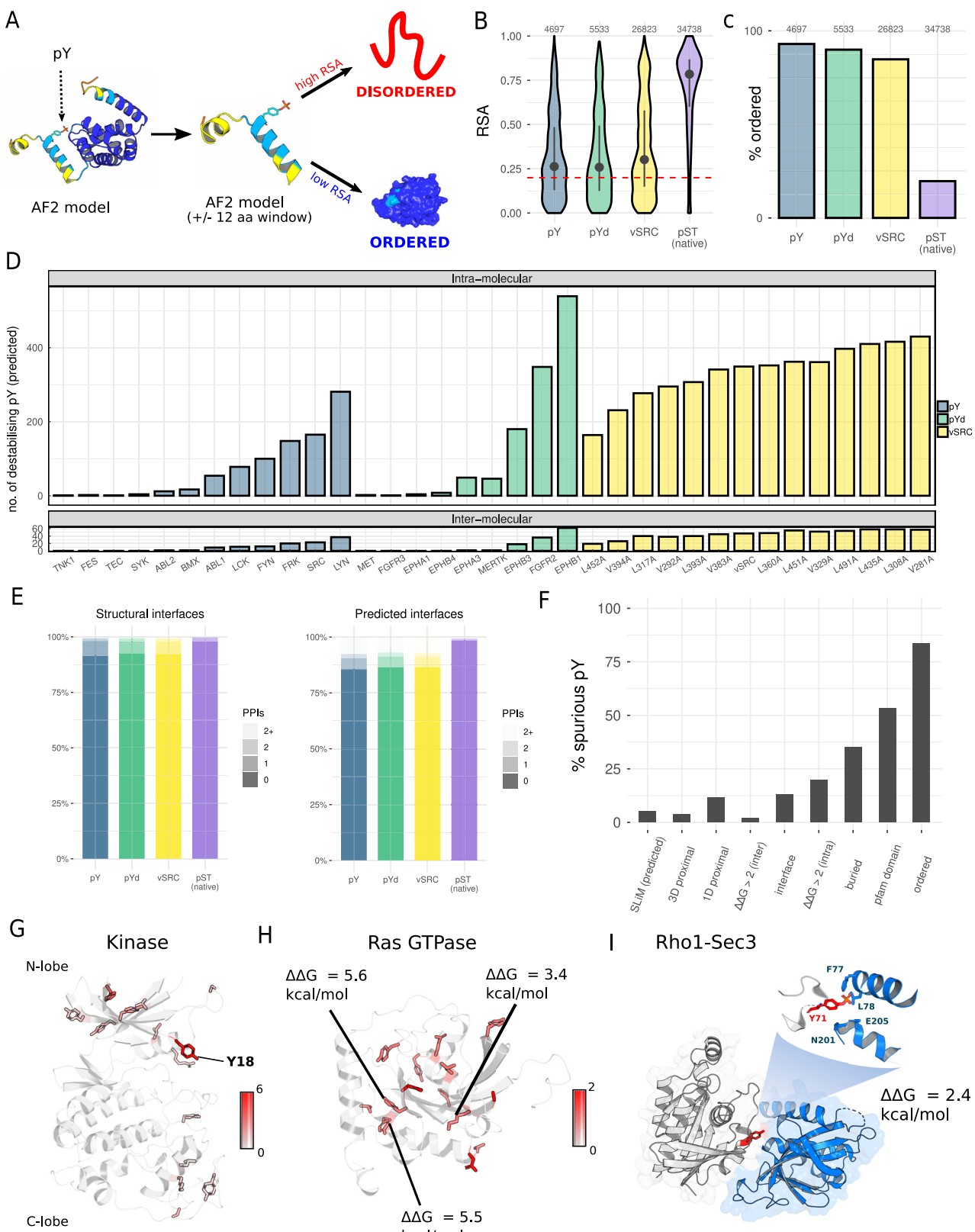

native pS/pT site, and around ~3.5% of spurious pY sites are proximal in 3D space to at least one native pS/pT site (Fig. 2F). We include the relative fraction of spurious phosphosites mapping to short linear motifs (SLiMs) also, which may mediate

protein–protein interactions (PPIs) with modular binding domains (Kumar et al, 2022). This analysis reveals that ~5.2% of spurious pY sites map to at least one predicted SLiM, and two putative examples are given in Appendix Fig. S7F,G. In terms of secondary structure,

Figure 2. Structural overview of spurious phosphorylation across the proteome.

(A) Use of AlphaFold2 (AF2) structural models to calculate the relative solvent accessibility (RSA) and disorder of spurious phosphosites. (B) The relative solvent accessibility (RSA) of spurious phosphosites generated by full-length tyrosine kinases (pY, $n = 4697$), tyrosine kinase domains (pYd, $n = 5533$), WT v-SRC and v-SRC mutants (v-SRC, $n = 26,823$), and the endogenous yeast pS/pT sites reported in (Leutert et al, 2023) (pST, $n = 34,738$). The red dashed line corresponds to the cut-off for buried residues, set at an RSA of 0.2. The points and bars correspond to the median and interquartile range. (C) The percentage of phosphosites that map to ordered regions for the pY ($n = 4697$), pYd ($n = 5533$), v-SRC ($n = 26823$) and pST groups ($n = 34738$). (D) For each of the tested kinases, the number of spurious pY sites (WT-dead) predicted to destabilise the protein fold (intra-molecular) or at least one protein–protein interface (inter-molecular), on the basis of a ΔΔG threshold of 2 kcal/mol. (E) For the pY, pYd, v-SRC and pST groups, the number of unique interfaces (per phosphosite) found in structural models (PDB, homology, or AF2) (left), or predicted from machine learning (right) (Meyer et al, 2018). (F) The structural profile of all unique pY phosphosites detected in this study. (G) The total number of unique pY sites that map to the protein kinase domain, represented by the AF2 model of the cyclin-dependent kinase Pho85. (H) The total number of unique pY sites that map to the Ras GTPase domain (including predicted destabilising pY), represented by the AF2 model of Gsp1. (I) The spurious phosphosite Rho1 pY71 is predicted to destabilise the Rho1-Sec3 interface with a ΔΔG of 2.4 kcal/mol (PDB: 3a58). Source data are available online for this figure.

the spurious phosphorylation profile does not differ from the random expectation (Appendix Fig. S7H). However, there is a weak but significant phosphorylation bias towards the protein N- and C-termini ($p = 1.9 \times 10^{-3}$, Kolmogorov–Smirnov test) (Fig. EV2G).

These data can be examined further to find examples of functionally significant proteins that may be perturbed by phosphorylation. For example, we find 27 unique spurious pY that map to the kinase domain of native *S. cerevisiae* kinases, reflecting the intrinsic specificity of kinases for other kinases as substrates (Invergo and Beltrao, 2018). Of these, 20 map to the smaller N-terminal lobe that is important for ATP binding and the regulation of kinase function (Pellicena and Kuriyan, 2006; Taylor and Kornev, 2011; McClendon et al, 2014), including a position in the glycine-rich loop that is frequently phosphorylated here and known to have an inhibitory effect in some kinases when natively phosphorylated (Steinberg, 2018) (Fig. 2G). The Ras GTPase family (including Ras/Rab/Ran/Ypt1 sub-families) is also critical for signal transduction and is spuriously phosphorylated at 19 unique sites in our data. These pY sites are more equally distributed in the 3D structure, but we predict 6 to destabilise the protein fold (Ras2 Y115, YPT32 Y101, YPT52 Y194, Rho5 Y216, Rho5 Y177 and GSP1 Y148), and 4 (Rho1 Y71, YPT1 Y109, Gsp1 Y149 and Gsp1 Y157) to map to interfaces covering 13 protein–protein interactions of known or predicted structure (Fig. 2H). In nine cases, the spurious pY is predicted to destabilise the protein–protein interface; in Fig. 2I we show as an example the destabilisation of the Rho1:Sec3 interface (PDB: 3a58, ΔΔG = +2.433 kcal/mol) required for targeting of the secretory protein Sec3 to the plasma membrane (Yamashita et al, 2010). Spuriously phosphorylated sites include positions homologous to pY32 and pY64 in human KRAS, which have inhibitory effects on RAS signalling (Kano et al, 2019; Wang et al, 2021), and are found here at pY37 for yeast Ypt1 and pY71 for yeast Rho1, proteins that have functions related to membrane trafficking and cell wall synthesis, respectively (Appendix Fig. S8). Finally, we give an example of spurious-native phosphorylation proximity in Tma19, a protein that stabilises microtubules and has apoptosis-related functions (Rinnerthaler et al, 2006). The spurious phosphosite pY18 is in close proximity (5.4 Angstroms) to a native phosphosite (pS15) (Holt et al, 2009) that is known to be functional because it gives a deleterious growth phenotype in methotrexate and ethanol conditions when mutated to alanine (S15A) (Appendix Fig. S7I) (Viéitez et al, 2022). In cases like this, the spurious phosphorylation may be either affecting the recognition of the native phosphosite or functionally mimicking it, as has been suggested for rapidly evolving protein regions (Holt et al, 2009).

In summary, our proteome-wide structural analysis of spurious pY predicts a potential for the widespread destabilisation of proteins and native protein–protein interactions.

## Effects of kinase expression on signalling

Given the large number of regulated endogenous pS/pT sites in the kinase overexpression conditions, we aimed at dissecting the regulation of underlying yeast kinases and pathways. Yeast kinase-substrate enrichment analysis (Leutert et al, 2023) indicated large effects for the master cell cycle kinase Cdc28 substrates (Fig. 3A). Depending on the overexpressed kinase, Cdc28 targets were either phosphorylated or dephosphorylated, indicating differential effects on the cell cycle. Cka2 and PKA kinases were activated in several tyrosine kinase overexpression conditions and the DNA damage kinases Mec1/Tel1 (human homologues ATM/ATR) were selectively activated in the TLK2 condition. Overall, the yeast kinase regulation pattern showed marked differences for the overexpressed kinases and did not resemble a classical environmental stress response.

Next, we analysed the phosphorylation of well-known activating tyrosine sites in yeast kinases to test how they are influenced by an overexpressed kinase (Fig. 3B). In many cases we found that overexpression of a tyrosine kinase led to increased phosphorylation of these critical tyrosine sites, many of them in activation loops of MAP kinases. To investigate if this tyrosine phosphorylation might happen through direct action by the overexpressed kinases or as a secondary consequence of pathway activation, we analysed known functional phosphorylation sites within the different MAPK pathways and their downstream targets (Fig. 3C). We found a striking pattern for several tyrosine kinases that showed no activation of upstream kinases, but activation of the effector kinase and phosphorylation of several downstream substrates. This indicates that some of the tyrosine kinases could activate endogenous MAPK signalling pathways in a non-canonical way, presumably through phosphorylation of critical tyrosine sites in the activation loop of MAPK effector kinases (Fig. 3D). However, we cannot exclude a more complicated mechanism (e.g. MAPK phosphatase inhibition) for these observations.

Finally, the analysis of pS/pT sites on kinase activation loops suggests the regulation of several other native kinases, especially in the v-SRC condition (Appendix Fig. S9). Flow cytometry analysis of the WT v-SRC strain indicated a sharp elevation in DNA content relative to the kinase-dead control, consistent with cell cycle dysregulation and mitotic block (Appendix Fig. S10) (Boschelli

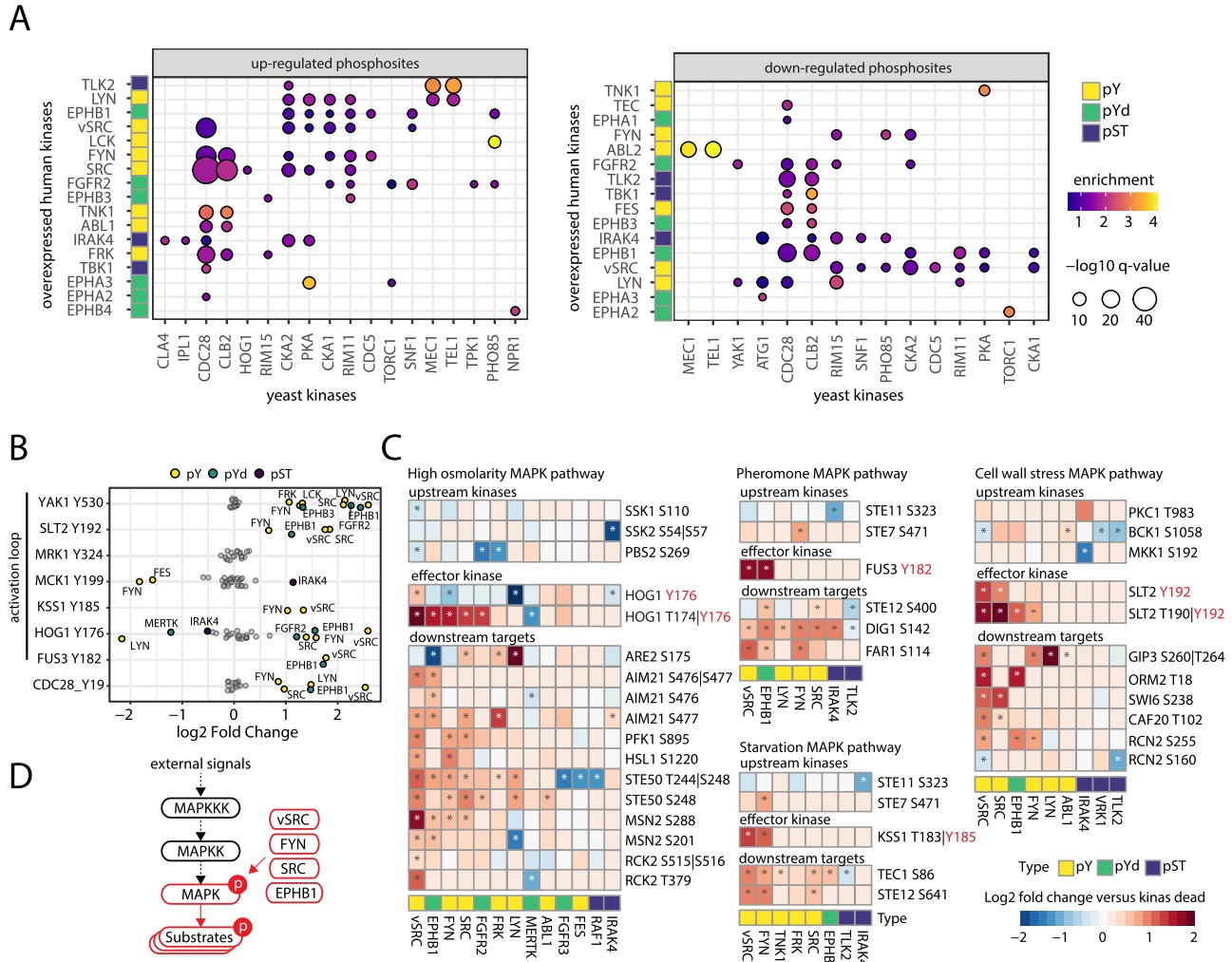

**Figure 3. Effects of kinase expression on signalling.**

(A) Yeast kinase-substrate enrichment analysis for endogenous phosphorylation sites that are significantly up- or downregulated in the different human kinase overexpression conditions. For kinases-substrate enrichment analysis, phosphosites were annotated with kinase-substrate relationships curated in (Leutert et al, 2023). Fisher exact tests with the whole phosphoproteome measured in this study as a background were performed. Benjamini–Hochberg multiple-hypothesis correction was applied and filtered for q values <0.01. (B) Scatter plot of log2 fold changes versus kinase-dead conditions for well-known tyrosine sites in yeast kinases. Conditions are colour-coded if the respective tyrosine phosphorylation site is significantly regulated. (C) Heatmap depicting regulation of known phosphorylation sites within the different yeast MAPK pathways across the overexpression conditions. Only phosphosites that are significantly regulated in at least one condition are shown. (D) Model that could explain interference of overexpressed kinases with the MAPK pathway.

et al, 1993). The effect of EPHB1 expression on genomic DNA content, however, is much weaker (Appendix Fig. S11) and further show that these kinases have distinct effects.

## Effects of kinase expression on fitness

We test the relationship between the activity of the kinases and their impact on fitness by measuring the growth of all strains expressing these kinases. The WT kinases are compared with their corresponding kinase-dead mutants to specifically determine the fitness effect of pY phosphorylation and not of heterologous protein expression. In total, the fitness effects of 44 kinases (13 Y kinases, 10 Y kinase domains, 7 S/T kinases, and 14 v-SRC variants) were tested across 41 conditions known to induce various stresses (Table EV2) and used previously in (Dionne et al, 2021).

We take the size of yeast colonies over time as a proxy for fitness and use an automated plate reader to measure changes in the colony size across 50 different time points (16 replicates per measurement). The area under the growth curve (AUC) was calculated for the WT kinase and inactive mutant, and then the difference in AUC between the two (WT-dead) was used to determine the fitness cost of pY phosphorylation for that kinase (Fig. 4A). A summary of the results are given in Fig. 4B,C; Appendix Fig. S12 and Dataset EV7. We find that 5/13 Y kinases, 3/10 Y kinases domains, 3/7 S/T kinases, and 14/14 v-SRC variants are significantly more deleterious (FDR-adjusted) for growth in at least one condition when comparing the WT kinase with the corresponding kinase-dead mutant. The kinase-dead strains generally have similar colony sizes (Appendix Fig. S13), demonstrating a comparable impact upon fitness even if some were poorly

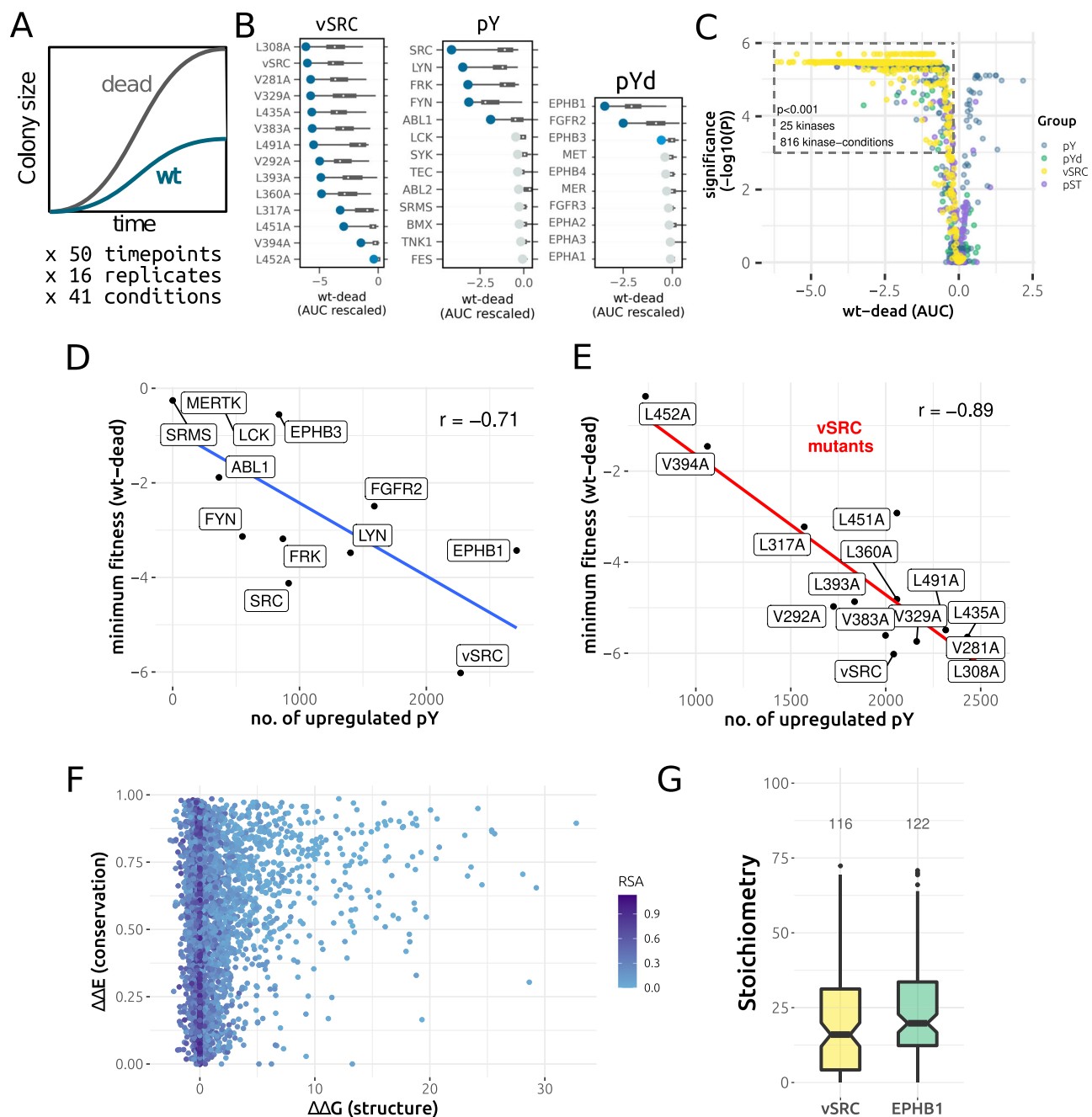

expressed in the Western blots (Appendix Fig. S1 and S2). Direct competition assays were performed for the pY and v-SRC strains against an empty landing-pad control and used to determine the relative growth of the kinase (Appendix Fig. S14; Dataset EV8, see methods), giving results that correlate significantly with the $AUC_{min}$ values derived from the colony sizes ($r_s = 0.70$, $p = 9.85 \times 10^{-5}$) and thus validate these assays.

We find that the toxicity of spurious phosphorylation can be sensitive to the conditions of growth (Appendix Fig. S15A), as shown previously in (Jehle et al, 2022). We also find that different conditions are highly correlated in terms of their fitness effect across kinases (Appendix Fig. S15B) and likewise, that kinases with similar activity levels correlate strongly across conditions (Appendix Fig. S15C),

suggesting few kinase-condition interactions overall. However, strong interactions can be observed for a small number of kinase-condition pairs (Appendix Fig. S15D). For example, the v-SRC mutation L491A has a minimum WT-dead fitness in the SDS 0.02% condition that is 3.5 times lower than the median fitness effect across all conditions (Appendix Fig. S15D). Differential fitness effects between conditions (Appendix Fig. S15E) could be explained by loss-of-function (LOF) phosphorylation effects mapping to conditionally sensitive genes (Bosch-Guiteras and van Leeuwen, 2022). However, cross-referencing the data here with a systematic conditional KO screen in *S. cerevisiae* (Viéitez et al, 2022) provides little support for this hypothesis (Appendix Fig. S15F). The mechanism underlying the conditional fitness defects, therefore, remains unclear.

**Figure 4. Measuring the fitness effect of human kinase expression in yeast.**

(A) Colony size is used as a proxy for fitness and the difference in area under the growth curve (AUC) between the WT kinase and the kinase-dead mutant is used to infer the fitness effect of spurious phosphorylation. (B) The fitness for each kinase tested, represented by the WT-dead fitness defect across 41 conditions. More negative AUC values (WT-dead) indicate greater toxicity. The colours of the dots indicate the statistical significance of the fitness score for the condition of minimum fitness (the minima): dark blue indicates $P < 0.001$, light blue indicates $P < 0.05$, and grey indicates non-significance. $P$ values (FDR-adjusted) were calculated from a Mann–Whitney $U$-test and are listed for all kinases and conditions in column R of Dataset EV7. The small white dot corresponds to the median (50th percentile), the bounds of the box to the lower quartile (25th percentile) and upper quartile (75th percentile), and the whiskers correspond to 1.5 x the interquartile range below and above the lower and upper quartiles. $n = 41$ for each kinase. (C) Volcano plot of WT-dead AUC (x-axis) against the FDR-adjusted $p$ value for the significance of the difference between the WT and kinase-dead mutant. pY: full-length tyrosine kinase, pYd: tyrosine kinase domain, v-SRC: v-SRC and mutants, pST: serine/threonine kinases. $P$ values (FDR-adjusted) were calculated from a Mann–Whitney $U$-test and are listed for all kinases and conditions in column R of Dataset EV7, $n = 1804$ (41 conditions × 44 kinases). (D) Correlation between the number of spurious pY (per kinase) and the minimum fitness (WT-dead) across conditions for all non-redundant kinases tested here. The Pearson's correlation coefficient is indicated in the plot. (E) Correlation between the number of spurious pY (per kinase) and the minimum fitness (WT-dead) across conditions for WT v-SRC and the v-SRC mutants tested here. The Pearson's correlation coefficient is indicated in the plot. (F) Scatter plot for the predicted effect of spurious pY on protein structure ($\Delta\Delta G$) and predicted effect on the basis of sequence conservation ($\Delta\Delta E$). More destabilising pY have higher $\Delta\Delta G$s and more conserved Y positions have values closer to 1. This includes all unique upregulated pY sites (WT-dead) where a structural and conservation analysis could be performed ($n = 3865$). (G) Distribution of stoichiometries for v-SRC pY substrates ($n = 116$) and EPHB1 pY substrates ($n = 122$), where significant phosphosite regulation was inferred ($q < 0.01$, see Methods). The black band corresponds to the median (50th percentile), the bounds of the box to the lower quartile (25th percentile) and upper quartile (75th percentile), and the whiskers correspond to 1.5 x the interquartile range below and above the lower and upper quartiles. Source data are available online for this figure.

We next relate the phosphoproteomic data presented above (Fig. 1) with the fitness data for each kinase. We find that the minimum fitness defect per kinase correlates strongly with the number of upregulated pY sites for that kinase, both across kinases (Fig. 4D) and between the v-SRC mutants (Fig. 4E). Similar results were found when using the median fitness defect across conditions (Appendix Fig. S16A and Appendix Fig. S16B).

It is unclear whether this strong correlation represents a general phosphorylation burden distributed across many sites or, at the other extreme, a very small number of deleterious pY that increase in stoichiometry as the kinase activity also increases. To support the first hypothesis we try to determine if the fitness-phosphorylation correlation can be improved by considering only those pY that we predicted to destabilise the protein fold (Fig. 2). However, the resulting correlation ($r = -0.64$) is lower than that calculated from all pY sites ($r = -0.71$). We next account for sequence conservation by performing variant effect prediction (VEP) for all upregulated pY sites (~4000 sites on ~2000 proteins), achieved by retrieving substrate homologues across the protein universe and predicting the phosphorylation effect on the basis of a global epistatic model (Laine et al, 2019) (Figs. 4F and EV4). However, incorporating this data on predicted deleterious effects offers no improvement to the fitness-phosphorylation correlation across a range of structural and conservation thresholds (Appendix Fig. S16C,D).

Likewise, incorporating information on gene essentiality (Appendix Fig. S16E) or predicted protein–protein interfaces (Appendix Fig. S16F,G) has generally little effect on the fitness-phosphorylation correlation. Conservation ($\Delta\Delta E$) and structural parameters ($\Delta\Delta G$) for all pY phosphosites are presented in Figs. 4F and EV4, and listed in Dataset EV9 alongside their surface accessibilities.

Finally, we consider that the Y kinase domain EPHB3 has ~800 upregulated pY sites, of which 191 are predicted to be destabilising and 125 are predicted to be deleterious on the basis of sequence conservation (Figure EV4; Appendix Fig. S17). However, expression of this kinase has no significant fitness effect in most conditions (Dataset EV7; Appendix Fig. S16A) and only a small effect for the condition of minimum fitness (Fig. 4D). The results taken together suggest that a significant fraction of spurious pY have a negligible fitness effect in spite of the in silico predictions. One possibility is that many of the pY detected by mass spectrometry are of low stoichiometry, as steady-state phosphorylation levels in vivo reflect a balance between protein synthesis, degradation, phosphorylation by kinases and dephosphorylation by phosphatases (Hunter, 2009). We probe this point further by performing a general measurement of phosphorylation stoichiometry across substrates. As we use a regulation-based approach to infer stoichiometry (see Methods), we choose the two most strongly active kinases in our dataset: EPHB1 and v-SRC (Fig. EV4; Appendix Fig. S17). The relatively high abundance of phosphorylated peptides in these conditions ensures high regulation between wt and kinase-dead, which results in more accurate stoichiometry estimations (Appendix Figs. S18, S19). For the phosphosites that are sufficiently regulated between the WT and dead conditions, we infer a median stoichiometry of 16.0% for v-SRC and 19.8% for EPHB1 (Fig. 4G), suggesting that the effect of phosphorylation would be much weaker than if the equivalent positions were mutated in the genome (i.e. at 100% effective stoichiometry). Indeed, haploinsufficiency screens in yeast suggest that even the removal of one gene copy (~50% protein reduction) does not have a measurable fitness effect for the vast majority of genes (Deutschbauer et al, 2005). While some predicted deleterious pY are still found at intermediate-high stoichiometries and are highlighted in Appendix Fig. S20 (v-SRC and EPHB1), we do not generally observe a decrease in abundance for the substrates of such sites (Appendix Figs. S21, S22). Stoichiometry values are given in Dataset EV10.

## Limited overlap between the spurious and native phosphoproteomes

All kinases used in this study (except v-SRC) have native substrates in the human proteome that act as functional effectors of the kinase. For spurious phosphorylation, it is an open question whether the non-native kinase will preferentially phosphorylate the homologues of its native substrates or, instead, random proteins across the proteome. We address this question first by mapping all yeast-human orthologs where possible for the spurious pY substrates we identified. These data reveal that spurious substrates are more likely to have human orthologs than random S. cerevisiae proteins (Fig. 5A, $p = 9.8 \times 10^{-9}$, odds ratio = 1.5, Fisher test). Moreover, we find that human orthologs of yeast proteins are more

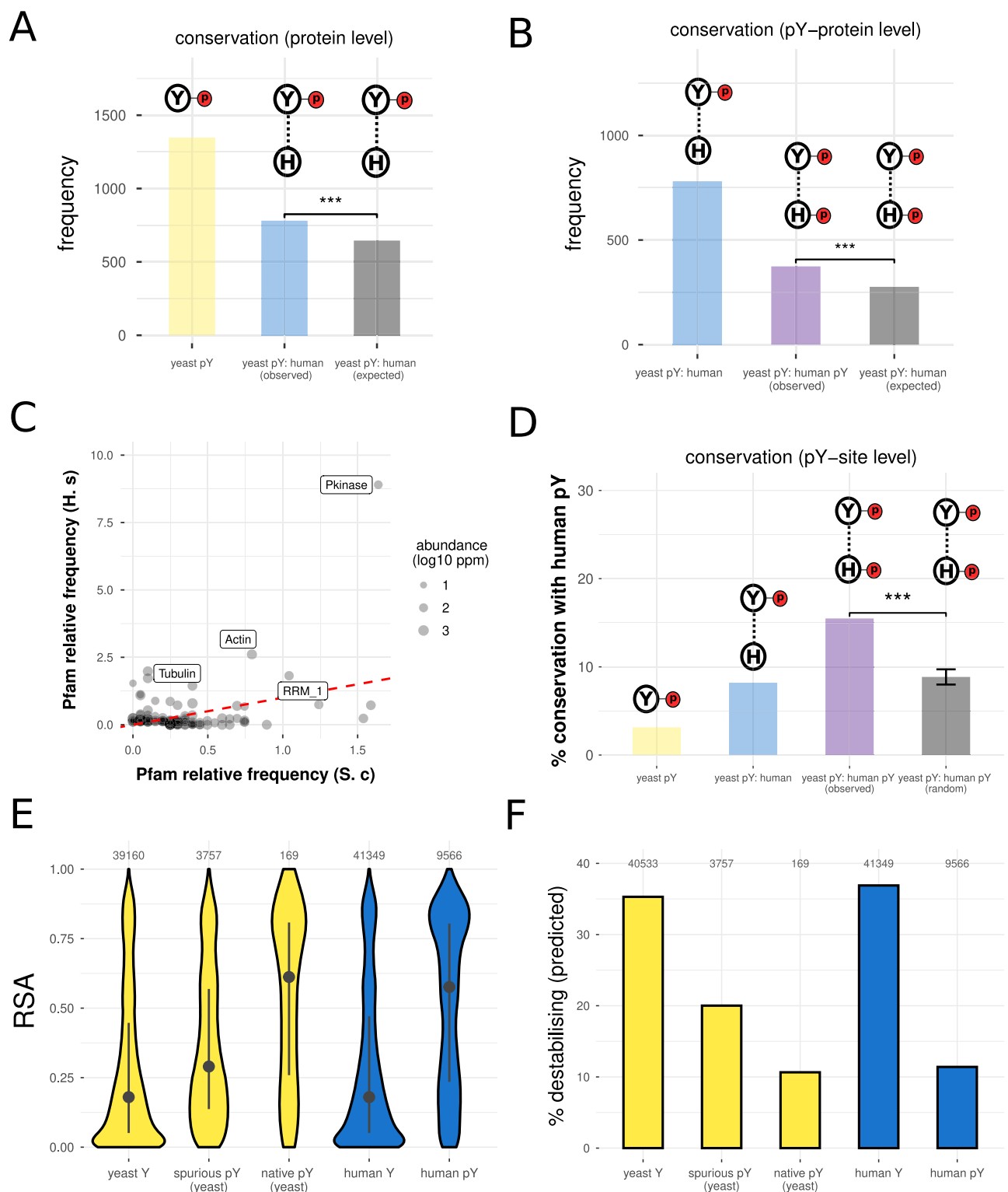

likely to be tyrosine-phosphorylated in humans if the yeast protein was tyrosine-phosphorylated in this study (Fig. 5B, $p = 1.3 \times 10^{-7}$, odds ratio = 1.68, Fisher test). For native pY sites in yeast, there is no protein-level conservation (Appendix Fig. S23A) but the phosphorylation state is highly conserved, meaning that the

orthologs of native pY proteins in yeast are highly likely to also be phosphorylated in humans (Appendix Fig. S23B).

For cases where the pY state is conserved between human and yeast orthologs, we use known kinase-substrate relationships to determine if the sites are phosphorylated by the same kinase in

**Figure 5. Conservation between spurious pY in yeast and native pY in human.**

(A) Human–yeast conservation at the whole protein level. 'Yeast pY': number of unique spurious pY proteins in yeast, excluding any native pY proteins. 'Yeast pY: human (observed)': number of observed unique spurious pY proteins in yeast with at least one ortholog in humans. 'Yeast pY: human (expected)': number of expected unique spurious pY proteins in yeast with at least one ortholog in humans. Expected frequencies derived from the number of observed orthologs for yeast proteins that are not tyrosine-phosphorylated. H symbol: human. Y symbol: yeast. P symbol: phosphosite. Dotted lines represent orthology relationships. Significance determined by a Fisher exact test. '****' indicates $p < 0.01$. The exact $p$ value is $9.8 \times 10^{-9}$. (B) Human–yeast conservation at the level of whole protein pY phosphorylation. 'Yeast pY: human': number of observed unique spurious pY proteins in yeast with at least one ortholog in humans. 'Yeast pY: human pY (observed)': number of observed unique spurious pY proteins in yeast with at least one ortholog in humans that is Y-phosphorylated. 'Yeast pY: human pY (expected)': number of expected unique spurious pY proteins in yeast with at least one ortholog in human that is Y-phosphorylated. Expected frequencies derived from the number of observed tyrosine-phosphorylated orthologs for yeast proteins that are not tyrosine-phosphorylated. H symbol: human. Y symbol: yeast. P symbol: phosphosite. Dotted lines represent orthology relationships. Significance determined by a Fisher exact test. '****' indicates $p < 0.01$. The exact $p$ value is $p = 1.3 \times 10^{-7}$. (C) Relative frequency (summing to 100) of Pfam domain phosphorylation in yeast (x-axis) and humans (y-axis). Abundance is given in the units of log10 (parts per million). The scatter plot includes only domains supported by at least five unique pY sites in either human or yeast ($n = 136$). The dashed red line corresponds to the y = x line through the origin. (D) Site-specific (i.e., alignment-based) conservation between spurious pY and human native pY. As a percentage of all unique yeast spurious pY (yeast pY, yellow), all unique yeast spurious pY with at least one human ortholog (yeast pY: human, blue), all unique yeast spurious pY with at least one Y-phosphorylated human ortholog (yeast pY: human pY (observed), purple), and all unique yeast spurious pY with at least one pY-phosphorylated human ortholog with x100 randomisations of the human pY positions. The error bar represents the standard deviation, and significance was determined from an empirical $p$ value. '****' indicates $p < 0.01$. $p$ value is <0.01 as it was determined empirically from permutation ($n = 100$) and not from a formal statistical test. (E) For yeast non-pY Y residues ($n = 39160$), spurious pY residues ($n = 3757$), native pY residues ($n = 169$), human non-pY Y residues ($n = 41349$) and human pY residues ($n = 9566$), the distribution of surface accessibility (RSA). The $p$ value is extremely small and approximated as 0 from the statistical test (Kruskal–Wallis test). The black point indicates the median (50th percentile) and the vertical lines connect the lower and upper quartiles (25th percentile and 75th percentile). (F) For yeast non-pY Y residues ($n = 40,533$), spurious pY residues ($n = 3757$), native pY residues ($n = 169$), human non-pY Y residues ($n = 41,349$) and human pY residues ($n = 9566$), percentage predicted to be destabilising for the protein fold (using a threshold of $\Delta\Delta G > 2$ kcal/mol)). The $p$ value is extremely small and approximated as 0 from the statistical test (Kruskal–Wallis test). Source data are available online for this figure.

both species. While the completeness of KSR annotations is limiting for this analysis (Invergo and Beltrao, 2018; Needham et al, 2019), this data reveals that the extent of conservation between human and yeast KSRs is very low across all kinases (Appendix Fig. S23C). We broaden this analysis by examining the conservation of protein domain preference among the pY sites in humans and yeast (Fig. 5C), revealing conserved phosphorylation on a small number of domains, including the kinase, actin, and tubulin domains. However, 32% of all spuriously phosphorylated domains in yeast are not observed to be tyrosine-phosphorylated in humans, revealing the recurrent spurious phosphorylation of non-native domains.

While the phosphorylation state can be conserved at the level of the whole protein or protein domain (as described) the effect of phosphorylation on protein function may differ depending on the position of the phosphosite in the protein sequence. We, therefore, check the extent to which yeast pY (spurious) and human pY (native) align at the site-specific level. The overall level of site-specific conservation across all spurious pY is very low (3.10%). As expected, the percentage conservation increases (to 8.20%) when considering only spurious pY with at least one human ortholog, and more so to 15.49% when considering only spurious pY with at least one human ortholog that is also phosphorylated in humans (i.e. pY conservation at the protein level) (Fig. 5D). Through 100 random permutations of the phosphosite positions we show that this percentage is significantly higher than the chance expectation (Fig. 5D). This supports the previous finding that spurious pY sites in yeast are significantly more tyrosine-conserved in animals and their unicellular relatives than yeast non-pY sites (Corwin et al, 2017). The observed extent of site-specific conservation is modestly improved when considering proximal pY in an alignment window (–/+0, –/+3, –/+5, –/+ 7, –/+9 alignment positions), which may reflect a slight under-estimation of site-conservation caused by alignment errors, phosphosite mislocalisation, or selectively neutral shifts in phosphosite position (Landry et al, 2014) (Appendix Fig. S23D).

Spurious pY sites in yeast tend to map to buried and ordered protein regions, as discussed (Fig. 2). We repeat this structural analysis for native human pY sites to check if shifts in phosphosite structural preference can explain the low level of pY conservation at the site-specific level between yeast and humans. For context, we also perform this analysis for yeast non-phosphorylated Y, human non-phosphorylated Y, and endogenous pY in yeast. The results reveal that, compared to yeast spurious pY, human native pY is significantly more likely to be accessible (Fig. 5E) and disordered (Appendix Fig. S23E), and therefore less likely to destabilise the protein structure, confirmed using $\Delta\Delta G$ predictions in silico (Fig. 5F). In turn, yeast spurious pY is more accessible, more disordered, and less destabilising than the expectation for non-phosphorylated Ys in the proteome, given that we predict ~20% of our spurious pY to be destabilising compared to a rate of ~35% for Ys not modified according to mass spectrometry. The data, therefore, demonstrate significant structural differences between the spurious and native phosphoproteomes in yeast and humans (respectively), and also between spurious phosphorylation and the null expectation for random (non-modified) Ys, revealing some structural constraint upon spurious phosphorylation (Fig. 5E,F, 'yeast Y'). However, these observations poorly explain low site-conservation as pY sites that do not align have consistent order/disorder predictions 79% of the time (Appendix Fig. S23F).

Taken together, this analysis reveals that spurious phosphorylation is weakly biased towards homologues of native kinases substrates, but there is also significant phosphorylation of completely new (i.e. non-homologous) substrates. Spurious-native pY conservation is low at the level of the whole proteins and even more so at the site-specific level, although it is still higher than the expectation for completely random Y phosphorylation.

## Tyrosine counter-selection in metazoan proteomes

Tyrosine kinases are absent from fungal species and the small level of detectable Y phosphorylation on fungal proteins likely arises

from dual-specificity kinases (Lim and Pawson, 2010; Corwin et al, 2017; Leutert et al, 2023). It has been proposed that the emergence of bona fide Y kinases in animals resulted in the selective loss of Y content in their proteomes to avoid spurious phosphorylation (Tan et al, 2009, 2011), a claim that has been challenged by later studies (Su et al, 2011; Pandya et al, 2015; Kritzer et al, 2018). Having at hand a set of sites that could be phosphorylated if such kinases were present in fungi, we can examine further this model. We explicitly test this hypothesis at the level of the proteome (Fig. 6A,B; Appendix Fig. S24), proteins (Fig. 6C,D), linear motifs (Appendix Fig. S25), and individual sites (Figs. 6E–G and EV5) using the spurious pY data generated in this study.

We are able to recapitulate the previously observed negative correlation between the number of predicted tyrosine kinases and proteomic Y content ($r^2 = 0.69$ compared to $r^2 = 0.66$ in (Tan et al, 2009), Appendix Fig. S24A), using available reference proteomes and a state-of-the-art method for kinase prediction and classification (see Methods). We then take advantage of the increased availability of genome sequences in recent years to repeat this analysis across a larger set of species that is more phylogenetically balanced and representative of major phyla across the metazoa and fungi (Appendix Fig. S24B). Using this new data, we find no significant relationship between the number of predicted tyrosine kinases and proteome Y content ($r^2 = 0.006$, Fig. 6A). We also generated a species tree to calculate the evolutionary distance between species (Appendix Fig. S24B), which allows this analysis to be repeated while accounting for the branch lengths connecting data points. The original correlation remains statistically significant after controlling for phylogenetic non-independence ($r^2 = 0.52$, $p = 0.002$, Appendix Fig. S24C, in agreement with (Tan et al, 2011), while we still find no significant relationship between the number of tyrosine kinases and proteome Y content for our more recent set of species after applying this control ($r^2 = 0.036$, $p = 0.376$, Appendix Fig. S24D). The analysis was repeated after stratifying the data according to 'buried/accessible' status of the pY sites using proteome-wide AF2 models (across all species) but again without a significant correlation (buried: $r^2 = 0.0047$; surface: $r^2 = 0.0002$; Fig. 6B). Finally, we test the expectation that orthologs of the spurious substrates identified here will be subject to stronger selection against spurious phosphorylation than non-substrates, assuming some conservation of structure, abundance, and sub-cellular localisation between animal and fungal species. While we indeed observe a stronger correlation for pY orthologs ($r = -0.25$, $p = 0.24$, Appendix Fig. S24E left-top) compared to the rest of the proteome ($r = -0.19$, $p = 0.38$, Appendix Fig. S24E left-bottom), larger differences are found for other amino acids such as aspartate ($r_{ortho} = -0.297$, $r_{non-ortho} = -0.015$, Appendix Fig. S24E-middle; Dataset EV11). Taken together, these results fail to strongly support the hypothesis of proteome-wide pY counter-selection in animal species.

Evidence for pY counter-selection may instead be present at the level of individual proteins. The presence of PTM 'deserts' has been suggested as a mechanism for the avoidance of off-target modifications (Fredrickson et al, 2013; Boomsma et al, 2016; Sharma et al, 2017), prompting recent studies into lysine-depleted regions and their relation to spurious lysine ubiquitination and degradation by the proteasome (Kampmeyer et al, 2023; Szulc et al, 2023). By analogy, we investigate the possibility of 'tyrosine deserts' that may evade tyrosine phosphorylation. At the same time, we

examine deserts for the other 19 amino acids. Differences in the proteomic amino acid composition and disorder content across species bias the expected number of amino acid deserts under selective neutrality. We therefore randomly simulate each proteome 100 times, while accounting for the amino acid and disorder content of each species, and then compare the expected number of amino acid deserts with those observed from real sequences (Fig. 6C). We find on average a 3.29% increase in tyrosine deserts beyond the neutral expectation, behind lysine (3.41%), histidine (4.66%), tryptophan (6.87%), methionine (7.07%) and cysteine (10.65%) (Dataset EV12). However, tyrosine desert frequency does not correlate with the absolute or relative number of tyrosine kinases in the kinome (Fig. 6D). We also do not observe strong evidence for pY counter-selection after checking for the depletion of tyrosine-containing motifs '(avoided words') in human proteins not tyrosine-phosphorylated but that are orthologs of spurious pY substrates in *S. cerevisiae* (Koulouras and Frith, 2021; Georgakopoulos-Soares et al, 2021) (Appendix Fig. S25).

Finally, in the absence of support for tyrosine counter-selection at the level of the whole proteome, selection against tyrosine residues may still be evident at the level of individual phosphosites that were significantly upregulated between the WT and dead conditions. We test this by checking for the preferential counter-selection of *S. cerevisiae* sites detected as Y-phosphorylated here, compared to non-phosphorylated tyrosines and while controlling for surface accessibility. For this purpose we use an evolutionary model that takes as an input a multiple sequence alignment (MSA) and phylogenetic tree of spuriously phosphorylated substrates and their orthologs across animal and fungal species. This software (*Pelican,* (Duchemin et al, 2023)) aims to detect shifts in amino acid preferences between lineages by modelling each clade using a separate equilibrium frequency vector of amino acids (Fig. 6E). Using this approach, we determine whether an equilibrium depletion of tyrosine in animal species offers a significantly better fit to the data than using a uniform equilibrium vector across all species (animal and fungal).

From this point, we refer to shifts in amino acid equilibrium frequencies inferred from the phylogeny and MSA as shifts in the amino acid 'preference'. At a false discovery rate (FDR) of 0.05, around ~35% of all Y positions among the spurious substrates detected here exhibit a significant change in amino acid profile between animal and fungal sequences (Fig. EV5A). We then test specifically for shifts in Y preference by comparing the Y preference in animals with the Y preference in fungal species ($Y_{\pi\text{-fungi}} - Y_{\pi\text{-metazoa}}$). Values close to 1 correspond to cases where Y is strongly preferred in the fungi but not in the metazoa, and vice versa for negative values. We also stratify the data according to accessibility (buried, intermediate, exposed) given established relationships between residue exposure and the rate of evolution (Echave et al, 2016; Bricout et al, 2023), and that the molecular impact of phosphorylation will likely depend upon the accessibility of the phosphosite. This analysis does not reveal any significant evidence for Y counter-selection for the pYs detected in this study relative to non-pYs (Fig. 6F; Dataset EV13). It is possible that some of the 'non-pY' tyrosines in this analysis are susceptible to phosphorylation but were simply not targeted among any of the active Y kinases used to generate this data. We, therefore, repeat this analysis but only using non-pY sites with a low motif score

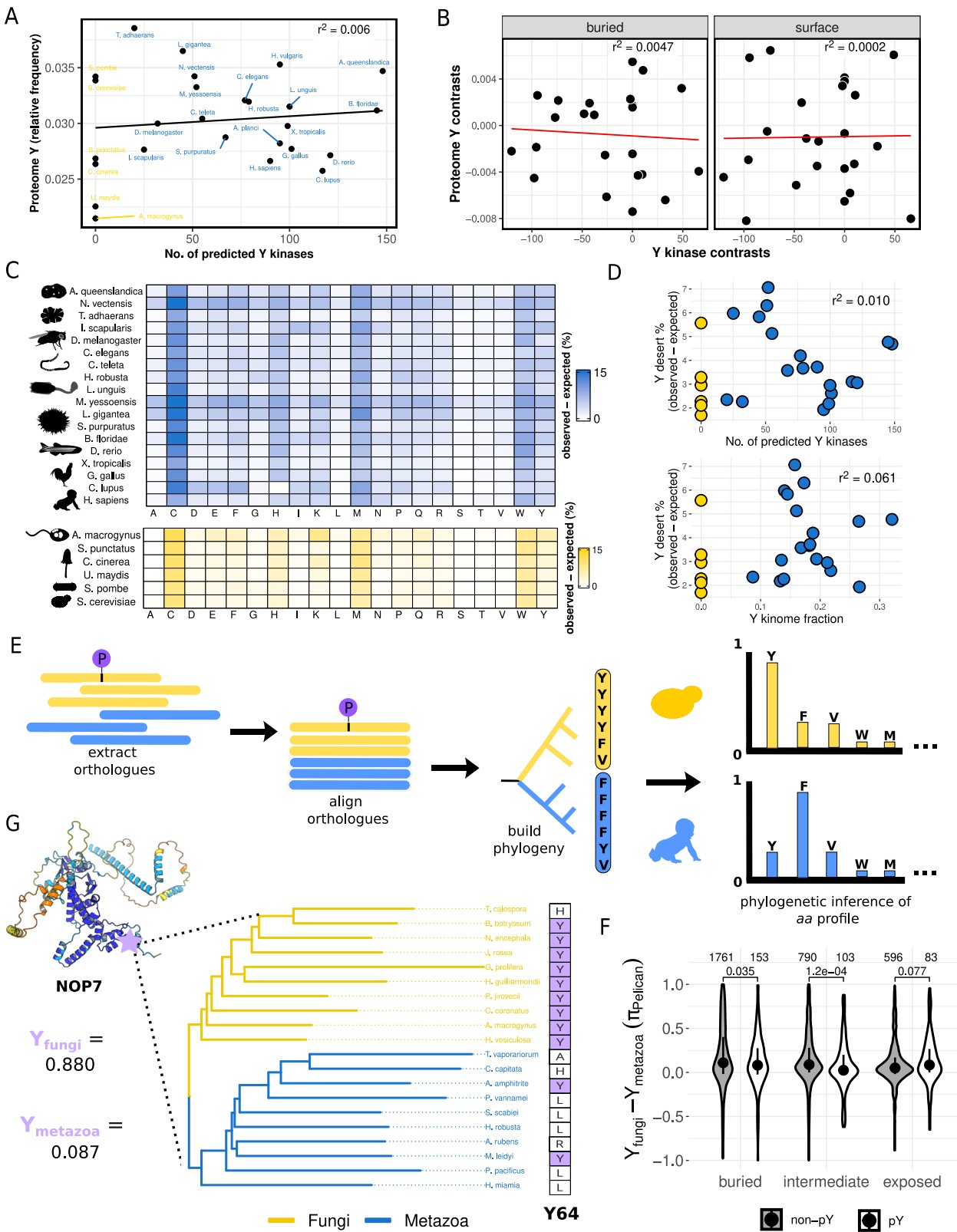

◀   **Figure 6.   Testing for counter-selection against spurious pY residues in animal species.**

(A) Spearman's correlation between the number of predicted tyrosine kinases and proteomic Y content (relative frequency) for 20 animal species (blue) and 6 fungal species (yellow). The $r^2$ coefficient is shown in the panel. (B) Pearson's correlation between the number of predicted tyrosine kinases and proteomic Y content (relative frequency) after applying a phylogenetic correction (phylogenetically independent contrasts, (Felsenstein, 1985)). The left panel represents buried tyrosines (RSA <0.2) proteome-wide and the right panel surface tyrosines (RSA >0.4) proteome-wide. The species and kinases analysed are the same as in panel (A). The $r^2$ coefficient is shown in both panels. (C) Percentage of amino acid deserts (observed - expected) for all 20 canonical amino acids. An amino acid desert is defined as a protein where more than 50% of the protein length is missing the amino acid. The percentage observed is compared to the percentage calculated for 100 simulated proteomes (see Methods). Animal species in blue, fungal species in yellow. (D) Top: correlation between the number of predicted tyrosine kinases (per species) and percentage of tyrosine deserts (observed - expected). Bottom: correlation between the number of predicted tyrosine kinases as a fraction of the total number of kinases (per species) and percentage of tyrosine deserts (observed - expected). The $r^2$ coefficient is shown in both panels. (E) A schematic for testing Y counter-selection at the level of individual sites in multiple sequence alignments (MSAs). For each unique spurious pY, orthologs are extracted, the ortholog sequences are aligned, a phylogenetic tree is constructed, and a profile of amino acid preference is inferred on the basis of the MSA and phylogenetic tree. Animal species in blue, fungal species in yellow. (F) Difference between the inferred preference for Y in fungal species and metazoa species, calculated as their difference in equilibrium frequencies (π) by the Pelican software (Duchemin et al, 2023). The black point indicates the median (50th percentile) and the vertical lines connect the lower and upper quartiles (25th percentile and 75th percentile). The results are given for pY and non-pY tyrosines and separated according to their solvent accessibility (buried: RSA >0.2, intermediate: 0.2 < RSA < 0.4, exposed: RSA >0.4). Sample sizes are buried non-pY ($n = 1761$), buried pY ($n = 153$), intermediated non-pY ($n = 790$), intermediate pY ($n = 103$), exposed non-pY ($n = 596$), exposed pY ($n = 83$). Results were only given for sites with significant changes in amino acid profile (between animals and fungi) at an adjusted $p$ value <0.05. Distributions (pY and non-pY) were compared using a Kolmogorov–Smirnov two-tailed test. pY here refers to all sites identified in this study as being significantly upregulated (WT—dead) after excluding native pY sites in budding yeast. (G) Example of a protein (NOP7) with predicted counter-selection against Y (Y64) in animal species. Inferred Y preferences (0–1) are represented by $Y_{fungi}$ and $Y_{metazoa}$. A small sample of representative species are shown for the fungal and metazoan clades.

across all human Y kinases for which we have specificity models (Sugiyama et al, 2019). Applying this control again does not support the hypothesis of pY counter-selection (Fig. EV5B). Another possibility is that there is strong counter-selection against pY motif residues flanking the phosphoacceptor to prevent Y phosphorylation (Deng et al, 2014; Li et al, 2023; Sugiyama et al, 2019). However, we do not observe strong evidence for motif counter-selection (Fig. EV5C,D). Finally, we check the possibility that 'high quality' substrates—those that are phosphorylated by many independent kinases—will be subject to stronger counter-selection than sites targeted by a small number of Y kinases. Recurrently phosphorylated Y are more strongly counter-selected than rarely phosphorylated Y on average but the difference is not significant ($p = 0.084$, Kolmogorov–Smirnov, one-sided, Fig. EV5E).

Taken together, our findings do not support the hypothesis of proteome-wide pY counter-selection or that this can serve as a driver of proteomic Y content. However, we do not discount the possibility of pY counter-selection for restricted phylogenetic lineages or a smaller number of sites and proteins functionally related to signalling. For example, Y64 of the yeast protein Nop7, which is an essential protein that is required for the synthesis of some ribosomal subunits (Adams et al, 2002) (Fig. 6G). The functional consequences of Y absence/presence in such candidates requires further experimental research.

## Discussion

Here we have expressed human tyrosine kinases in *S. cerevisiae* and then measured the impact of kinase expression both on yeast fitness and the yeast phosphoproteome, each time comparing the WT kinase to its kinase-dead (KD) mutant. From our structural analysis, we predict around 20% of spurious pY sites to be destabilising for protein folding (Fig. 2F). Additionally, we perform proteome-wide variant effect prediction (VEP) on the basis of phosphoacceptor (Y) conservation and predict >1000 spurious pY sites overall to negatively impact protein function (Fig. 4F and EV4). Paradoxically, while we observe a strong negative correlation

between the number of spurious pY sites and fitness (Fig. 4D,E), it is also clear that many such sites predicted to be deleterious have a negligible effect on fitness. For example, expression of the kinase EPHB3 produces 274 predicted deleterious pY sites but only has a small fitness defect in one of the tested growth conditions (Fig. 4D; Appendix Fig. S16).

A potential explanation for the surprisingly weak fitness defects is that the pY sites may individually be of low stoichiometry, and that the number of observed sites is serving as a proxy for kinase activity. However, increasing kinase activity would be expected to increase both phosphorylation stoichiometry and the number of unique sites modified, so that identifying the direct cause of toxicity (total number of unique modifications or increased stoichiometry of a few important phosphosites) is non-trivial. In silico methods used for VEP assume 100% stoichiometry for DNA-encoded mutations, but phosphorylation of tyrosines occurs post-translationally and is reversed by phosphatases and protein degradation, implying incomplete substrate phosphorylation. Tyrosine phosphatases are present in yeast despite their lack of bona fide tyrosine kinases (Pincus et al, 2008; Hunter, 2009; Lim and Pawson, 2010; Chen et al, 2017); this includes established tyrosine phosphatases such as the PTP and Cdc25 families that can dephosphorylate functional pY sites on native S/T kinases (Hunter, 2009), though there are a number of other phosphatase families in yeast with reported dual-specificity for pY and pS/pT (Chen et al, 2017). In agreement, we found that a phosphatase inhibitor cocktail (containing the PTP inhibitor orthovanadate) was one of the most harmful growth conditions among the 41 treatments we tested (Appendix Fig. S15A). As additional evidence, we found low estimated stoichiometries among the strongly active kinases v-SRC and EPHB1 (Fig. 4G), which is also consistent with the generally low stoichiometry of pY phosphorylation in human tissues in the absence of specific stimulation (Sharma et al, 2014; Tsai et al, 2015, 2022). We further note that yeast lacks conventional SH2 and PTB domains for the binding of phosphotyrosine (Kaneko et al, 2012), which can shield the pY residue from phosphatase activity (Jadwin et al, 2018; Hunter, 2009). While budding yeast encodes one SH2 domain protein, the histone chaperone Spt6, the tandem SH2 domain of Spt6 can bind non-canonically to pS, pT, or pY

(Sdano et al, 2017; Brázda et al, 2020; Connell et al, 2022). Overall, while we predict the destabilisation of proteins and native protein–protein interfaces (Fig. 2D–F), it is not clear if these effects are contributing directly to toxicity. Finally, we consider that a haploinsufficiency screen in yeast, with a theoretical reduction in protein abundance of 50%, generated a measurable fitness defect for only ~3% of the yeast genome (Deutschbauer et al, 2005). In principle, many proteins can, therefore, individually experience a reduction in abundance without a strong impact on fitness.

We also examined the effect of spurious phosphorylation on the activity of native signalling pathways in yeast. This analysis indicated changes in S/T kinase activities (for example the cell cycle kinase Cdc28), and alterations in ploidy that are suggestive of cell cycle dysregulation (Appendix Figs. S10, S11). These findings are consistent with historical research demonstrating aneuploidies, spindle defects, increased Cdc28 activity, changes in cell morphology, and cell cycle arrest upon v-SRC expression in *S. cerevisiae* (Boschelli et al, 1993; Xu and Lindquist, 1993; Florio et al, 1994). From our Y and S/T phosphorylation data we also observe the inappropriate phosphorylation of native MAPKs and their downstream effectors. This raises one possibility that toxicity may be caused not just by completely new pY sites but also in part by the promiscuous phosphorylation of native pY sites above background levels. Notably, for 'activatory' phosphorylations on enzymes such as MAPKs, low stoichiometry Y phosphorylation may be sufficient for toxicity. PTPs in budding yeast are largely known for their negative regulation of endogenous pY sites on MAPKs (Pincus et al, 2008; Lim and Pawson, 2010), and this ancestral phosphatase activity may have enabled the emergence and gradual evolution of dedicated tyrosine kinases by offering some protection against spurious phosphorylation, as has been suggested (Hunter, 2009). This hypothesis is supported by the previous observation that v-SRC toxicity in budding yeast can be partly reversed by the co-expression of a mammalian PTP (human PTP1B) (Florio et al, 1994).

Our data also allows for a comparison between the spurious pY sites generated here and the native human pY proteome. Crucially, the former represents sites that have never been subject to evolutionary constraints while the latter have been exposed to natural selection. We observe that, compared to human pY sites, spurious sites are less accessible, more likely to map to ordered regions, and more likely to destabilise the protein fold (Fig. 5E,F, Appendix Fig. S23). This implies a structural shift towards more benign phosphorylation in systems with evolved phosphotyrosine signalling. This is especially important given the general tendency of phosphotyrosine to map to ordered and domain regions compared to pS/T (Fig. 2C) (Corwin et al, 2017; Ramasamy et al, 2022). However, it is not clear whether this represents the action of purifying selection against destabilising pY directly or the requirement for many pY sites to be accessible to SH2 domains, (PTB) domains, and tyrosine phosphatases for reversible modification. How such buried sites (~35% spurious pY) came to be phosphorylated in the first place is another important question. Buried phosphorylation has been documented in a small number of cases, indicating the transient exposure of the phosphoacceptor during protein dynamics (Li et al, 2008; Henriques and Lindorff-Larsen, 2020; Orioli et al, 2022; Swadling et al, 2022); a related explanation is that binding of the substrate in vivo to other molecules promotes a different conformational state to the one

predicted by AlphaFold2 as the default structure (Mayer, 2015; Del Alamo et al, 2022). Another possibility is that phosphorylation occurs on a partly unfolded subpopulation of each substrate. However, the exact mechanism for buried Y phosphorylation remains an open question.

Further examination of the pY sites generated here also reveals the recurrent spurious phosphorylation of targets across large swathes of the proteome. This implies that, in the context of off-target signalling, counter-selection against adventitious binding will not be restricted to native targets of the effector protein (Fig. 5). Such contexts may arise in nature due to: the sudden shift in protein expression levels (Faulkner et al, 2009; Young et al, 2015), the sudden shift in protein localisation (Nguyen Ba et al, 2014), the horizontal transfer of a PTM to a new host (Cummings et al, 2022), and/or the generation of a fusion protein leading to the aberrant specificity and activation of the effector (e.g. BCR-Abl kinase) (Smolnig et al, 2023). This result, in principle, is consistent with the proposal in 2009 that the animals and their unicellular relatives underwent a proteome-wide reduction in Y to avoid off-target phosphorylation by the emergent tyrosine kinases (Tan et al, 2009, 2011). However, we find no strong support for the hypothesis after testing it at the level of the proteome, proteins, motifs, and individual sites. Our work is therefore in closer alignment with other studies that have challenged this hypothesis (Su et al, 2011; Pandya et al, 2015; Kritzer et al, 2018). In particular, the 2018 finding that v-SRC toxicity can be suppressed by the overexpression of a single S/T kinase (SMK1) in spite of persistent tyrosine phosphorylation in the proteome (Kritzer et al, 2018). Proteome-wide selection against off-target Y phosphorylation, therefore, seems unlikely, although we do not exclude the possibility of Y counter-selection for a smaller subset of spurious targets or phylogenetic lineages. The correlated evolution of tyrosine kinases and PTPs is also likely relevant for understanding the selective constraints on the pY phosphoproteome in any given species (Chen et al, 2017).

In conclusion, even in spite of clear toxicity, this work suggests that many spurious interactions—represented here by phosphosites generated from non-native kinase-substrate interactions—have a minimal impact on fitness. Consistent with this, we do not observe any excess counter-selection of such spurious sites relative to sites not at risk of promiscuous binding. The findings indicate off-target protein interactions can be tolerated to a certain extent during evolution, possibly providing a means by which functional systems may evolve from non-functional ancestors. It is a remaining challenge to understand the conditions under which toxicity may arise and the fraction of spurious interactions that can give rise to fitness defects.

# Methods

## Strain construction

Out of a total of 31 kinases, coding sequences (CDS) of 17 kinases were obtained from the Human Kinase ORF Kit (Yang et al, 2011), three from the Corwin et al study (Corwin et al, 2017), v-SRC was kindly gifted by Dr. Josée Lavoie and the tyrosine kinase domains were gifted by Dr. Nicolas Bisson. Kinase-dead mutants for all kinases were generated by mutating the catalytic aspartate residue

(D167 in human PKA) in the kinase active site to asparagine (Kung and Jura, 2016; Reinhardt and Leonard, 2023).

Following a methodology described in Ryan et al (Ryan et al, 2016), yeast (AKD0643, *ura3 his3 met15 leu2::GEM gal4::NAT gal1::stuffer-linker-GFP-HIS3*) competent cells were co-transformed with a pCAS plasmid (Addgene plasmid 6084747) expressing both the gRNA for the stuffer (Dionne et al 2021) and *Streptococcus pyogenes* Cas9, and a donor DNA. For the construction of all the strains, a single gRNA was used that targeted a genomic location (*GAL1* replaced by a stuffer sequence) flanked by *Gal1p* and C-terminal GFP tag. The donor DNA with flanking homology arms was generated by amplifying the coding sequence of the kinase (CDS) using oligos containing homology with the target sequence (Oligo sequences in Dataset EV1). As a linker between the CDS and the C-terminal tag, a small flexible protein sequence (GGGGSGGGGS) that has minimal effect on the structure of the linked proteins was used. Transformed cells were selected on YPD plates containing 200 μg/mL G418 (plasmid selection, Bioshop Canada) and 100 μg/mL NAT (*gal4::NAT* deletion, Cedarlane Labs). To allow plasmid loss, randomly selected colonies were grown in YPD media without G418 for 2 days. Subsequently, the plasmid loss was confirmed by no growth on YPD plates containing 200 μg/mL G418. The genomic integrations of the CDSs were validated by PCR and their sequences were confirmed by Sanger sequencing (Oligo sequences in Dataset EV1).

To construct the mutants of the kinases, the same method was followed, except that as donor DNA, either PCR product from the amplification of the mutated CDS from a plasmid or overlapping PCR with the mutagenesis oligos, was used. All oligo sequences are included in Dataset EV1.

## Cell pellet preparation

Pre-cultures of yeast cells were prepared by overnight growth at 30 °C with agitation, in synthetic complete (SC) medium (1.74 g/L Yeast nitrogen base without amino acid without ammonium sulfate, 1 g/L Monosodium Glutamate salt, 20 g/L Glucose, 1.34 g/L complete dropout, see Table EV3 for the dropout mixture). Pre-cultures were inoculated at an O.D. of 0.05 in a fresh SC media and grown for 12 h at 30 °C with agitation. Then, the cells were pelleted. The pellet was washed with PBS, and frozen at −80 °C. In the 'induced' condition, the fresh medium contained 1 μM of beta-estradiol inducer, whereas in the 'not induced' condition, the fresh media contained the same volume of DMSO as the inducer.

## Activity assay by Western blotting

The cell pellets for the activity assays were prepared as described in the Cell pellet preparation section above. For cell lysis, the pellets were resuspended in Laemmli buffer (16.25 mM Tris-HCl pH 6.8, 10% glycerol, 2% SDS, 0.01% bromophenol blue and 5% β-mercaptoethanol) and heated at 95 °C for 10 mins. The protein extracts were then spun down and the supernatants were migrated on 10% polyacrylamide gels (volumes were normalised based on the final OD of the different yeast strains). Once separated, the proteins were transferred to nitrocellulose membranes for Western blotting and correct protein loading was validated with ponceau staining. All membranes were blocked in a 5% milk solution and then incubated for 12 h at 4 °C with either the mouse anti-

phosphotyrosine 4G10 (05321, Millipore) or the rabbit anti-GFP (Thermo A11122) primary antibodies. Signals following a 1 h incubation at room temperature with the appropriate HRP-conjugated secondary antibodies (anti-mouse NEB 7076 S or anti-rabbit NEB 7074S) were acquired using the Clarity Western ECL Substrate (Bio-Rad) and an Amersham Imager 600RGB (GE Healthcare).

## Protein abundance assay by Western blotting

For this validation, 20 proteins containing pY sites at intermediate-high stoichiometries and predicted to affect protein stability were selected and tagged with a 3X-FLAG epitope. Each protein was tagged in four different backgrounds (v-SRC, v-SRC kinase-dead, EPHB1 and EPHB1 kinase-dead) and tagging was validated by Western blots. To check the effect of phosphorylation on the protein of interest, we grew each strain in triplicates with 100 nM beta-estradiol for 6 h at 30 °C (from 0.15 OD/ml to–1 OD/ml). Following the growth, the cells were spun down and washed with sterile water. The pellet was then frozen for further use. The pellet was resuspended at 0.05 U OD/μL in lysis buffer (Complete Mini, Roche and PhosStop (ᴅ-mannitol, sodium molybdate dihydrate, sodium orthovanadate, cantharidine), Roche) and 250 μl was processed for cell lysis. Cell lysis was performed by adding glass beads to the cell suspension and then vortexed on a Turbomix for 5 min. Then, 25 μL of 10% SDS was added to the mixture and boiled for 10 min. Finally, the tube was spun at 16,000×*g* for 5 min. Samples for migration were prepared by mixing 17.5 μL of clear supernatant, 2.5 μL of 1M DTT and 5 μL of 5X loading buffer (250 mM Tris-Cl pH 6.8, 10% SDS, 0.5% bromophenol blue, 50% glycerol). The percentage of acrylamide gel needed was determined based on theoretical protein size, as small proteins will have better separation on 15% gel than large proteins on 8% gel. Following protein separation on SDS-PAGE, they were transferred onto a nitrocellulose membrane at 0.8 mA/cm$^2$ for 1h15min. Each membrane was stained with Ponceau to confirm proper loading and appropriate transfer. The membrane was blocked overnight in a blocking buffer (Intercept® (PBS) Blocking Buffer, 927-70003, Mandel Scientific). The next morning, an Anti-FLAG M2 antibody (Anti-Flag M2, F3165-1MG, Millipore-Sigma) was applied to the membrane for 30 min to the protein of interest and with Anti-GFP (Millipore-Sigma, 11814460001) for kinase expression. Secondary antibodies (Anti-Mouse 800, LIC-926-32210, Mandel Scientific) were applied for 30 min and then the membrane was imaged with an Odyssey Fc instrument (Licor, Mandel Scientific) in 700 and 800 channels.

## DHFR PCA assays

To test for the effect of spurious phospho-tyrosine on protein–protein interaction (PPI) we crossed our kinase WT and dead with a strain containing both DHFR fragments. We selected the Rvs167/Sla1 interaction, since Rvs167 Y476 is phosphorylated and contained inside an SH3 domain. The resulting diploid strains were grown overnight in SC complete (MSG) pH 6.0 at 30 °C. The next morning, cultures were diluted at 1 OD/ml in water before being diluted to 0.1 OD/ml in the different conditions. Each strain was grown in four different media: PCA media + DMSO, PCA media + DMSO + 100 nM estradiol, PCA media + MTX, and PCA media + MTX + 100 nM estradiol (PCA

media: 0.67% Yeast nitrogen base without ammonium sulfate without amino acids, DO -adenine, 2% glucose, DMSO is at 2% and MTX is at 200 μg/ml). Growth in each condition was recorded every 15 min for 3 days at 30 °C in an Epoch 2 plate reader (Agilent).

To make sure the effect we saw on the PPI was from the phospho-tyrosine, we decided to mutate the Rvs167 SH3 domain and check for the effect on the Rvs167/Sla1 interaction. The Rvs167 SH3 domain mutations (Y53E and Y53F using the SH3 numbering) were done using fusion PCR. WT and mutated SH3 domain were inserted using the same CRISPR/Cas9 approach into a landing pad expressing the domain of interest fused with the DHFR[1,2] fragment. Those strains were then mated with the Sla1-DHFR[3]. The resulting diploids were grown overnight in SC complete (MSG) pH 6.0 at 30 °C. The next morning, we followed the same procedure as before, but we had only three conditions: PCA media + MTX, PCA media + MTX + 15 nM estradiol and PCA media + MTX + 30 nM estradiol. Growth in each condition was recorded every 15 min for 3 days at 30 °C in a Spark instrument (Tecan).

## Examination of ploidy levels

Kinase (v-SRC and EPHB1) WT and dead strains, empty landing pad strain, BY4741 (*his3 leu2 ura3 met15 Mat a*) and BY4743 (*his3/his3 leu2/leu2 ura3/ura3 MET15/met15 lys2/LYS2* diploid) were grown overnight in SC complete (MSG) pH 6.0 in triplicates at 30 °C. The next morning, strains were diluted four times at 0.15 OD/ml in 3 mL fresh media and grown for 5 h at 30 °C. Then, 100 nM estradiol was added to two out of four cultures for each strain. Following estradiol addition, one culture with and without estradiol were incubated for either 3 h or 6 h at 30 °C. After the 3 or 6 h incubation with estradiol, cells were spun down at 630 × *g* and resuspended in 1 mL of 70% ethanol. To make sure cells were fixed properly, the cell suspension was stored at 4 °C overnight. The next morning, cells were spun down at 630 × *g*, and ethanol was removed. Cells were resuspended with 1 mL of sodium citrate, 50 mM, pH 7 and spun down at 630 × *g* to wash the remaining ethanol. The supernatant was removed and the pellets were resuspended in 1 mL sodium citrate, 50 mM, pH 7 with 25 μL RNAse A 10 mg/ml. RNAse A treatment was incubated overnight at 37 °C. Cells were then washed twice with 1 mL sodium citrate, 50 mM, pH 7. Pellets were resuspended in 1 mL sodium citrate, 50 mM, pH 7. For the DNA staining step, 100 μL of the cell suspension was transferred in a 96-well plate (VWR, 82050-764) containing 150 μL of sodium citrate, 50 mM, pH 7 with 1 μM Sytox Green (Thermo Fisher, S7020). The 96-well plate was incubated overnight at room temperature in the dark. Finally, 5000 events were recorded for each condition with a flow cytometer (Guava EasyCyte instrument, Cytek Bio).

## Flow cytometry competition assay

For this experiment, kinase and kinase-dead mutants were competed against a WT parental strain expressing mCherry (*PDC1-mCherry*). The two competing strains were mixed at a ratio of 1:1 in 200 μL of SC complete media. Each competition was performed with and without beta-estradiol. The competition was performed over 3 days (~30 generations). Each day, for the next round of the competition, cultures were diluted 10 μL in 190 μL of sterile fresh media. Samples were analysed using a flow cytometer

(Guava EasyCyte instrument, Cytek Bio) for 5000 events each. The bimodal distributions of the fluorescence intensities were fitted to two-component Gaussian mixture models, to obtain thresholds to classify the events with signal and those without signal. The threshold for each sample was calculated as the intensity level at the intersection of the fitted curves, lying between the distributions of the events with and without signal. Because the cells are expected to have either GFP or mCherry fluorescence, but not both, the events classified as not having signal in channels corresponding to GFP and mCherry fluorescence and the ones with signal in both the channels (doublets) were discarded. Among the remaining events, the relative frequency of a given strain harbouring a kinase in a sample was calculated as the proportion of events without mCherry signal.

## Fitness measurements across conditions

In total, 77 strains (31 wt kinases, 31 kinase-dead mutants, 13 mutants of v-SRC, 1 strain without a kinase coding sequence (empty landing pad) and the background strain BY4741) were arrayed onto four 384 OmniTray plates using a robotically manipulated pin tool (S&P robotics). After growth for 2 days at 30 °C, the plates were then used for condensation onto 1,536 OmniTray plates containing SC complete media (1.74 g/L Yeast nitrogen base without amino acid without ammonium sulfate, 1 g/L monosodium glutamate salt, 20 g/L Glucose, 1.34 g/L complete drop, 20 g/L agar), using the robotically manipulated pin tool. On these plates, the positions of the strains and their 16 replicates were randomised in order to account for any location-specific undue growth effects. Also, the background strain BY4741 was positioned at the border of the plates (two rows and two columns) in order to reduce any border effects. These plates were incubated for 2 days at 30 °C. Next, these plates were used as source plates to replicate the 41 different growth conditions with and without beta-estradiol (0 and 1000 nM in DMSO), referred to as destination plates. The destination plates were incubated at 37 °C in a SpImager custom platform. The images of the plates were acquired every 90 min for 3 days. The conditions selected are known to trigger stress responses and were used previously in (Dionne et al, 2021).

## Fitness screen: analysis

Images obtained from the SpImager custom platform were used to obtain growth parameters using python-based pyphe software (Kamrad et al, 2020). To generate the input for the pyphe software, the python based scikit-image tool (van der Walt et al, 2014) was used. Using scikit-image, images were first preprocessed to retain only the area containing the colonies in the images. This was done by removing the borders of the plates from the images. The images were then converted to grayscale and their intensity range was inverted. The resulting images were provided as input to the pyphe-quantify tool using these parameters: `batch --grid auto_1536 --s 0.1`. The measurement of the circularity of the colonies was used to filter the colonies with abnormal shapes. Next, the `pyphe-quantify` tool was used with the following parameters: `timecourse --grid auto_1536 --s 0.1`, to obtain the growth curves. For each growth curve, the colony size at the initial time point was subtracted from the colony sizes at every time point. This correction helped in reducing the effect of possible

unequal replication of colonies by the pin tool. Next, the corrected growth curves were provided as input to the pyphe-growth curves with default parameters, to obtain the growth parameters. Additionally, the growth curves were used to measure the Area Under the Curve (AUC) according to composite Simpson's rule using python based scipy package's (Jones et al, 2016) integrate.simps module, with default parameters.

Growth conditions in which $CuSO_4$ was added to the media produced red coloration of the colonies. These plates were preprocessed to include intensity from the red channel only. The following steps of the analysis were the same as other plates.

The values of the growth parameters were rescaled by those of the reference strain without a kinase CDS. This within-plate rescaling allowed for carrying out comparisons between different plates. The fitness score for a wild-type kinase was calculated as a difference between its rescaled growth parameters and that of the corresponding kinase-dead mutant. The statistical significance of the difference was tested using Mann–Whitney $U$-test corrected for multiple tests.

## (Phospho-)proteomics: sample processing and measurement

Cell pellets were thawed and mixed with 750 µl of freshly prepared urea lysis buffer (8 M urea, 75 mM NaCl, 100 mM Tris pH 8.5) and tubes were filled up with 0.5 mm zirconia/silica beads. Cells were lysed via four cycles of bead-beating (1 min each with 30 s cooling on ice in between). After removing any water sticking to the tube, a hole was poked using a needle, and the lysate was transferred to a new tube by centrifugation at $420 \times g$ for 1 min. Cell debris was discarded and lysate transferred to a new tube after 10 min of 21k×$g$ centrifugation at 4 °C. Lysates were sonicated for 20 s each and BCA assay was performed to determine protein concentration. Lysates were reduced with 5 mM DTT (30 min, 55 °C), alkylated with 20 mM CAA (30 min, room temperature in the dark) and quenched with another 5 mM DTT (15 min, room temperature).

For (phospho-)proteome preparation, 250 µg of protein lysate were processed using an adjusted R2-P2 protocol (Leutert et al, 2019). In brief, 250 µg protein was diluted to 1 µg/ul, with 2 µg beads per 1 µg protein added. 750 µl of ACN were added to precipitate proteins. Washing was performed with 1 ml each of 100% ACN, 100% MeOH, 100% ACN, 80% ACN and 80% EtOH. Digestion was performed using 2.5 µg trypsin diluted in 200 µl of 100 mM Tris pH 8.5 for 12 h at 37 °C. For proteome measurement, digests were acidified by adding 20 µl 50% FA to 5% total. The supernatant was separated from bead remnants on a magnetic rack, and centrifuged for 10 min at 21k×$g$. The supernatant was filtered through a C8 StageTip (50 µl MeOH, 50 µl ACN, sample, 50 µl 80% ACN in 0.1% FA), dried in a speed-vac and stored at −20 °C until MS measurement. For phospho-proteome measurement, digests were acidified by adding 20 µl 50% TFA to 5% total and 800 µl ACN and centrifuged for 10 min at 21k×$g$. The supernatant was used in R2-P2 phosphopeptide enrichment. Enrichment was performed using 62.5 µl of a 5% solution $Fe^{3+}$-NTA magnetic beads (PureCube Fe-NTA MagBeads, Cube Biotech), washed 3x with 80% ACN in 0.1% TFA and eluted into 100 µl of 2.5% $NH_4OH$ in 50% ACN. Phosphopeptides were acidified with 10 µl of 50% FA to 10% final, filtered through a C8 StageTip (50 µl MeOH, 50 µl

ACN, sample, 50 µl 80% ACN in 0.1% FA), dried in a speed-vac and stored at −20 °C until MS measurement.

For MS measurement, peptides were re-solubilized using 4% ACN in 0.1% FA. 2.5 µl of the sample was analysed by nLC-MS/MS using an Orbitrap Exploris 480 Mass Spectrometer (Thermo Fisher) equipped with an Easy1200 nanoLC system (Thermo Fisher). Peptides were loaded onto a 100 µm ID × 3 cm precolumn packed with Reprosil C18 3 µm beads (Dr. Maisch GmbH), and separated by reverse-phase chromatography on a 100 µm ID × 35 cm analytical column packed with Reprosil C18 1.9 µm beads (Dr. Maisch GmbH), housed in a column heater set at 50 °C, using two buffers: (A) 0.1% FA in water, and (B) 80% ACN in 0.1% FA in water at 450 nl/min flow rate. For MS measurement of the 390 unique phospho-proteomics samples, kinase conditions were measured in four blocks corresponding to Ser/Thr kinases (pST), full-length tyrosine kinases (pY), tyrosine kinase domains (pYd) and v-SRC mutants (v-SRC). Each block consisted of five replicate blocks, with kinase condition order randomised in each replicate. Wild-type and kinase-dead mutants of the same kinase were always measured back-to-back, the order switching with each replicate. A wash was run after every sixth sample to prevent LC column blockage, with extensive additional washes between blocks. The 20 proteome samples were measured in one batch with the same blocking and randomisation as described above.

For LCMS measurement, peptides were separated over a 60 or 90 min LC gradient ramping from 6% B to 32% in 43 or 69 min, respectively, and washing at 95% before re-equilibrating to 3%. Data-independent acquisition (DIA) was performed as a staggered window approach (Pino et al, 2020) (Table EV4). A full MS1 scan was recorded after every DIA cycle at 60k resolution with standard AGC target and automatic maximum IT. Two distinct DIA cycles that were shifted 12 Th covered an effective range of 438 to 1170 m/z (phospho) and 363 to 1095 m/z (proteome). One DIA cycle consisted of 30 windows of 24 Th size at 30k resolution, 27% NCE HCD, charge state 3 and AGC target 1000%. Data-dependent acquisition (DDA) analysis was performed using a 3 s cycle approach. A full MS1 scan was recorded every 3 s at 120k resolution with a 300% AGC target, automatic maximum IT and 450 to 1150 m/z. MS2 precursors were selected using filters MIPS peptide, 5e3 intensity threshold, charge states 2–6 and 30 s dynamic exclusion. MS2 scans were recorded at 30k resolution, 27% NCE HCD, 1.6 Th isolation window, charge state 3 and standard AGC target. For both DDA and DIA acquisition, gas phase fractionated (GPF) measurements were performed on pooled samples. For DDA-GPF, five measurements covering 4x 125 m/z and 1x 200 m/z of the total scan range were performed. For DIA-GPF, seven measurements covering 100 m/z each with 25 windows of 4 m/z isolation width each were conducted.

## (Phospho-)proteomics: data processing and analysis

During phospho-proteomics MS measurement, replicate five of the v-SRC wild-type in the v-SRC mutants sample block repeatedly caused LC column blockages and could not successfully be measured. During phospho-proteomics data analysis, replicates 5, 2 and 2 of MERTK wild-type, vSRC mutant L308A and V292 A, showed comparably low mean intra-condition Pearson correlations of 0.69, 0.47 and 0.47, respectively. For comparison, the minimum

and mean observed Pearson correlations of all other conditions were 0.82 and 0.94, respectively. Thus for the phospho-proteomics analysis, four out of 78 conditions only have four instead of five replicates available, yielding a total of 386 unique samples with 389 available MS raw files. For proteome analysis, four replicates were measured per condition.

MS raw files were converted to mzML files using MSConvert v3.0.22248-ce619fc (Chambers et al, 2012). For staggered DIA files, filters peakPicking at vendor msLevel 1- and demultiplex with optimisation overlap_only and massError 10 ppm were used. Database and spectral library search was performed using FragPipe v18 (Yu et al, 2023) with a FASTA file combining yeast and human kinases (wild-type and kinase-dead) downloaded from Uniprot on 2021-11-08. DDA, DDA-GPF and DIA-GPF files were used to create a spectral library in FragPipe. Default settings for strict trypsin and high-resolution MS data were combined with variable modifications 15.9949 on M, 42.0106 on protein N-terminus, 79.96633 on STY, −17.0265 on peptide N-terminal QC and −18.0106 on peptide N-terminal E. DIA files were searched using the created spectral library in FragPipe and quantified with DIA-NN (Demichev et al, 2019) in mode 'Any LC (high accuracy)'. Phospho PTM localisation was performed by adding the parameter "--var-mod UniMod:21,79.966331,STY --monitor-mod UniMod:21".

LCMS data analysis was performed using custom scripts in RStudio v2023.03.0 and R v4.2.3, with packages including data.table, ggplot2, Biostrings and cowplot. All scripts are available at https://gitlab.com/public_villenlab/spurious-phosphorylation. DIA data were filtered at ≤1% FDR level with Q.Value and Global.Q.Value. Phosphopeptide data was further filtered at PTM.Site.Confidence ≥ 75%. For stoichiometry calculation, the phosphopeptide data was additionally filtered at ≤1% FDR level with PTM.Q.Value for extra stringency. Dimensionality reduction via tSNE was performed using the R package Rtsne (Maaten and Hinton, 2008; van der Maaten, 2014) with perplexity 10, theta 0.0 and max iterations 5000. Statistically significant regulation was tested using the R packages samr (Tusher et al, 2001) and limma (Ritchie et al, 2015). For SAM, default settings were used. For LIMMA, contrast matrices were created between kinase wild type and kinase-dead conditions, as well as v-SRC mutants and v-SRC kinase-dead conditions. For each test, only peptide precursors that were quantified in at least 75% of replicates in at least one condition were used. Of the remaining precursors, if all replicates of one condition were not quantified, intensities for this condition were imputed. Imputation was performed the same way as the Perseus computational platform (Tyanova et al, 2016), using a normal distribution of both conditions in the test (sd = 0.3 x sd_of_data; mean = mean_of_data - 1.8 x sd_of_data). LIMMA testing was performed using functions lmFit, constrasts.fit and robust eBayes. *P* values were multiple testing corrected using Benjamini–Hochberg. Motif analysis was performed using the R package dagLogo (Ou et al, 2020). Enriched amino acids were filtered by *p* value ≤0.05 and visualised using the R package ggseqlogo (Wagih, 2017).

## (Phospho-)proteomics: stoichiometry

Phosphorylation site stoichiometry values were calculated by correlating intensities of phosphorylated peptide precursors and their non-phosphorylated counterparts (Presler et al, 2017; Hogrebe et al, 2018). To adjust for changes in protein abundance

between wild-type and kinase-dead conditions, proteome maxLFQ intensities were calculated and used to normalise peptide precursor intensities (Cox et al, 2014). Crucially, only non-phosphorylated peptide precursors whose phosphorylated counterparts did not exhibit significant regulation as determined by LIMMA were included in the maxLFQ calculations. For LIMMA testing only, intensities were imputed if peptide precursors could only be quantified in max 25% of one condition, and min 75% in the other. maxLFQ-normalised peptide precursors were filtered: (1) to retain only the highest abundant charge state per modified peptide; (2) to have phosphorylated peptide precursors fully quantified in at least one condition; (3) to have all non-phosphorylated peptide precursors fully quantified in all conditions; (4) to retain only peptide precursors with exactly 1 phosphorylated counterpart to the non-phosphorylated peptide precursors (called a 'stoichiometry group'). Finally, the remaining phosphorylated peptide precursors were imputed to be 0 whenever not quantified. After stoichiometry calculation, stoichiometry groups yielding a positive slope were removed, as they could yield stoichiometry values below 0% or above 100% (called 'illegal').

## Structural bioinformatics

Structural models for kinase substrates were sourced from AlphaFold2 where possible (Jumper et al, 2021; Varadi et al, 2022). The solvent-accessible surface area (SASA) and secondary structure of tyrosine residues was calculated with DSSP v. 3.1.4 (Kabsch and Sander, 1983) using the dssp() function in the R package bio3d (Grant et al, 2021). The SASA was normalised by the maximum Y accessibility (255 Å$^2$) to give the relative solvent accessibility (RSA) (Tien et al, 2013). The Y disorder state was predicted using an AF2-based approach (Akdel et al, 2022; Piovesan et al, 2022), where a smoothened RSA in a ±12 amino acid window surrounding the central Y was used as a proxy for protein disorder prediction (Fig. 2A). Tyrosines with a smoothened RSA greater than or equal to 0.581 were predicted as disordered (Piovesan et al, 2022). The effect of phosphorylation on the Gibbs free energy of folding (ΔΔG) was predicted using FoldX (Schymkowitz et al, 2005; Studer et al, 2016). All protein structures were first repaired using the FoldX RepairPDB command (--ionStrength=0.05, --pH=7, --water=CRYSTAL --vdwDesign=2). The ΔΔG of phosphorylation (Y to pY) was then calculated using the BuildModel command, with 5 runs per phosphosite. Finally, the effect of pY phosphorylation on protein interactions was predicted by calculating the ΔΔG for complex formation using the PSSM command, with five runs per phosphosite. This command predicts destabilisation of the protein interaction independently of whether the variant contributes to the intra-molecular destabilisation of one of the interactors. Destabilising pYs were called using a standard ΔΔG threshold of 2 kcal/mol (Nishi et al, 2011; Wagih et al, 2018; Høie et al, 2022). Structural annotations of protein–protein interfaces for *S. cerevisiae* were derived from the 'Highest confidence interfaces' dataset of Interactome INSIDER (Meyer et al, 2018). This resource includes annotations of PDB structures and homology models present in Interactome3D (Mosca et al, 2013).

We supplement this data with a recent AF2-based screen of protein interactions in *S. cerevisiae* (Humphreys et al, 2021). Interface residues were determined using the approach outlined in (Meyer et al, 2018) i.e. residues with an RSA >0.15 and with a change in SASA >1.0 Å$^2$ upon complex formation. Interface residues were filtered further to have an AF2 predicted aligned

error (PAE) less than 8 Å. Finally, we include Interactome INSIDER's machine learning-based predictions of interface residues for our analysis (Fig. 2E; Appendix Fig. S7E)

Calculations of spurious-to-native and native-to-native 1D and 3D distances were made with respect to the ultra-deep reference yeast phosphoproteome presented in (Leutert et al, 2023). Sites are considered 1D proximal if they are within four amino acids of each other. Sites are considered 3D proximal if they are within 8 Å of each other (Humphreys et al, 2021). pS/pT and pY sites mapping to low-confidence positions (pLDDT <70) were excluded from this analysis. Phosphosites were added to the AF2 structures using the PyTMs package in PyMOL (Warnecke et al, 2014).

Phosphosites matching to putative short linear motifs (SLiMs) were extracted using the *gget elm* resource, based on regular expression matches of *S. cerevisiae* SLiMs to the budding yeast proteome (Kumar et al, 2022; Luebbert et al, 2024). Matches were restricted to solvent-exposed sites outside of annotated pfam domains to increase the confidence of the predictions.

## Conservation-based variant effect prediction (VEP)

Homologues of all unique spurious substrates were retrieved using hhblits v. 3.3.0 (Remmert et al, 2011). Each sequence was searched with one iteration ($n = 1$) against the UniRef30 database (2022 version) that contains non-redundant UniProtKB sequences clustered at 30% sequence identity (Suzek et al, 2015; Mirdita et al, 2022). Homologues were retrieved with an e-value threshold of $e^{-10}$ and the resulting alignment was filtered to retain only positions found in the query sequence (i.e. the *S. cerevisiae* copy). Hits with 50% or more gapped positions were then removed (Høie et al, 2022). Variant effect prediction (VEP) based upon the sequence alignment was performed with GEMME software for all alignments containing at least 20 hit sequences (Laine et al, 2019). Results were taken from the combined evolutionary model ('evolCombi') and then rank-normalised (within proteins) such that the most harmful mutations were assigned a score of 1 and the least harmful a score of 0.

GEMME and similar softwares perform VEP for amino acid mutations but not for phosphorylation. We, therefore, weight GEMME scores for the 19 amino acids by their biophysical similarity to pY. Estimates of AA-to-pY biophysical similarity were obtained by mutating a large sample of spurious pY ($n = \sim2300$) in Silico from Y to the 19 other amino acids and pS/pT/pY using the PSSM command in FoldX (Schymkowitz et al, 2005). Energy terms were taken for the backbone Hbond, VdW, electrostatics, polar solvation, hydrophobic solvation, VdW clashes, sidechain entropy, mainchain entropy, torsional clashes, backbone clashes, helix dipole and ionisation energy. Correlation matrices were generated for each energy term, and then the average correlation of each amino acid to pY was used to calculate a pY-weighted mean of the GEMME scores. Annotations for essential proteins were retrieved from YeastMine (Balakrishnan et al, 2012).

## Evolutionary bioinformatics (human–yeast conservation)

Orthologs of the kinase substrates were retrieved from the vertebrate, fungi, and metazoa databases of Ensembl Compara using the Ensembl REST API (Vilella et al, 2009; Yates et al, 2015). Tyrosine-phosphorylated proteins in humans were annotated using

PhosphoSitePlus (Hornbeck et al, 2015), while requiring at least five supporting sources to ensure high-confidence annotations (Ochoa et al, 2020; Kalyuzhnyy et al, 2022). Assessing the conservation of kinase-substrate relationships (Appendix Fig. S23C) was performed using the ProtMapper resource for human substrate annotations (preprint: Bachman et al, 2022), and the substrates identified here (via mass spectrometry) for the yeast substrate annotations. Protein domain annotations were assigned using pfam (Mistry et al, 2021), and protein abundance data was retrieved from PaxDB (Wang et al, 2015).

To generate ortholog alignments, all unique orthologs were first filtered to remove redundant orthologs at a sequence identity threshold of 95% using CD-HIT (Fu et al, 2012). Ortholog sequences were aligned to the *S. cerevisiae* copy using the 'phmmer' command from HMMER 3.3 software and then the HMM-sequence alignments were concatenated to give a single aligned sequence for every ortholog. The resulting alignments between the *S. cerevisiae* copy and its orthologs were used to determine site-level conservation between *S. cerevisiae* pY and human pY (Fig. 5D). For further evolutionary analysis (described in the section below), candidate false positives were removed by excluding any predicted ortholog that aligned to less than 50% of the sequence of the *S. cerevisiae* copy. The trimAl software was then used to remove positions from the alignment with more than 20% gaps across all sequences (Capella-Gutiérrez et al, 2009), and an additional filter was applied to remove spurious homologues in the MSA also using trimAl (parameters: -resoverlap 0.75, -seqoverlap 80) (Capella-Gutiérrez et al, 2009).

The structural analysis of pY accessibility (RSA), order/disorder, and effect on stability ($\Delta\Delta G > 2$) was undertaken using the same approach outlined above ('Structural bioinformatics' section). Human proteins were considered to be pY-phosphorylated if supported by at least five sources from PhosphoSitePlus (Hornbeck et al, 2015). For native pY sites, we use a high-confidence group found to be endogenously phosphorylated in both (Lanz et al, 2021) and (Leutert et al, 2023). For the spurious pY group, we conservatively exclude any pY site found to be endogenously phosphorylated either in (Lanz et al, 2021) or (Leutert et al, 2023).

## Evolutionary bioinformatics (testing for pY counter-selection)

Non-redundant reference proteomes were taken from UniProt for each of the species analysed (UniProt Consortium, 2023). Kinases were predicted and classified (e.g. as tyrosine kinases) using the Kinannote software (Goldberg et al, 2013).

The species tree in Appendix Fig. S24 was constructed using orthologs from 19 metazoan species and 6 fungal species selected to give broad taxonomic coverage, with each phylum represented by at least one species if a reference proteome was available. Orthologous groups were used if they contained at least one ortholog from each of the 25 species. Orthologous sequences were aligned using MAFFT L-INS-i (Katoh and Standley, 2013), and alignment positions with more than 20% gaps were removed using trimAl (Capella-Gutiérrez et al, 2009). Orthologs aligning to less than 50% of the *S. cerevisiae* sequence (i.e. the query sequence) were removed. For species containing more than one ortholog in an alignment (i.e. one-to-many or many-to-many copies), the copy with the highest sequence identity to the *S. cerevisiae* copy (i.e. the query sequence)

was retained and the others removed. The resulting set of alignments was then used to construct a species tree with IQ-TREE2 using the 'concatenation/supermatrix' approach (Chernomor et al, 2016; Minh et al, 2020). Optimal substitution models for each alignment were determined automatically within IQ-TREE2 using ModelFinder (Kalyaanamoorthy et al, 2017), and branch confidence was determined with 1000 ultrafast bootstrap replicates (Hoang et al, 2018).

For the correlation of proteome Y content against the number of Y kinases, the phylogenetic non-independence of data points was corrected using phylogenetic independent contrasts (Fig. 6B; Appendix Fig. S24C–E) (Felsenstein, 1985), as it is implemented in the R package ape (Paradis and Schliep, 2018). Data for *Hydra vulgaris* was excluded from this analysis as Ensembl Compara lacked the *H. vulgaris* orthology relationships required to generate the species tree that this method uses as input. In Fig. 6B, proteome-wide disorder and accessibility predictions were performed for all species using the approach outlined above (section: structural bioinformatics) with scripts from: https://github.com/BioComputingUP/AlphaFold-disorder (Piovesan et al, 2022). Sites with an RSA <0.2 were considered to be buried and sites with an RSA >0.4 were considered to be exposed. *Acanthaster planci* was excluded from this analysis as there were too few AF2 structures for a reliable estimate of Y content in buried and exposed regions.

For the analysis of PTM deserts (Fig. 6C,D), proteome-wide disorder and RSA prediction was performed as described directly above (Piovesan et al, 2022). Each proteome was then simulated 100 times while accounting for the order:disorder content, the amino acid composition of ordered and disordered regions, and the length of each protein in the proteome. A protein was considered to have a Y desert if a contiguous sequence with length 50% or more of the whole protein was missing tyrosine (Szulc et al, 2023). Short proteins (length <150 amino acids) were excluded from the analysis.

The 'avoided word' (Appendix Fig. S25) analysis was performed with AW software (https://github.com/solonas13/aw) that aims to detect DNA or amino acid strings that appear at a significantly lower (or higher) frequency than the chance expectation (Almirantis et al, 2019). The AW algorithm was executed to find avoided words (-w 0) of variable length at a score threshold of −3 (-t -3) for the three protein sets described in Appendix Fig. S25.

Testing for pY counter-selection at the site-specific level was performed using Pelican software (Duchemin et al, 2023). This software takes as an input a multiple sequence alignment (MSA) and a phylogenetic tree with trait values annotated to each tip and ancestral node. Processed MSAs across all orthologs were taken from the phmmer-derived alignments described in the section above (Evolutionary bioinformatics: human–yeast conservation). Phylogenies for each MSA were generated with FastTree using default parameters (Price et al, 2010). We required that the fungal and metazoan sequences were successfully resolved in the phylogenetic tree by excluding trees where the fungal clade contained >2.5% metazoan sequences or vice versa. The annotation of binary ancestral traits ('fungi' or 'metazoa') was performed using the parsimony method within Pelican (Duchemin et al, 2023). Pelican was executed with the 'pelican scan discrete' command using an amino acid alphabet. Pelican *p* values ('aagtr') were retrieved for all Y amino acids (pY and non-pY) in the *S. cerevisiae* copies and then the Benjamini–Hochberg method was applied across all orthologous groups to correct for multiple-hypothesis

testing (Benjamini and Hochberg, 1995). For the analysis of these results we exclude as potential confounders any native pY detected in (Lanz et al, 2021) or (Leutert et al, 2023), and any spurious pY aligning to a native human pY (Fig. 5D).

## Data availability

The mass spectrometry proteomics data (Dataset EV14) have been deposited to the ProteomeXchange Consortium via the PRIDE (Perez-Riverol et al, 2022) partner repository with the dataset identifier PXD045466 (https://www.ebi.ac.uk/pride/archive/projects/PXD045466). All R code relating to mass spectrometry data analysis is publicly available under https://gitlab.com/public_villenlab/spurious-phosphorylation. All R code relating to the analysis of protein structure, fitness, and evolution is publicly available under: https://github.com/Landrylab/Bradley_2023_spurious_pY.

The source data of this paper are collected in the following database record: biostudies:S-SCDT-10_1038-S44318-024-00200-7.

## Peer review information

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

## Acknowledgements

We acknowledge the three anonymous reviewers for their helpful comments. We would like to thank members of the Villen and Landry labs for providing feedback on the science. We would like to thank Ricard A. Rodriguez-Mias in the Villen lab for assistance with mass spectrometry. We would like to thank Angel F. Cisneros for assistance with the FoldX analysis. We would also like to thank Louis Duchemin, Philippe Verber, and Bastien Boussau for help using the Pelican software, and Solon P. Pissis for help using the Avoided Word

(AW) software. We would like to thank the Sanger Sequencing platform CHUL for the sequencing. Finally, we would like to thank Rahul Nikam and Michael Gromiha for their assistance with the ProThermDB data. This work was primarily funded by the Human Frontier Science Programme research grant RGP34/2018 (to JV and CRL). CRL was also funded by the Canadian Institutes of Health Research Foundation grant number 387697. JV was also supported by NIH grants R35GM119536, R35GM152061 and R01AG056359. DB was funded by the EMBO Long-Term Fellowship (LTF) (ALTF 1069-2019). AH was supported by the DFF International Postdoctoral Grant (0131-00031B) and EMBO non-stipendiary Postdoctoral Fellowship (ALTF481–2020). RD was funded by the FRQS postdoctoral fellowship. ML was supported by the Swiss National Science Foundation grant P5R5PB_211122. UD was funded by an NSERC graduate fellowship. AC is supported by NIH training grant T32HG000035. CRL Holds the Canada Research Chair in Cellular Systems and Synthetic Biology.

## Author contributions

**David Bradley**: Conceptualisation; Data curation; Software; Formal analysis; Investigation; Visualisation; Methodology; Writing—original draft; Writing—review and editing; Co-first author with Alexander Hogrebe.
**Alexander Hogrebe**: Conceptualisation; Data curation; Software; Formal analysis; Investigation; Visualisation; Methodology; Writing—original draft; Writing—review and editing; Co-first author with David Bradley. **Rohan Dandage**: Conceptualisation; Data curation; Software; Validation; Investigation; Visualisation; Methodology; Writing—review and editing. **Alexandre K Dubé**: Validation; Investigation; Visualisation; Methodology; Writing—review and editing. **Mario Leutert**: Data curation; Software; Formal analysis; Investigation; Visualisation; Methodology; Writing—review and editing. **Ugo Dionne**: Data curation; Validation; Investigation; Visualisation; Methodology; Writing—review and editing. **Alexis Chang**: Investigation; Visualisation; Methodology; Writing—review and editing. **Judit Villén**: Conceptualisation; Resources; Supervision; Funding acquisition; Investigation; Project administration; Writing—review and editing; Co-corresponding author with Christian Landry. **Christian R Landry**: Conceptualisation; Resources; Supervision; Funding acquisition; Investigation; Project administration; Writing—review and editing; Co-corresponding author with Judit Villén.

Source data underlying figure panels in this paper may have individual authorship assigned. Where available, figure panel/source data authorship is listed in the following database record: biostudies:S-SCDT-10_1038-S44318-024-00200-7.

## Disclosure and competing interests statement

The authors declare no competing interests.

# Expanded View Figures

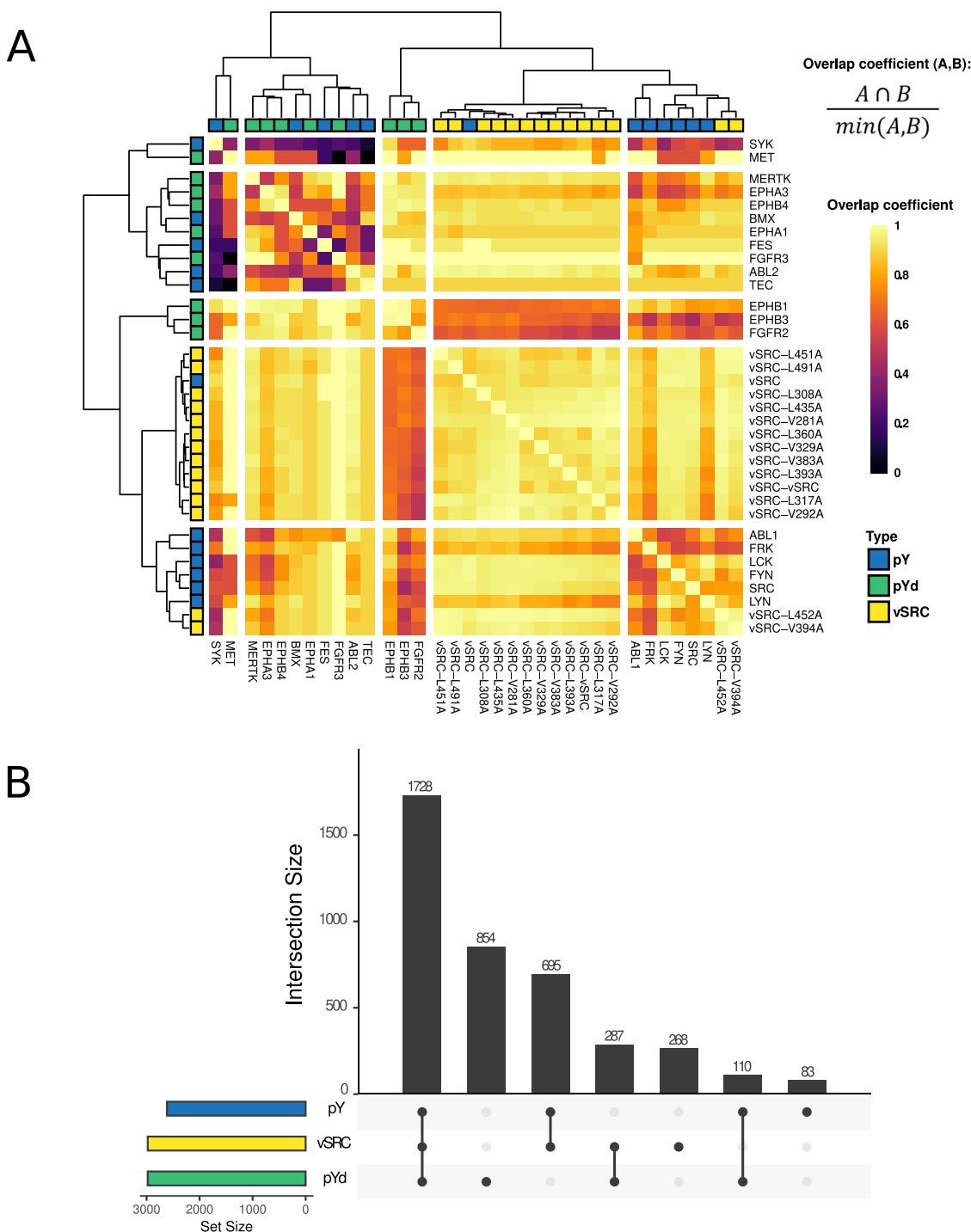

**Figure EV1.  Substrate overlap between different kinases expressed in yeast.**

(A) Phosphoproteome overlap between the different kinase strains. Overlap is with respect to the unique upregulated (WT-dead) pY sites per kinase. Overlap is calculated in terms of the 'overlap coefficient' (top-right), which is the size of the intersection divided by the size of the smallest set. (B) Overlap between the pY sets above for the major pY, v-SRC, and pYd groups. Visualisation is in the form of an UpSet plot.

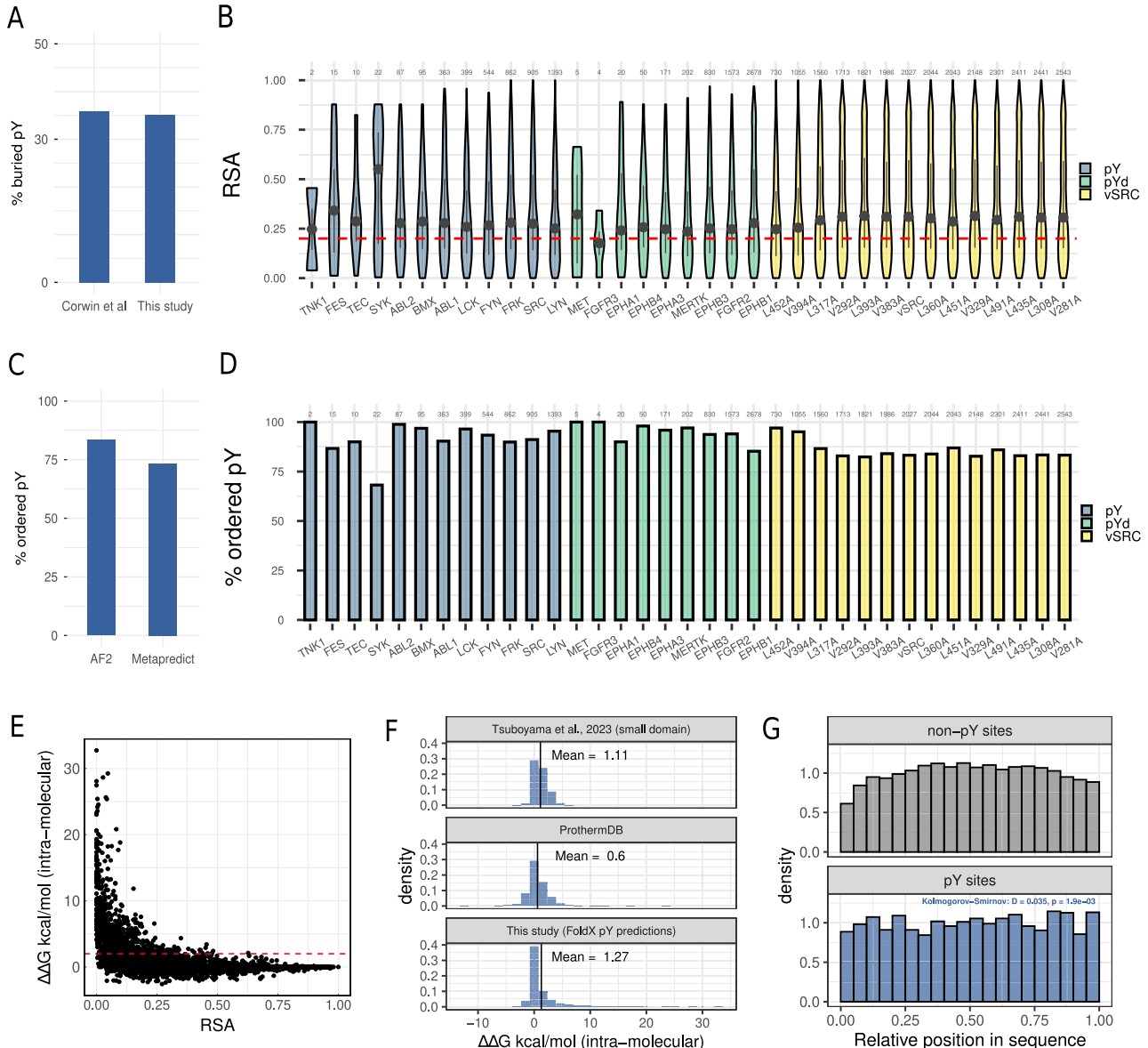

**Figure EV2.   Structural features of spurious phosphorylation.**

(A) For all unique spurious pY reported in (Corwin et al, 2017) and this study, the percentage of sites predicted to be buried from AF2 structures based upon a relative solvent accessibility (RSA) threshold of 0.2 (B) The RSA of upregulated pY sites (WT-dead) for each kinase, divided by group. pY: full-length tyrosine kinases, pYd: tyrosine kinase domains, v-SRC: WT v-SRC and v-SRC mutants. The red dashed line corresponds to the cut-off for buried residues, set at an RSA of 0.2. The black point indicates the median (50th percentile) and the vertical lines connect the lower and upper quartiles (25th percentile and 75th percentile). Numbers above plots represent the number of unique upregulated pY sites (WT-dead) per kinase that could be mapped to an AF2 structure. (C) For all unique spurious pY phosphosites reported in this study, the percentage mapping to predicted ordered regions based upon the AF2 structures (left) and the sequence-based predictor Metapredict (right) (Emenecker et al, 2021). (D) The percentage of spurious pY sites mapping to ordered regions for each kinase, divided by group. Disorder/order predictions made upon the basis of the AF2 models (Akdel et al, 2022; Piovesan et al, 2022). (E) For each unique spurious pY that could be mapped to an AlphaFold2 model ($n = 3981$), the relationship between the RSA and the predicted ΔΔG of phosphorylation (Y to pY). The red dashed line corresponds to a ΔΔG threshold of 2 kcal/mol for destabilising pY. (F) ΔΔG (kcal/mol) distribution for the amino acid mutations reported in the empirical protease-based screen of (Tsuboyama et al, 2023) (top), a compilation of experimental ΔΔGs reported in ProThermDB for amino acid mutations (Nikam et al, 2021) (middle), and the ΔΔGs predicted in this study for spurious tyrosine phosphorylation. (G) Distribution of the relative pY position along the protein length for all unique spurious pY sites (pY) and non-phosphorylated sites on the same proteins (non-pY).

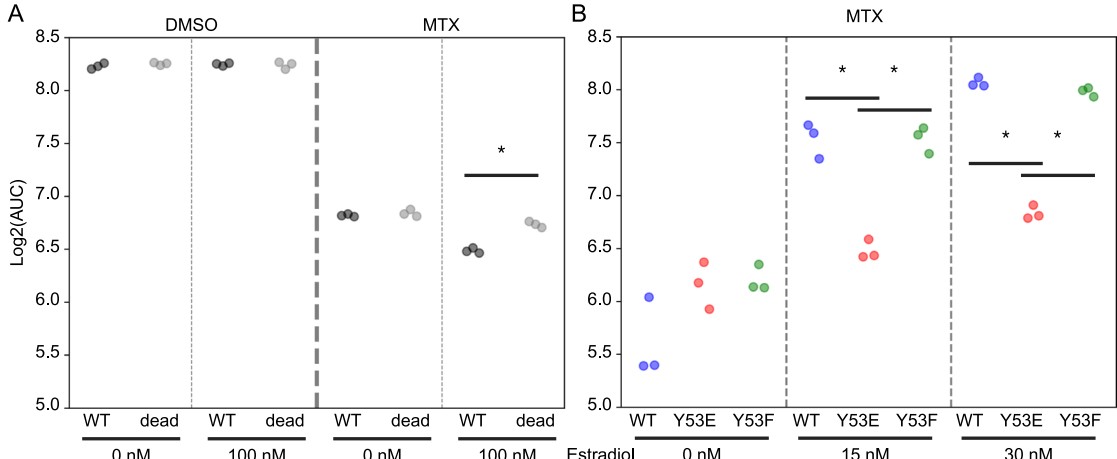

**Figure EV3. Effect of EPHB1 expression on a specific PPI as measured by DHFR PCA.**

(**A**) EPHB1 WT or dead was expressed in a strain to measure the interaction between Rvs167-DHFR[1,2| and Sla1-DHFR[3]. DHFR PCA is a method in which two interaction partners are fused to fragments of methotrexate (MTX) resistant DHFR. If both partners interact, the two fragments assemble into a functional DHFR and allow the growth of the yeast strain in the presence of MTX. The growth of the yeast strain is proportional to the amount of protein complex formed by the two partners. EPHB1 WT expression induces phosphorylation at position Y476 in Rvs167, which is predicted to destabilise the Rvs167-Sla1 interface at a $\Delta\Delta G$ of $+2.15$ kcal/mol, but without any predicted destabilisation of the Rvs167 fold ($\Delta\Delta G$ of $-0.24$ kcal/mol). Strains were grown in PCA media with or without MTX and with or without 100 nM estradiol to induce kinase expression. Strains were grown in triplicates for 72 h and the area under the curve (AUC) was calculated for the three replicates. A statistically significant difference (t-test, WT vs dead, $p = 3.45 \times 10^{-4}$, *) is shown on the graph. (**B**) Rvs167$_{SH3}$ with specific phosphomimetic mutations in the SH3 domain (Y53E and Y53F, corresponding to position Y476 in the full protein) was expressed with estradiol (15 or 30 nM). Y53F prevents phosphorylation; Y53E partially mimics the negative charge of the phosphate, but we caveat that the phosphate group on tyrosine has a larger negative charge and pY is very distinct structurally from the glutamate sidechain (Hunter, 2012; Reinhardt and Leonard, 2023). In this case, we only checked for the effect of the phosphomimetic mutants on the interaction between Rvs167 and Sla1, without the presence of the kinase. This removed all the fitness effects kinase expression could have on cell growth and focused only on the interaction destabilisation. Interaction with Sla1-DHFR[3] was tested for 72 h and AUC was calculated for the three replicates. Statistically significant differences (t-test, 15 nM estradiol $p$ values WT vs Y53E $= 1.49 \times 10^{-3}$, 15 nM estradiol Y53F vs Y53E $= 5.89 \times 10^{-4}$, 30 nM estradiol WT vs Y53E $= 1.01 \times 10^{-5}$, 30 nM estradiol Y53F vs Y53E $= 1.25 \times 10^{-5}$, *) are shown on the graph. These results are in agreement with previous work demonstrating that tyrosine phosphorylation of this position in SH3 domains can perturb SH3 domain-dependent interactions (Dionne et al, 2018). However, we caution that the Y53E mutation is predicted to have a destabilising effect on the SH3 domain itself ($\Delta\Delta G$ of 2.03 kcal/mol), which would also contribute to the reduced formation of the Rvs167-Sla1 interaction (in addition to the destabilisation of the Rvs167-Sla1 interface). The figure was created with Biorender.com.

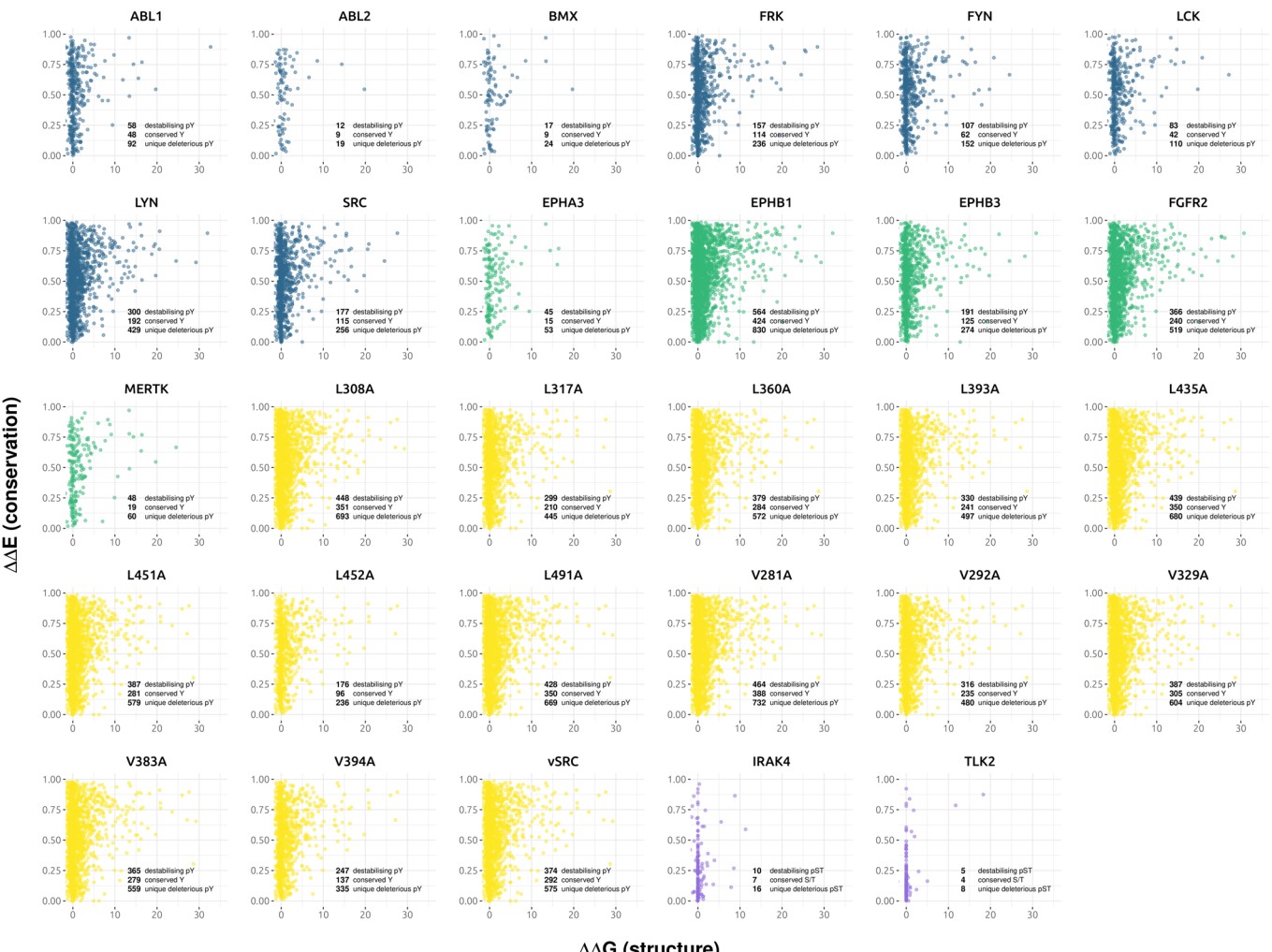

**Figure EV4. Variant effect prediction (VEP) of spurious phosphorylation with respect to protein stability (x-axis) and sequence conservation (y-axis).**

Higher ΔΔG values correspond to more destabilising pY (protein-level) whereas higher ΔΔE values correspond to pY mapping to more conserved Y positions. ΔΔG > 2 and ΔΔE > 0.8 were the thresholds used to determine deleterious pY via structure and conservation, respectively. Kinases are coloured by their groups: blue (full-length tyrosine kinases), green (tyrosine kinase domains), yellow (v-SRC and its mutants), and purple (pS/T kinases).

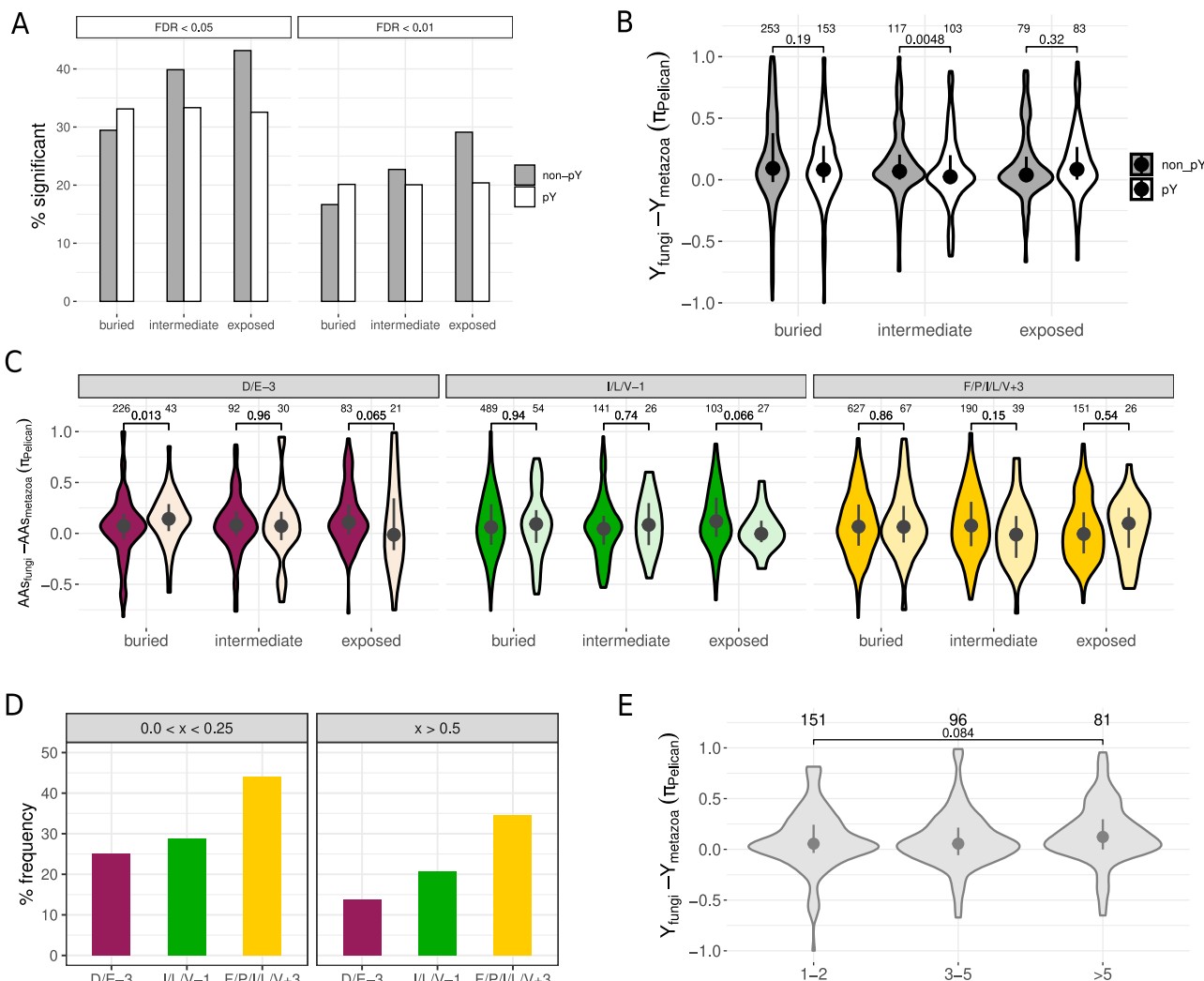

◄

**Figure EV5.   Supplementary results for the site-based analysis of pY counter-selection in metazoan species.**

(A) For spuriously phosphorylated tyrosines (pY) and non-phosphorylated tyrosines (non-pY), the percentage with a significant shift in amino acid profile between animal and fungal species, as inferred by Pelican software (Duchemin et al, 2023). (B) The same analysis as in Fig. 6F but after excluding non-pY sites that are poor candidates for Y phosphorylation on the basis of their sequence motif. Scores with a normalised motif score (0-1) maximum (across Y kinases) lower than 0.7 were excluded. Y kinase specificity models were constructed from data presented in (Sugiyama et al, 2019). The y-axis represents the difference between the inferred preference for Y in fungal species and metazoa species, calculated as their difference in equilibrium frequencies ($\pi$) by the Pelican software (Duchemin et al, 2023). The results are given for pY and non-pY tyrosines and separated according to their solvent accessibility (buried: RSA >0.2, intermediate: 0.2 < RSA <0.4, exposed: RSA >0.4). Sample sizes are buried non-pY ($n = 253$), buried pY ($n = 153$), intermediated non-pY ($n = 117$), intermediate pY ($n = 103$), exposed non-pY ($n = 79$), exposed pY ($n = 83$). The black point indicates the median (50th percentile) and the vertical lines connect the lower and upper quartiles (25th percentile and 75th percentile). In each case, a two-sided Kolmogorov–Smirnov test was performed. (C) Testing for counter-selection against residues that are often found in Y kinase phosphorylation motifs: D/E-3, I/L/V-1, and F/P/I/L/V + 3 (Deng et al, 2014; Li et al, 2023; Sugiyama et al, 2019). The analysis was performed and presented the same way as it is described in Fig. 6F for the central Y residue. The y-axis represents the difference between the inferred preference for the motif amino acids (D + E at position -3, or I + L + V at position-1, or F + P + I + L + V at position +3) in fungal and metazoan species, calculated as their difference in equilibrium frequencies ($\pi$) by the Pelican software (Duchemin et al, 2023). The black point indicates the median (50th percentile) and the vertical lines connect the lower and upper quartiles (25th percentile and 75th percentile). The results are separated according to their solvent accessibility (buried: RSA > 0.2, intermediate: 0.2 < RSA <0.4, exposed: RSA >0.4). The non-phosphorylated sites are in the darker colour for all three panels. The sample sizes above the plots indicate the number of unique sites that fall into each category in terms of their phosphorylation status (non-pY vs pY), motif (D/E-3, I/L/V-1, and F/P/I/L/V + 3), and accessibility (buried, intermediate, exposed). In each case, a two-sided Kolmogorov–Smirnov test was performed to determine statistical significance. (D) Determining whether these motif residues are more likely to be found in strongly counter-selected Ys (right) compared to weakly counter-selected Ys (left). The 'x' parameters refer to the difference in inferred Y preference between fungal and animal species ($\pi_{fungi}$ - $\pi_{metazoa}$), with values closer to 1 being stronger candidates for Y counter-selection in animal species. (E) Test to determine if sites that are phosphorylated by many Y kinases in this dataset are more strongly counter-selected than sites that are phosphorylated by only a small number of kinases. The analysis was performed and presented the same way as it is described in Fig. 6F. The y-axis represents the difference between the inferred preference for Y in fungal species and metazoa species, calculated as their difference in equilibrium frequencies ($\pi$) by the Pelican software (Duchemin et al, 2023). The black point indicates the median (50th percentile) and the vertical lines connect the lower and upper quartiles (25th percentile and 75th percentile). The x-axis represents the number of unique kinases targeting the Y phosphosite. Sample sizes indicate the number of upregulated pY (WT-dead) in each category. This analysis excludes the v-SRC mutants, which overlap strongly in terms of their substrate profiles. The *p* value was inferred from a one-sided Kolmogorov–Smirnov test.

