## [Peer Review File · The EMBO Journal]

The fitness cost of spurious phosphorylation

David Bradley, Alexander Högberg, Rohan Dandage, Alexandre Dube, Mario Leutert, Ugo Dionne, Alexis Chang, Judit Villen, and Christian Landry

Corresponding author(s): Judit Villen (jvillen@uw.edu), Christian Landry (christian.landry@bio.ulaval.ca)

Review Timeline:

Submission Date:	25th Oct 23
Editorial Decision:	11th Dec 23
Revision Received:	7th May 24
Editorial Decision:	22nd Jun 24
Revision Received:	23rd Jul 24
Accepted:	24th Jul 24

Editor: Hartmut Vodermaier / Ioannis Papaioannou

Transaction Report:

Dr. Christian Landry
Université Laval
1030 Avenue de la médecine
Quebec G1V 0A6
Canada

11th Dec 2023

Re: EMBOJ-2023-115986
The fitness cost of spurious phosphorylation

Dear Christian,

Thank you again for submitting your study on the fitness cost of spurious phosphorylation to The EMBO Journal. I apologize for the delay in getting back to you with a decision, but we have now finally received a complete set of comments from three expert referees. As you will see from the reports copied below, the referees appreciate the interest of the topic and the substantive amount of data presented. At the same time, they also note a number of aspects that could be strengthened, in particular with regard to fitness significance and evolutionary implications, as well as overall ramifications of the described findings.

Should you be able to adequately address the listed concerns, we would be happy to consider a revised manuscript further for publication. Since it is our policy to consider only a single round of major revision and therefore important to fully answer to all comments at the time of resubmission, I would invite you to get back to me with a tentative response letter/revision plan already during the early stages of the revision work. On the basis of this response, we could then further discuss how specific major and conceptual issues raised in the three reports might be clarified or answered, and which experimental extensions would seem realistic. I should add that we could also offer extension of the default three-months revision period if needed, with our 'scooping protection' (meaning that competing work appearing elsewhere in the meantime will not affect our considerations of your study) remaining of course valid also throughout this extension.

Detailed information on preparing, formatting and uploading a revised manuscript can be found below and in our Guide to Authors. Thank you again for the opportunity to consider this work for The EMBO Journal, and I look forward to hearing from you in due time.

With kind regards,

Hartmut

9) Digital image enhancement is acceptable practice, as long as it accurately represents the original data and conforms to community standards. If a figure has been subjected to significant electronic manipulation, this must be clearly noted in the figure legend and/or the 'Materials and Methods' section. The editors reserve the right to request original versions of figures and the original images that were used to assemble the figure. Finally, we generally encourage uploading of numerical as well as gel/blot image source data; for details see: embopress.org/page/journal/14602075/authorguide#sourcedata

At EMBO Press, we ask authors to provide source data for the main manuscript figures. Our source data coordinator will contact you to discuss which figure panels we would need source data for and will also provide you with helpful tips on how to upload and organize the files.

In the interest of ensuring the conceptual advance provided by the work, we recommend submitting a revision within 3 months (10th Mar 2024). Please discuss the revision progress ahead of this time with the editor if you require more time to complete the revisions. Use the link below to submit your revision:

Link Not Available

Referee #1:

Summary:

In this manuscript, Bradley et al. design an intriguing experiment to examine whether spurious phosphorylation across a proteome is detrimental to cellular fitness. To ask this question, the express various tyrosine kinases in yeast (which do not contain canonical tyrosine kinases and have very low levels of endogenous tyrosine phosphorylation) and conduct extensive phosphoproteomics experiments along with yeast growth/fitness measurements. It is well established that tyrosine kinase activity is detrimental to yeast fitness, but it is not well-understood why this is the case, nor how much spurious tyrosine phosphorylation across the proteome can be tolerated. The main experiments in this manuscript are very well-designed: each kinase is controlled for with a catalytically dead variant, a range of different kinases (in terms of intrinsic activity and substrate specificity) are chosen, the phosphoproteomics has deep coverage and good reproducibility, and yeast fitness is assessed across a range of conditions. Many of the key conclusions are intriguing and also well-substantiated by their data. Most notably, they show that kinases which phosphorylate more substrates in yeast tend to have a more negative effect on fitness. They also show, surprisingly but convincingly through their phosphoproteomics and structural modeling/analysis, that many of the spurious phosphorylation sites are on buried residues

Ultimately, the authors turn their attention to evolutionary questions. Given my expertise, I am less qualified to assess the validity of their analyses/conclusions, but a few key points were still compelling. For example, they reevaluate the question of whether there has been selective pressure in eukaryotes to reduce the frequency of tyrosine residues in proteomes as tyrosine

kinases became more abundant in those organisms. Their analysis with a more phylogenetically balanced set of organisms convincingly refutes the observation that there is even a negative correlation between tyrosine abundance and tyrosine kinase abundance. Arguments about whether the human tyrosine kinase, expressed in yeast, phosphorylate conserved substrates were much harder to follow.

Overall, I found this manuscript to be interesting and a positive contribution to our understanding of tyrosine kinase activity and evolution. I think it illustrates that proteomes are actually able to buffer/tolerate a substantial amount of biochemical "noise" and provides exciting food for thought with respect to the evolution of signaling systems. Furthermore, the phosphoproteomics datasets here could be of broad interest to researchers interested in yeast signaling pathways (indeed ectopic tyrosine expression seems to modulate endogenous yeast signaling) as well as researchers interest in tyrosine kinase activity and specificity.

Several (mostly minor) suggested improvements to the manuscript are suggested here. A few more substantial experiments are suggested to strengthen the paper, but some may be beyond the scope of necessary revisions.

Suggested main points to address:

1. What are the units for the predicted $\Delta\Delta G$ values in Supplementary Figure 2? How do the magnitude of these predicted free energy changes relate to experimentally measured $\Delta\Delta G$ values from protein folding and mutagenesis experiments? This should be discussed.
2. Can the authors discuss the overlap in spurious phosphorylation sites across their tested kinases? There are some indications of overlap or lack of overlap from the Pearson's correlation coefficients and the tSNE plots in Figure 1 and the logos in Supplementary figure 4, but no explicit numbers are given. Is there a common core set of phosphosites that is phosphorylated by most (if not every) kinase tested? Are there substrates that are only phosphorylated by the kinases that have a strong fitness effect, but not those that do not have a strong fitness effect? This seems like a critical thing to check.
3. One of the more intriguing findings here is that expression of some of the tyrosine kinases results in up- and down-regulation of pS/pT sites, suggesting that the spurious tyrosine phosphorylation results in altered cell signaling. Given that the authors observe modifications on endogenous yeast kinases, can anything specific be said about how these modifications might drive changes in the yeast pS/pT phosphoproteome in response to spurious tyrosine kinase activity? E.g. What signaling pathways are being impacted? Are activation loop S/T residues being differentially phosphorylated?
4. Similar to the point above for kinases, what can be said about the GTPase phosphorylation sites observed here? Do any of these map to known regulatory tyrosine phosphorylation sites on GTPase (such as the well-established Y32 and Y64 phosphosites on mammalian Ras GTPases, see papers by Michael Ohh and co-workers)?
5. The findings on page 11 and in Figure 2H-I about spurious phosphorylation disrupting protein-protein interactions are probably the most experimentally testable conclusions in this paper that are drawn from computational analyses. The paper would be strengthened if a few of the observed effects on protein-protein interactions were tested, for example via co-immunoprecipitation in cells expressing a WT kinase or kinase-dead enzyme.
6. The section where phosphorylation site occupancy/stoichiometry is found to not correlate with fitness was particularly interesting. The authors do quantitative proteomics on the EPHB1 and vSRC samples and show that the median stoichiometry of phosphorylation is similar for these two kinases, even though their fitness effects are very different. This feels like not quite the right experiment to really infer that stoichiometry is not critical, because these kinases are probably phosphorylating different subsets of the phosphoproteome (as per Figure 1. It seems more likely that stoichiometry is critical for fitness, but on a specific set of substrates. This could probably be tested more clearly by comparing the vSrc mutants, which show a range of fitness effects but have very overlapping substrate scopes. Presumably in this context, stoichiometry will correlate very strongly with fitness.
7. Going back to point 2, above, if there are a core set of substrates that are differentially phosphorylated by the "toxic" kinases over the "non-toxic" ones, this would be the key set to look at for stoichiometry-dependence.

Suggested minor points to address:

1. Very minor point, but it might be worthwhile to specify in the main text (not just the methods section) that the kinase-dead mutants were catalytic Asp-to-Asn mutations, since some groups chose to mutate an active site Lys-to-Met. The Asp-to-Asn mutation tends to be more reliably kinase-dead, and indeed this is clear from the data in the paper.
2. It's curious that most of the GFP blots in Supplementary Figure 1 have many bands, and more so that the band pattern/intensity in the GFP blots depends on whether the kinase is active or a kinase-dead mutant. The authors should address if the multiple bands are truncations or something else, and speculate why this differs if the kinase is dead. Additionally, the Landing Pad lane in the blots also has a band. Can the authors clarify if the landing pad alone encodes GFP?

3. The X-axis label "ddG (experimental)" in Supplementary Figure 5e is a bit misleading. Although the text clearly states that these are ddG predictions based on experimental structures vs AlphaFold models, one might mistake the X-values for experimentally determined ddG values. Is there a better way to label this?

Referee #2:

Here, the authors set out to explore how tyrosine phosphorylation might have evolved in an organism that only utilizes Ser/Thr phosphorylation, without causing deleterious physiological effects due to spurious Tyr phosphorylation of proteins. To measure the impact of artificial TK-mediated protein phosphorylation on fitness, they used budding yeast, which lacks conventional tyrosine kinases (TKs), and engineered yeast strains inducibly expressing individually GFP-tagged versions of 24 TKs (either full length or for the RTKs catalytic domains, and for v-Src a set of 13 mutants) and 7 Ser/Thr kinases (lacking obvious yeast orthologues), and then used MS analysis of IMAC-enriched tryptic phosphopeptides to quantitatively analyze their post-induction phosphoproteomes, identifying ~30,000 phosphosites mapping to ~3,500 proteins in five biological replicates, in each case comparing a strain expressing the WT kinase to a strain expressing the cognate kinase-dead mutant. The induced phosphosites sequences were in general consistent with the known primary sequence specificities of the expressed kinases. Using AlphaFold2 (AF2) structural predictions and relative solvent accessibility of the sites to assess the possible functional consequences of the spurious phosphorylation events, they predicted that 80% of pTyr sites map to ordered/buried regions or PPI interfaces. In this way, they defined >1,000 pTyr events that might be deleterious, possibly as a result of destabilizing protein folding. The fitness effects of the 44 protein kinases (13 TKs, 10 TK catalytic domains, 7 Ser/Thr kinases, and 14 v-Src TK mutants) were tested under 41 conditions known to induce stress, finding that 5, 3, 3, and 14 kinases, respectively, in these four categories were deleterious for growth/proliferation. In general, they observed correlation between the number of spurious pTyr sites and decreased yeast growth, but a large number of the TKs and spurious pTyr sites apparently had little effect on yeast fitness, possibly due to a low phosphorylation stoichiometry; for instance, expression of EPHB3, which led to 800 spurious pTyr sites, was not toxic. Experimentally, they determined an average phosphosite stoichiometry of 16% for v-Src and 20% for EPHB1 sites, and showed that phosphorylation of Tyr sites in essential proteins did not give a more significant correlation. They also found that compared to spurious pTyr sites in yeast, native pTyr sites in human proteins were significantly more likely to be accessible and in disordered regions, with spurious phosphorylation of yeast proteins being weakly biased towards orthologues of human proteins that are known TK targets. Finally, they reexamined the issue of whether the advent of tyrosine phosphorylation might have resulted in counterselection against the presence Tyr in the proteome, and found that the representation of Tyr was not significantly lower in organisms that express TKs, other than in the yeast proteome.

In these studies, the authors have addressed experimentally an important question, namely how protein tyrosine phosphorylation might have evolved as a regulatory mechanism in an organism utilizing only Ser/Thr protein kinases. By using yeast, which lacks tyrosine kinases, as a model organism in which to inducibly express a series of TKs, they found that expression of some TKs is deleterious to yeast proliferation, whereas expression of others is not, and by characterizing the phosphoproteomes in these cells developed plausible explanations for the deleterious effects of spurious Tyr phosphorylation. However, as the authors admit, a limitation of this extensive analysis is that it remains unclear whether it is the pTyr sites in a specific subset of proteins whose phosphorylation is deleterious to yeast proliferation or whether it is a cumulative effect of multiple phosphorylations at less specific sites that decreases fitness. They did not attempt to make any Tyr to Phe mutants at sites in target proteins predicted to be particularly deleterious to test this, although this would be a major undertaking without any guarantee of success. A major omission is that the authors do not mention anywhere in the paper the possible role of protein-tyrosine phosphatases (PTPs) in determining the effects of spurious Tyr phosphorylation on yeast fitness, which is surprising given that budding yeast express four PTPs that could, perhaps selectively, dephosphorylate some of the spurious pTyr sites. This seems to be a serious oversight that needs to be rectified (and possibly tested experimentally - see point 1).

Nonetheless, these are interesting studies that have some bearing on how tyrosine phosphorylation might have evolved as a protein modification involved cell signaling in single cell eukaryotes a few hundred million years ago that are worth reporting, even if the conclusions are not totally definitive.

1. Budding yeast express three classical PTPs possessing HCSAGCGR PTP catalytic motifs and PTP activity (Ptp1, Ptp2, and Ptp3), and the yeast cell cycle progression is regulated by dephosphorylation of pY15 CDK by Mih1, the budding yeast CDC25 orthologue. PTPs have very high turnover numbers, and determining the consequences of knocking out one or more of the three Ptp genes on pTyr site stoichiometry and the toxicity of exogenous TK expression could be informative. In this context what happens to pTyr levels if yeast cells are treated with pervanadate to inhibit the three canonical yeast PTPs? The authors used a phosphatase inhibitor cocktail to inhibit endogenous phosphatases and induce cellular stress (Figure S8), but it is not stated what inhibitors were included in the cocktail - was pervanadate one of them? In terms of discussing the relevance of dephosphorylation of spurious pTyr to the observed phenotypes, other less specific intracellular small molecule phosphatases might also be able to dephosphorylate pTyr residues, if they are exposed on the surface of proteins. Finally, in from an evolutionary aspect it has been suggested that PTPs had to evolve before TKs in order to avoid the deleterious effects of spurious Tyr phosphorylation (PMID: 19269802).

With regard to whether spurious pTyr dephosphorylation mediated by endogenous phosphatases is important it would be

informative to know how fast, if at all, spurious pTyr sites are dephosphorylated when the TK in question is switched off with a selective TK inhibitor. If a spurious pTyr phosphosite cannot be dephosphorylated in the cell, then its level of phosphorylation should continuously accumulate after induction of TK expression - was a post-induction phosphoproteomic time course carried out?

2. The authors have inferred that the observed cell phenotypes are due to spurious Tyr phosphorylation based on the fact that they are induced by expression of the WT but not kinase-dead kinase. However, another way of establishing this would be to culture the cells in the presence of cognate inhibitors of toxic TKs to allow proliferation (as has often been done when toxic TKs are expressed in bacteria and also in yeast). Also, did the authors try inducibly expressing a mammalian PTP, YopH or lambda PP to determine whether this reverted any growth phenotypes?

Other points: 1. Page 5: Several of the TKs that were expressed exogenously are known to be clients for the CDC37/HSP90 kinase-specific chaperone complex - are the Cdc37/Hsp90 yeast genes important for deleterious exogenous TK activity?

3. Page 5: Did the authors make use of the primary sequence specificities recently reported by Johnson et al. (PMID: 3663161) for the 7 S/T kinases they analyzed? In addition to the TK specificities published in the recent Shah group paper (PMID: 36927728), which the authors used, the Cantley group has presented at meetings the results of a companion study on the primary sequence specificities of all the active human TKs; this paper will presumably be published soon and the TK primary sequence specificities should be useful for further analysis of the authors' pTyr proteomic datasets.

4. Pages 6 and 10: Did expression of any of the TKs increase CDK1/Cdc28 pY14 levels and cause cell cycle arrest?

5. Did the authors test a pTyr-peptide specific enrichment protocol, e.g. with super-binder SH2 protein, to see whether additional pTyr sites could be identified? It is not clear from the methods whether phosphatase inhibitors were used in the lysis procedure, and, given the existence of PTPs in budding yeast, the inclusion of a PTP inhibitor, such as pervanadate, would be advisable.

6. Page 7/Figure 2A: Solvent accessibility per se may not be a stringent enough criterion to define accessible Tyr sites for TK phosphorylation, since in general Ser/Thr/Tyr in alpha helices are not efficiently phosphorylated, if at all, by most ePKs. Another question is when do the buried pTyr sites become phosphorylated - are such Tyr sites phosphorylated during co-translationally when the protein is unfolded, or alternatively are they phosphorylated in a "denatured" subpopulation of the protein?

7. Page 10: Did the authors try AlphaFold2-multimer to predict effects of pTyr phosphorylation on PPIs, in addition to InteractomeInsider? As noted, many PPIs involve SLiMs - were any of the novel pTyr sites in or near known SLiMs?

8. Page 10: How many endogenous yeast pTyr sites are known?

9. Page 10: Did the authors attempt to measure the effect of spurious Tyr phosphorylation on the function of any of these yeast PKs or other proteins using in vitro phosphorylation by the TK in question of WT compared to a Tyr to Phe mutant form of the protein? Alternatively, they could use expanded genetic code technology to stoichiometrically install a pTyr at a spurious pTyr site of interest (PMID: 2804693) to determine its effect on protein folding and function.

10. Figure 2H: Y32 is a known phosphorylation site in KRAS, and is proposed to inhibit its activity; was phosphorylation of the equivalent Y39 observed in the Ras1/2 proteins in any of the strains expressing active TKs?

11. Page 14: The ultimate experiment would be to mutate a set of spurious pTyr sites in the proteins predicted to be the most important for a particular TK, although this would be beyond the scope of this paper.

12. Page 16: Here, by comparing peak intensities of unphosphorylated and phosphorylated peptides from WT and dead kinase expressing cells, they authors assessed the stoichiometry of spurious pTyr site phosphorylation, an issue they really should have raised earlier. Can the authors confirm by another method that the 16% median stoichiometry they inferred for v-SRC phosphosite phosphorylation is an absolute stoichiometry.

13. Figure 4/page 18: They showed that TK protein targets in yeast commonly have a pTyr protein orthologue in humans, and it is interesting that 20% of the pTyr sites are conserved in the human orthologue, but in human proteins do the mapped pTyr sites lie in a SLiM, or another kinase interacting surface, etc.?

14. It would be interesting to know how many more of the spurious, evolutionarily conserved pTyr sites in yeast orthologues would be detected in human cells treated with pervanadate to abolish PTP activity.

15. Page 22: In addition to undergoing a WGD, it has been suggested that budding yeast may also have lost a number of PK families during evolution, and it is possible that the reduced level of Tyr in the proteome may have occurred earlier during budding yeast evolution for another reason.

16 Page 22: Since PTPs may have had to evolve before TKs to avoid the deleterious effects of spurious Tyr phosphorylation

(PMID: 19269802), it is possible that the lack of correlation depended on the level of either specific or non-specific pTyr phosphatases expressed in an early organism.

17. Page 27: Yeast does have a single SH2 domain protein, Spt6, which acts as histone chaperone, but may bind pSer rather than pTyr, and so would be considered atypical.

18. As an aside, Michael Snyder's group, which worked extensively on the yeast kinome a few years ago, reported in abstract form that 20% of recombinant yeast PKs can phosphorylate polyGluTyr. The physiological significance of this activity was not determined, but could be a reason why PTPs were evolved first.

Referee #3:

Bradley et al. The fitness cost of spurious phosphorylation,

Bradley et al are assessing artificial tyrosine phosphorylation in yeast by introducing tyrosine kinases. It has been shown that high tyrosine activity is deleterious to yeast cell growth, a fact which is not understood mechanistically. The authors describe various aspects of spurious (i.e. all the proteins which are phosphorylated) phosphorylation through computational analyses of the yeast substrates.

Spurious sites are recorded from recording phospho-proteomes from strains overexpressing GFP tagged non RTKs (13), kinase domains of RTKs (10), S/T kinases (7) and a series of vSRC (13) variants. In total, this is 4,082 upregulated pY sites mapping to 1,970 proteins and 9,014 up- and down-regulated pS/T sites mapping to 2,361 proteins, however the analyses deal with the fraction of pY sites only.

In the first part the authors analyze effects of all pY sites (as the union of all kinase overexpressing strains combined into one data set) on protein stability and PPIs based on computational structural models. In fact, the pY sites largely stem from the vSRC strains, and the other kinases contribute little if at all (the potential effects of individual experiments are completely lost, are there common sites etc. that relate to fitness?).

Some analyses are recaps from the literature, some are novel. Figure 2f is key, summarizing all the findings such as sites predicted to destabilize a protein (using alpha fold models), a small fraction at interfaces predicted to destabilize PPIs (using interactome 3D). It is very hard to follow the conclusion that the analysis of spurious pY predicts the widespread destabilization of proteins and their PPIs. Only a small fraction actually maps to interaction surfaces and an even smaller subset is predicted to change ddG. Actually, the data are presented in a descriptive manner using certain cut offs and no statistical assessment, e.g. through randomization etc. is given in Figure 2. I want to clearly say the analyses are interesting but the relevance is unclear. Moreover the fitness data do not support the analysis which is at some point also stated by the authors themselves: "Paradoxically, while we observe a strong negative correlation between the number of spurious pY sites and fitness (Figure 3e-f), it is also apparent that many such sites predicted to be deleterious have a negligible effect on fitness."

The second part deals with fitness. The fitness effects of the above mentioned 44 kinases and were tested across 41 conditions known to induce various stresses (size of yeast colonies over time as a proxy for fitness), which again results in an extensive body of data. There are some conditional fitness effects, with the underlying mechanism remaining unclear. Figure 3f: fitness correlates with the number of pY sites! This is nice and two explanations are offered as possible explanation: spurious phospho-burden or the increase of stoichiometry of deleterious phospho-sites. However, except for the 3f correlation, no other clear result is obtained that connects the fitness effects to spurious phosphorylation / stoichiometry. Notably, higher stoichiometry, which stems from more active kinases (vSrcs), will also result in a higher number of yeast proteins being phosphorylated.

Evolutionary aspects: Comparisons of pY-proteins in yeast to human pY proteins, on a structural level reveals low conservation.

The Y counter selection hypothesis: Using new phylogenetic data/ new species, the authors do not find significant relationship between the number of predicted tyrosine kinases and proteome Y content, in contrast to the hypothesis by Linding et al. The analysis does contribute to the remaining controversy in the literature through novel sequence analysis, however is not based on the data presented in the manuscript (It is rechecked on spuriously phosphorylated proteins in the supplement). They also investigate Y deserts in individual proteins again without significant results. Also evidence for counter selection on the basis of individual sites is not strong but cannot be rejected based on the presented analysis either. This analysis does not root on the data presented in the manuscript and therefore feels like a relatively loosely related appendix to the analyses.

In summary, the topic is very interesting, the data body is substantial, however the analyses are very descriptive with no clear outcome of the many analyses. The authors do not develop a clear hypothesis (e.g. for further testing) from the analyses. A potential key finding put forward in the manuscript, namely that spurious phosphorylation impacts through protein interactions on fitness, is not sufficiently supported by the data/analyses. Similarly, effects on protein stability can not be connected to the fitness phenotypes. However, I do understand that it is not a case of identifying individual sites ...

Reviewer 1

Summary:

In this manuscript, Bradley et al. design an intriguing experiment to examine whether spurious phosphorylation across a proteome is detrimental to cellular fitness. To ask this question, they express various tyrosine kinases in yeast (which do not contain canonical tyrosine kinases and have very low levels of endogenous tyrosine phosphorylation) and conduct extensive phosphoproteomics experiments along with yeast growth/fitness measurements. It is well established that tyrosine kinase activity is detrimental to yeast fitness, but it is not well-understood why this is the case, nor how much spurious tyrosine phosphorylation across the proteome can be tolerated. The main experiments in this manuscript are very well-designed: each kinase is controlled for with a catalytically dead variant, a range of different kinases (in terms of intrinsic activity and substrate specificity) are chosen, the phosphoproteomics has deep coverage and good reproducibility, and yeast fitness is assessed across a range of conditions. Many of the key conclusions are intriguing and also well-substantiated by their data. Most notably, they show that kinases which phosphorylate more substrates in yeast tend to have a more negative effect on fitness. They also show, surprisingly but convincingly through their phosphoproteomics and structural modeling/analysis, that many of the spurious phosphorylation sites are on buried residues

Ultimately, the authors turn their attention to evolutionary questions. Given my expertise, I am less qualified to assess the validity of their analyses/conclusions, but a few key points were still compelling. For example, they reevaluate the question of whether there has been selective pressure in eukaryotes to reduce the frequency of tyrosine residues in proteomes as tyrosine kinases became more abundant in those organisms. Their analysis with a more phylogenetically balanced set of organisms convincingly refutes the observation that there is even a negative correlation between tyrosine abundance and tyrosine kinase abundance. Arguments about whether the human tyrosine kinase, expressed in yeast, phosphorylate conserved substrates were much harder to follow.

We appreciate that the reviewer recognizes the significance of our study, but apologize that the section on human-yeast conservation was difficult to follow. We have now made major changes to this section to make it more accessible. Firstly, we have reduced the length of the main text to make it less descriptive and to emphasise the major findings. Secondly, we have reduced the number of main figures panels from 9 to 6, again to emphasise our most significant findings only. The remaining panels are now in an Appendix Figure S23. Finally, for the analysis of human-yeast conservation (protein level, protein-pY level, and site level), we have added cartoon icons to the figure panels to show explicitly the relationship between proteins in terms of their orthology and phosphorylation status.

The new figure is as follows:

For clarity we also show the new figure legend here (new text in blue):

Figure 5 Conservation between spurious pY in yeast and native pY in human. A) Human-yeast conservation at the whole protein level. 'Yeast pY': number of unique spurious pY proteins in yeast, excluding any native pY proteins. 'Yeast pY: human (observed)': number of observed unique spurious pY proteins in yeast with at least one ortholog in human. 'Yeast pY: human (expected)': number of expected unique spurious pY proteins in yeast with at least one ortholog in human. **Expected frequencies derived from the number of observed orthologs for yeast proteins that are not tyrosine-phosphorylated.** H symbol: human. Y symbol: yeast. P symbol: phosphosite. **Dotted lines represent orthology relationships. B)** Human-yeast conservation at the level of whole protein pY phosphorylation. 'Yeast pY: human': number of observed unique spurious pY proteins in yeast with at least one ortholog in human. 'Yeast pY: human pY (observed)': number of observed unique spurious pY proteins in yeast with at least one ortholog in human that is Y-phosphorylated. 'Yeast pY: human pY (expected)': number of expected unique spurious pY proteins in yeast with at least one ortholog in human that is Y-phosphorylated. **Expected frequencies derived from the number of observed tyrosine-phosphorylated orthologs for yeast proteins that are not tyrosine-phosphorylated.** H symbol: human. Y symbol: yeast. P symbol: phosphosite. **Dotted lines represent orthology relationships. C)** Relative frequency (summing to 1) of Pfam domain phosphorylation in yeast (x-axis) and human (y-axis). Abundance is given in the units of log₁₀ (parts per million). Scatter plot includes only domains supported by at least 5 unique pY sites in either human or yeast. **D)** site-specific (i.e. alignment-based) conservation between spurious pY and human native pY. As a percentage of all unique yeast spurious pY (yeast pY, yellow), all unique yeast spurious pY with at least one human ortholog (yeast pY: human, blue), all unique yeast spurious pY with at least one pY-phosphorylated human ortholog (yeast pY: human pY (observed), purple), and all unique yeast spurious pY with at least one pY-phosphorylated human ortholog with x100 randomisations of the human pY positions. **E-F)** For yeast non-pY Y residues, spurious pY residues, native pY residues, human non-pY Y residues, and human pY residues, distribution of surface accessibility (RSA), **and** percentage predicted to be destabilising for the protein fold (using a threshold of $\Delta\Delta G > 2$ kcal/mol). **In both cases $p << 1 \times 10^{-16}$ (Kruskal-Wallis test).**

We thank the reviewer for pointing this out and hope that these changes combined together will make this section easier to understand.

Overall, I found this manuscript to be interesting and a positive contribution to our understanding of tyrosine kinase activity and evolution. I think it illustrates that proteomes are actually able to buffer/tolerate a substantial amount of biochemical "noise" and provides exciting food for thought with respect to the evolution of signaling systems. Furthermore, the phosphoproteomics datasets here could be of broad interest to researchers interested in yeast signaling pathways (indeed ectopic tyrosine expression seems to modulate endogenous yeast signaling) as well as researchers interest in tyrosine kinase activity and specificity.

Several (mostly minor) suggested improvements to the manuscript are suggested here. A few more substantial experiments are suggested to strengthen the paper, but some may be beyond the scope of necessary revisions.

Suggested main points to address:

1. What are the units for the predicted ddG values in Supplementary Figure 2? How do the magnitude of these predicted free energy changes relate to experimentally measured ddG values from protein folding and mutagenesis experiments? This should be discussed.

The ddG units are in kcal/mol. This has been added to the appendix figures / EV figure panels where it was previously missing. We have now compared our predicted ddGs (for pY phosphorylation) with two different data sets based on empirical data. The first is the recent mega-scale protease-based screen of folding stability published in Nature (Tsuboyama et al., 2023, PMID: 37468638), where the data generated was combined with a thermodynamic model to infer the ddG of folding after mutation. The second is a compilation of experimental ddGs (for mutations) from the ProThermDB database (Nikam et al., 2021, PMID: 33196841).

The results are as follows:

As can be seen, the ddG distributions are quite similar across the three data sets. The main differences are that from this study we very rarely predict phosphosites that will stabilise protein folds (i.e. ddG < 0), and also from this study we see a tail of extreme ddG values (>7 kcal/mol, ~95th percentile) that are rarely observed in the empirical data sets.

We have added a new sentence to the main text and a new EV figure panel to describe these results (**EV2F**). We thank the reviewer for this helpful suggestion.

2. Can the authors discuss the overlap in spurious phosphorylation sites across their tested kinases? There are some indications of overlap or lack of overlap from the Pearson's correlation

coefficients and the tSNE plots in Figure 1 and the logos in Supplementary figure 4, but no explicit numbers are given. Is there a common core set of phosphosites that is phosphorylated by most (if not every) kinase tested? Are there substrates that are only phosphorylated by the kinases that have a strong fitness effect, but not those that do not have a strong fitness effect? This seems like a critical thing to check.

We thank the reviewer for this useful suggestion. We have now prepared an EV figure (EV figure 1) that shows how the set of upregulated pY sites (wt vs dead) per kinase overlaps between different human kinases. For this purpose, we use the 'overlap coefficient', which is equal to the size of the intersection divided by the size of the smaller set being compared (see below).

Indeed we see that the overlap is quite substantial and the different kinase groups (pY, pYd, and vSRC) are clearly discriminated. For example, about two thirds of all sites phosphorylated by WT vSRC are also phosphorylated by EPHB1. Compared to the correlation heatmap in Figure 1B, the vSRC mutants here overlap strongly with most pY/pYd kinases, with the relatively low correlations in Figure 1B probably reflecting differences in (indirect) pS/pT regulation between kinase strains.

The results can be seen in the heatmap below (panel A):

We now provide an EV table (EV Table 3) that gives the overlap coefficient for each combination that can be seen in the heatmap.

In panel B, we have generated an UpSet plot to show the overlap in upregulated sites (WT - dead) between the main kinase groups (pY, vSRC, pYd). This UpSet plot shows that there is a core set of 1728 sites that is phosphorylated across all of the major kinase groups.

Finally, we follow the reviewer suggestion and check for substrates that are only phosphorylated by the kinase found to be toxic in our fitness screens. We give the results in an UpSet plot:

As can be seen, the 'strongly toxic' only set contains 2460 pY sites and is not very discriminating in terms of identifying the phosphosites that are responsible for toxicity. The possibility also remains that the phosphosites causing toxicity are phosphorylated by both kinase sets but at higher stoichiometry by the 'strongly toxic' kinases.

3. One of the more intriguing findings here is that expression of some of the tyrosine kinases results in up- and down-regulation of pS/pT sites, suggesting that the spurious tyrosine phosphorylation results in altered cell signaling. Given that the authors observe modifications on endogenous yeast kinases, can anything specific be said about how these modifications might drive changes in the yeast pS/pT phosphoproteome in response to spurious tyrosine kinase activity? E.g. What signaling pathways are being impacted? Are activation loop S/T residues being differentially phosphorylated?

We agree with the reviewer that this would be interesting to check. Therefore, we have examined multiple aspects of pS/pT phosphorylation to infer changes in the cell signaling state after human TK expression. These are represented by a new figure (Figure 3) and section we have generated to show this analysis:

In panel A we infer changes in S/T kinase activity based upon the increased or decreased phosphorylation of known S/T kinase substrates. This indicates for example the increased activation of the CDC28 kinase after the expression WT Src and vSRC.

In panel B we represent the phosphorylation (or dephosphorylation) of native pY sites in yeast that are known to have a regulatory effect on S/T kinase activity. Most of these map to S/T kinase activation loops. For example, for the MAPKs SLT2 and HOG1, we see increased phosphorylation by some Y kinases of regulatory Y sites on the activation loop (HOG1 Y176 and SLT2 Y192).

In panel C, we turn our attention to well-characterised MAPK pathways and check the phosphorylation of known functional pY sites mapping to upstream kinases in the MAPK cascade (top layer), the MAPK itself (middle layer), and known downstream targets of the MAPK cascade (bottom layer). We see increased Y phosphorylation of the MAPK itself and known downstream targets of the pathway. However, we do not observe phosphorylation of upstream kinases in the MAPK cascade.

From this, we present a model in panel D where spurious Y kinase activity leads to inappropriate activation of MAPK cascades by targeting the MAPK directly instead of the upstream kinases (MAPKK or MAPKKK).

During the course of the revisions, we performed some other analyses related to pS/pT signalling.

For example, we checked also to see if any of the pS/pT phosphorylation profiles related to distinctive signatures ('modules') first described in Leutert et al., 2023 (NSMB, PMID: 37845410):

We see the strongest activation of module 18 that is associated with ion transport across the plasma membrane.

As mentioned, this module is involved in ion transport across the plasma membrane and was discussed previously in Leutert et al., 2023 (NSMB, PMID: 37845410); visual description of module 18 is pasted below.

We also compare the pS/pT we see with that from other stress responses (e.g. cell wall stress, metal stress, cold stress, etc.) in yeast, again using data from Leutert et al., 2023 (NSMB, PMID: 37845410). From this analysis we do not see any strong correlation with other stress responses, though there is weak anti-correlation with C/N starvation:

We have included these other analyses in the response to the reviewers for the record, but decided not to include them in the revised manuscript, as the strength of the signal is low in each case.

However, we have mapped the regulated pS/pT sites to native yeast activation loops (following the analysis of activation loop pYs in Figure 3B), and have chosen to represent this data in the new Appendix figure S9:

We have also added a new sentence to the main text describing that we see many regulated sites mapping to the kinase activation loops and especially for the v-SRC kinase variants.

4. Similar to the point above for kinases, what can be said about the GTPase phosphorylation sites observed here? Do any of these map to known regulatory tyrosine phosphorylation sites on GTPase (such as the well-established Y32 and Y64 phosphosites on mammalian Ras GTPases, see papers by Michael Ohh and co-workers)?

We have now checked this. While most of our Ras GTPase pY sites do not map to these positions, we have found two that do: one on Ypt1 Y37 and one on Rho Y71. Given the literature cited by the reviewer (from Michael Ohh), this likely indicates inhibited GTPase activities for these proteins.

We have added a sentence to describe these findings and also have generated a new appendix figure (Appendix figure S8) that indicate the log2-fold change and significant upregulation of each relevant phosphopeptide:

5. The findings on page 11 and in Figure 2H-I about spurious phosphorylation disrupting protein-protein interactions are probably the most experimentally testable conclusions in this paper that are drawn from computational analyses. The paper would be strengthened if a few of the observed effects on protein-protein interactions were tested, for example via co-immunoprecipitation in cells expressing a WT kinase or kinase-dead enzyme.

We have followed up this question experimentally. To do this we performed the protein complementation assay (PCA) that has been mastered in the Landry lab. This is a split-protein complementation assay where the binding of the bait and prey proteins is indicated by the reconstitution of a full DHFR enzyme that is resistant to drug (methotrexate) treatment, thus allowing growth. We performed this test for the interaction between Rvs167 and SLA1, which we predict to be destabilised by the phosphorylation of Rvs167 at Y476 ($\Delta\Delta G = +2.15$ kcal/mol), which is position 53 in the SH3 domain. The results are as follows:

Panel A (left) is performed in the presence of the DMSO control where there are no limits on growth. Panel A (right) shows the results in the presence of methotrexate (MTX), which inhibits native DHFR; growth only occurs when the tagged Rvs167 and SLA1 proteins interact to reconstitute MTX-resistant DHFR. Estradiol is used to induce human kinase expression (EPHB1). Comparing the WT vs dead results in the presence of 100nM of estradiol shows that kinase expression significantly weakens the Rvs167-SLA1 interaction.

As a further control, we performed the DHFR-PCA assay in the absence of the human kinase EPHB1 (panel B). Estradiol here controls the expression of the Rvs167 mutant constructs. The Y53F mutation prevents phosphorylation, whereas the Y53E mutation mimics the negative charge of pY. Y53E significantly weakens the Rvs167-SLA1 interaction at 15nM and 30nM. The effect size is stronger than what we see for phosphorylation most likely because the Y53E mutant is present at an effective stoichiometry of 100%.

We note also that these results are in agreement with our previous work demonstrating that the tyrosine phosphorylation of this position in the SH3 domain can perturb SH3 domain-dependent interactions (Dionne et al. 2018).

We have now provided this new result as **EV figure 3** alongside a new sentence in the main text.

6. The section where phosphorylation site occupancy/stoichiometry is found to not correlate with fitness was particularly interesting. The authors do quantitative proteomics on the EPHB1 and vSRC samples and show that the median stoichiometry of phosphorylation is similar for these two kinases, even though their fitness effects are very different. This feels like not quite the right experiment to really infer that stoichiometry is not critical, because these kinases are probably phosphorylating different subsets of the phosphoproteome (as per Figure 1. It seems more likely that stoichiometry is critical for fitness, but on a specific set of substrates. This could probably be tested more clearly by comparing the vSrc mutants, which show a range of fitness effects but have very overlapping substrate scopes. Presumably in this context, stoichiometry will correlate very strongly with fitness.

We thank the reviewer for this thoughtful comment. We would say that the point of our vSRC/EPHB1 stoichiometry analysis was more to show that average stoichiometry for both kinases seems far below 100%, and so the effect of phosphorylation in most cases would be less disruptive than what is predicted by bioinformatics softwares (which implicitly assume that every copy in the proteome will be mutated or modified). Indeed, in the paper we do not make the claim that stoichiometry is not critical for fitness because – as reviewer #1 correctly points out – vSRC and EPHB1 are phosphorylating different sets of substrates. This is seen more clearly now in the new EV figure 1 and EV table 3 that shows the overlap between EPHB1 and vSRC substrates.

We agree that the suggested experiment would be interesting in terms of being able to relate the activity of vSRC mutants to the average stoichiometry of substrates (i.e. stoichiometry should increase as vSRC becomes more active, as the reviewer states). However, in our view, this would still not enable us to pinpoint the phosphosites directly responsible for toxicity, or to fully address the critical question of whether toxicity is caused by a few important phosphosites or a larger number of phosphorylated substrates.

Therefore, in our respectful view the suggested experiments are outside the scope of this paper.

7. Going back to point 2, above, if there are a core set of substrates that are differentially phosphorylated by the "toxic" kinases over the "non-toxic" ones, this would be the key set to look at for stoichiometry-dependence.

We agree with the reviewer in principle. However, as explained in our response to point 2, we still see a large number of sites (n=2460) that are phosphorylated only by the toxic kinases. Also, it is possible that many of these sites do not contribute to toxicity but just reflect the greater enzyme activity (on average) of the 'toxic' kinases. Thus, without knowing for certain which sites are leading directly to toxicity, we would have to measure stoichiometry for all 'toxic-only' sites (a large set of sites) and across multiple kinase strains, which would be experimentally challenging. Again, the set of possible sites is so large that it may not greatly help us to pinpoint the mechanism of toxicity. The possibility also remains that the site(s) contributing to toxicity are phosphorylated by all kinases, but at very low stoichiometry by the 'non-toxic' ones so that no measurable fitness effect is observed.

In place of this, with added computational analysis we may prioritise sites that are most likely to be deleterious on the following basis:

- They map to essential proteins
- Their log₂fc (WT - dead) correlates with toxicity (WT - dead) among the vSRC mutants
- They map to a highly conserved position
- They destabilise the protein fold

The second bullet point corresponds to the log₂ fold change of phosphorylated sites (WT-dead) across vSRC strains correlated against changes in fitness (WT-dead) across vSRC strains. As expected, these correlations tend to be negative i.e. as the log₂fc of phosphorylation increases, fitness tends to decrease:

minimum fitness (WT–dead) vs. psite log2fc (WT–dead)

Sites with a large negative correlation are those whose phosphorylation correlates with a decrease in fitness. From this analysis we have produced a list of all unique pY phosphosites that are significantly upregulated (WT-dead) and ranked on the basis of predicted deleteriousness. This spreadsheet forms the new Table EV 11.

We first separate proteins according to whether they are essential or non-essential for growth. Then the sites are ranked (with the essential/non-essential) groups on the basis of the fitness vs. log2fc correlations described directly above. In the adjacent columns, ddE (conservation) and ddG (structure) values are given, and so the sites may easily be sorted on the basis of conservation or structural effects (as desired):

	A	B	C	D	E	F	G	H	I	J
1	gene	protein	pos_tag	ensembl_IDs	essentiality	vSRC_correlation	ddE_normalised	ddG_intra	paxdb_abundance	Native
2	DED1	P06634	Y220	YOR204W	essential	-0.956043956043956	0.384883619562375	0	349	No
3	RPT2	P40327	Y50	YDL007W	essential	-0.942857142857143	0.10815367939299	0	49.5	No
4	PAB1	P04147	Y319	YER165W	essential	-0.938461538461538	0.428532336039405	-0.383379	1976	No
5	DPS1	P04802	Y297	YLL018C	essential	-0.937062937062937	0.738020981003686	0.288623	823	No
6	PRE10	P21242	Y26	YOR362C	essential	-0.936363636363636	0.930542862365198	2.19968	301	No
7	ACT1	P60010	Y53	YFL039C	essential	-0.934065934065934	0.409838369641602	2.32757	2040	Yes
8	PRE5	P40302	Y94	YMR314W	essential	-0.923076923076923	0.107761529808774	-0.7755235	229	No
9	EMW1	P42842	Y859	YNL313C	essential	-0.920879120879121	0.366522030344506	-0.310013	158	No
10	MCM5	P29496	Y771	YLR274W	essential	-0.909090909090909	0.15186090736213	-0.515442	52.4	No
11	SNU13	P39990	Y115	YEL026W	essential	-0.907692307692308	0.0351097178683386	1.23615	349	No
12	PRE5	P40302	Y6	YMR314W	essential	-0.903030303030303	0.773003374578178	-0.622883	229	No
13	RPN2	P32565	Y857	YIL075C	essential	-0.902097902097902	0.0710942723423222	0	560	No
14	CDC12	P32468	Y134	YHR107C	essential	-0.898901098901099	0.159229077738973	-0.1449905	124	No
15	RFA1	P22336	Y586	YAR007C	essential	-0.895104895104895	0.768657032673645	-0.560763	131	No
16	SRP101	P32916	Y288	YDR292C	essential	-0.890909090909091	0.139325842696629	0	61.3	No
17	TRS31	Q03337	Y24	YDR472W	essential	-0.890909090909091	0.575232774674115	0	25.9	No
18	GCD14	P46959	Y43	YJL125C	essential	-0.89010989010989	0.553217821782178	0.703281	55.2	No
19	RPN7	Q06103	Y13	YPR108W	essential	-0.888111888111888	0.122729504172803	0	310	No
20	BET1	P22804	Y11	YIL004C	essential	-0.888111888111888	0.225980301059283	0	12.7	No
21	SCL1	P21243	Y12	YGL011C	essential	-0.888111888111888	0.766739788989867	-0.539773	380	No
22	NDC1	P32500	Y485	YML031W	essential	-0.885714285714286	0.0498309450974078	0	68.5	No
23	RIO2	P40160	Y95	YNL207W	essential	-0.884615384615385	0.368469975831939	0.0508436	69.3	No
24	RPL5	P26321	Y172	YPL131W	essential	-0.881318681318681	0.200478596118054	-0.663484	1843	Yes
25	ROM2	P51862	Y869	YLR371W	essential	-0.876923076923077	0.451487685111504	0	13.3	No
26	SAS10	Q12136	Y442	YDL153C	essential	-0.874125874125874	0.768712768712769	0.152136	32.1	No

In the example screenshot we have shown the top 20 sites ranked on this basis. We have also included other fields (e.g. protein abundance, pfam domains, etc.) that could inform the selection of interesting sites.

Suggested minor points to address:

1. Very minor point, but it might be worthwhile to specify in the main text (not just the methods section) that the kinase-dead mutants were catalytic Asp-to-Asn mutations, since some groups chose to mutate an active site Lys-to-Met. The Asp-to-Asn mutation tends to be more reliably kinase-dead, and indeed this is clear from the data in the paper.

We have now added a sentence in the main text to specify this and support our claim with a citation to a recent review paper that states explicitly that the catalytic D-to-N is the best way to inactivate a kinase (Reinhardt and Leonard, eLife 2023, PMID: 37470698).

2. It's curious that most of the GFP blots in Supplementary Figure 1 have many bands, and more so that the band pattern/intensity in the GFP blots depends on whether the kinase is active or a kinase-dead mutant. The authors should address if the multiple bands are truncations or something else, and speculate why this differs if the kinase is dead. Additionally, the Landing Pad lane in the blots also has a band. Can the authors clarify if the landing pad alone encodes GFP?

We know for sure that the full length sequence of each kinase (WT or dead) is present in the landing pad, since we used Sanger sequencing from start to finish for each construct. Once expressed, we cannot be totally sure of what happened – possibly translation differences or proteolysis. But, we would suggest that the expressed kinase is probably cleaved or degraded by the yeast machinery. Possibly this is to reduce the toxicity of the expressed kinase but we cannot be certain. Changes in stability of the kinase dead mutant vs the WT kinase can also be something

to consider since, for example in vSRC, the band that we see for the dead version is at 25 kDa, suggesting it is probably free GFP. The landing pad without kinase can express the GFP, since we inserted the kinase between the ATG and the GFP coding sequence.

3. The X-axis label "ddG (experimental)" in Supplementary Figure 5e is a bit misleading. Although the text clearly states that these are ddG predictions based on experimental structures vs AlphaFold models, one might mistake the X-values for experimentally determined ddG values. Is there a better way to label this?

Thank you. This indeed could be misleading. We made a revised figure panel to correct this:

Reviewer 2

Here, the authors set out to explore how tyrosine phosphorylation might have evolved in an organism that only utilizes Ser/Thr phosphorylation, without causing deleterious physiological effects due to spurious Tyr phosphorylation of proteins. To measure the impact of artificial TK-mediated protein phosphorylation on fitness, they used budding yeast, which lacks conventional tyrosine kinases (TKs), and engineered yeast strains inducibly expressing individually GFP-tagged versions of 24 TKs (either full length or for the RTKs catalytic domains, and for v-Src a set of 13 mutants) and 7 Ser/Thr kinases (lacking obvious yeast orthologues), and then used MS analysis of IMAC-enriched tryptic phosphopeptides to quantitatively analyze their post-induction phosphoproteomes, identifying ~30,000 phosphosites mapping to ~3,500 proteins in five biological replicates, in each case comparing a strain expressing the WT kinase to a strain expressing the cognate kinase-dead mutant. The induced phosphosites sequences were in general consistent with the known primary sequence specificities of the expressed kinases. Using AlphaFold2 (AF2) structural predictions and relative solvent accessibility of the sites to assess the possible functional consequences of the spurious phosphorylation events, they predicted that 80% of pTyr sites map to ordered/buried regions or PPI interfaces. In this way, they defined >1,000 pTyr events that might be deleterious, possibly as a result of destabilizing protein folding. The fitness effects of the 44 protein kinases (13 TKs, 10 TK catalytic domains, 7 Ser/Thr kinases, and 14 v-Src TK mutants) were tested under 41 conditions known to induce stress, finding that 5, 3, 3, and 14 kinases, respectively, in these four categories were deleterious for growth/proliferation. In general, they observed correlation between the number of spurious pTyr sites and decreased yeast growth, but a large number of the TKs and spurious pTyr sites apparently had little effect on yeast fitness, possibly due to a low phosphorylation stoichiometry; for instance, expression of EPHB3, which led to 800 spurious pTyr sites, was not toxic. Experimentally, they determined an average phosphosite stoichiometry of 16% for v-Src and 20% for EPHB1 sites, and showed that phosphorylation of Tyr sites in essential proteins did not give a more significant correlation. They also found that compared to spurious pTyr sites in yeast, native pTyr sites in human proteins were significantly more likely to be accessible and in disordered regions, with spurious phosphorylation of yeast proteins being weakly biased towards orthologues of human proteins that are known TK targets. Finally, they reexamined the issue of whether the advent of tyrosine phosphorylation might have resulted in counterselection against the presence Tyr in the proteome, and found that the representation of Tyr was not significantly lower in organisms that express TKs, other than in the yeast proteome.

In these studies, the authors have addressed experimentally an important question, namely how protein tyrosine phosphorylation might have evolved as a regulatory mechanism in an organism utilizing only Ser/Thr protein kinases. By using yeast, which lacks tyrosine kinases, as a model organism in which to inducibly express a series of TKs, they found that expression of some TKs is deleterious to yeast proliferation, whereas expression of others is not, and by characterizing the phosphoproteomes in these cells developed plausible explanations for the deleterious effects of spurious Tyr phosphorylation. However, as the authors admit, a limitation of this extensive analysis is that it remains unclear whether it is the pTyr sites in a specific subset of proteins whose phosphorylation is deleterious to yeast proliferation or whether it is a cumulative effect of multiple phosphorylations at less specific sites that decreases fitness. They did not attempt to make any Tyr to Phe mutants at sites in target proteins predicted to be particularly deleterious to test this, although this would be a major undertaking without any guarantee of success. A major omission is that the authors do not mention anywhere in the paper the possible role of protein-tyrosine

phosphatases (PTPs) in determining the effects of spurious Tyr phosphorylation on yeast fitness, which is surprising given that budding yeast express four PTPs that could, perhaps selectively, dephosphorylate some of the spurious pTyr sites. This seems to be a serious oversight that needs to be rectified (and possibly tested experimentally - see point 1).

We thank the reviewer for their insightful comments and comprehensive summary of our work. We refer to phosphatases a few times in the manuscript (Results and Discussion). However, it is true that we did not refer specifically to the PTP phosphatases and that this was an oversight on our part.

We have added a new sentence to describe PTPs in the Introduction (new text in blue):

In this context we consider phosphotyrosine signalling in the budding yeast *Saccharomyces cerevisiae*. Yeast lacks classical tyrosine kinases, SH2 domains, or PTB domains; however, they contain a class of phosphatases called PTPs (protein tyrosine phosphatases) that dephosphorylate tyrosine with high intrinsic efficiency (Pincus et al. 2008; Hunter 2009; Chen et al. 2017). There is a small amount of phosphotyrosine in the yeast proteome, but this is thought to derive from dual-specificity kinases (Manning et al, 2002; Pincus et al, 2008; Lim & Pawson, 2010; Kaneko et al, 2012; Leutert et al, 2023). Therefore, the heterologous expression of tyrosine kinases in

Also, in response to point 1 below, we refer to our new Western blot showing that pY levels increase in the presence of orthovanadate, which is a PTP inhibitor:

correct molecular weights. Finally, for one kinase (EPHB1) we repeated the Western blots in the presence of a phosphatase inhibitor cocktail containing orthovanadate (a PTP inhibitor), demonstrating that pY levels can be increased by the inhibition of endogenous phosphatases (Appendix Figure S3). Since most kinases were detected by Western blotting, we considered all kinases for downstream experiments even if their expression could be low. Almost all (~95%) of the tested kinases were detected in their

We have also added a new sentence to the Discussion section (new text in blue):

stoichiometry. Contrary to the *in silico* methods used for VEP that assume 100% stoichiometry for DNA-encoded mutations, phosphorylation of tyrosines occurs post-translationally and is reversed by phosphatases, which are present in yeast despite their lack of *bona fide* tyrosine kinases (Pincus et al, 2008; Hunter, 2009; Lim & Pawson, 2010; Chen et al, 2017). This includes established tyrosine phosphatases such as the PTP and Cdc25 families that can dephosphorylate functional pY sites on native S/T kinases (Hunter 2009), though there are a number of other phosphatase families in yeast with reported dual-specificity for pY and pS/pT (Chen et al. 2017). In agreement, we found that a phosphatase inhibitor cocktail (containing the PTP inhibitor orthovanadate) was one of the most harmful growth conditions among the 41 treatments we tested (Appendix figure S14A). As additional evidence, we found low

Finally, in the Discussion section we have added a new paragraph with more detailed commentary on endogenous yeast phosphatases and their role in the evolution of *bona fide* tyrosine kinases.

Nonetheless, these are interesting studies that have some bearing on how tyrosine phosphorylation might have evolved as a protein modification involved cell signaling in single cell eukaryotes a few hundred million years ago that are worth reporting, even if the conclusions are not totally definitive.

1. Budding yeast express three classical PTPs possessing HCSAGCGR PTP catalytic motifs and PTP activity (Ptp1, Ptp2, and Ptp3), and the yeast cell cycle progression is regulated by dephosphorylation of pY15 CDK by Mih1, the budding yeast CDC25 orthologue. PTPs have very high turnover numbers, and determining the consequences of knocking out one or more of the three Ptp genes on pTyr site stoichiometry and the toxicity of exogenous TK expression could be informative. In this context what happens to pTyr levels if yeast cells are treated with pervanadate to inhibit the three canonical yeast PTPs? The authors used a phosphatase inhibitor cocktail to inhibit endogenous phosphatases and induce cellular stress (Figure S8), but it is not stated what inhibitors were included in the cocktail - was pervanadate one of them? In terms of discussing the relevance of dephosphorylation of spurious pTyr to the observed phenotypes, other less specific intracellular small molecule phosphatases might also be able to dephosphorylate pTyr residues, if they are exposed on the surface of proteins. Finally, in from an evolutionary aspect it has been suggested that PTPs had to evolve before TKs in order to avoid the deleterious effects of spurious Tyr phosphorylation (PMID: 19269802).

We agree with the reviewer about the important role phosphatases may play in balancing spurious phosphorylation. As stated in the reply directly above, we now have a new paragraph in the Discussion section that gives much of the context that has been outlined by the reviewer.

We have referred back to our experimental protocol and confirm that the phosphatase inhibitor cocktail does indeed contain sodium orthovanadate, a known inhibitor of PTPs (PMID: 29257048). We have now described the composition of the phosphatase inhibitor cocktail in the Methods section ('Protein abundance assay by Western Blotting').

Experimentally, we have performed an anti-pTyr based Western blot to assess the effect of phosphatase inhibition (containing sodium orthovanadate) on pTyr levels after inducing kinase expression. The results are as follows:

This shows that PTP inhibition indeed slightly increases overall pY levels after the expression of EPHB1. This is now included as a new appendix figure (Appendix figure S3).

In our new paragraph in the Discussion we have a new sentence suggesting that the presence of PTPs may have enabled the later evolution of TKs by offering some protection against spurious phosphorylation.

With regard to whether spurious pTyr dephosphorylation mediated by endogenous phosphatases is important it would be informative to know how fast, if at all, spurious pTyr sites are dephosphorylated when the TK in question is switched off with a selective TK inhibitor. If a spurious pTyr phosphosite cannot be dephosphorylated in the cell, then its level of phosphorylation should continuously accumulate after induction of TK expression - was a post-induction phosphoproteomic time course carried out?

We thank the reviewer for this thoughtful suggestion. To pinpoint the kinetics of dephosphorylation we would have to use an effective kinase and proteasome inhibitor together, followed by a phosphoproteomics time course. Phosphorylation / dephosphorylation kinetics are interesting in their own right, and we would expect to see a rapid turnover of spurious pY sites, as the reviewer suggests above. While this would connect with our other observations about the low stoichiometry of pY phosphorylation, this ultimately would not enable us to infer the mechanism of toxicity or determine whether toxicity is caused by a small number of sites vs. a cumulative phosphorylation burden (i.e. the main open questions for this project). In our (respectful) view, the insight we would gain from this is not proportionate to the amount of new work that would be required at this stage of the manuscript.

We now specify in the text (see response above) that what we measure is the steady-state phosphorylation level, and the stoichiometry reflects a balance between phosphorylation and dephosphorylation by phosphatases and protein turnover (new synthesis and degradation). Figuring out which of these factors determines low stoichiometry (low kinase activity, high phosphatase activity, or high-protein turnover) would be challenging, as it may differ from one site to another. In any case, we appreciate the idea and agree that it would be an interesting expansion, but we believe it is outside of the scope of this manuscript at this stage.

2. The authors have inferred that the observed cell phenotypes are due to spurious Tyr phosphorylation based on the fact that they are induced by expression of the WT but not kinase-dead kinase. However, another way of establishing this would be to culture the cells in the presence of cognate inhibitors of toxic TKs to allow proliferation (as has often been done when toxic TKs are expressed in bacteria and also in yeast). Also, did the authors try inducibly expressing a mammalian PTP, YopH or lambda PP to determine whether this reverted any growth phenotypes?

In our view, the WT vs. kinase-dead comparison is the best and most reliable experimental design for this project as chemical inhibitors may either be non-specific or not able to achieve 100% kinase inhibition, in addition to potentially having indirect effects on other endogenous kinases that would confound the results. With our actual design, we have matching control conditions for each kinase, which we believe is optimal, and allows us to attribute the effects to the activity of the kinase.

We did not try to inducibly express exogenous phosphatases to revert any growth phenotypes. However, we note from the literature that in Florio et al., 1994 (PMID: 8049521) the authors were able to partially revert v-SCR toxicity by over-expressing human PTP1B. This is an interesting finding as it suggests that increases in PTP activity can offer some protection against spurious Y phosphorylation. We now have a sentence in the Discussion section that cites this finding:

2009). This hypothesis is supported by the previous observation that v-SRC toxicity in budding yeast can be partly reversed by the co-expression of a mammalian PTP (human PTP1B) (Florio et al. 1994).

Other points: 1. Page 5: Several of the TKs that were expressed exogenously are known to be clients for the CDC37/HSP90 kinase-specific chaperone complex - are the Cdc37/Hsp90 yeast genes important for deleterious exogenous TK activity?

Indeed this is an important question. In fact, the issue was investigated back in 1993 in the lab of Susan Lindquist (PMID: 7688470). The authors confirmed that HSP90 is required both for v-SRC activity and its associated toxicity in yeast, as lowering HSP90 expression rescues growth. A later study used temperature-sensitive mutations to confirm the requirement of Cdc37 in yeast for v-SRC mediated toxicity (PMID: 8885235). We have added a new sentence to the Introduction to describe this:

(Superti-Furga *et al*, 1993). v-SRC mediated toxicity in particular is dependent on the activity of the HSP90-CDC37 chaperone complex in *S. cerevisiae* (Xu and Lindquist 1993; Dey et al. 1996). In more recent years, this relationship between kinase activity

3. Page 5: Did the authors make use of the primary sequence specificities recently reported by Johnson *et al.* (PMID: 36631611) for the 7 S/T kinases they analyzed? In addition to the TK specificities published in the recent Shah group paper (PMID: 36927728), which the authors used, the Cantley group has presented at meetings the results of a companion study on the primary sequence specificities of all the active human TKs; this paper will presumably be published soon and the TK primary sequence specificities should be useful for further analysis of the authors' pTyr proteomic datasets.

Following this suggestion, we have now made use of the sequence specificities recently reported by Johnson *et al.*. However, for the 7 S/T kinases we used, only 2 are sufficiently active (IRAK4 and TLK2) for us to test the Johnson *et al.* PWMs against our phosphorylation data.

For our tyrosine kinases, while we have also heard about the upcoming human TK library, at the time of writing this study has not been published. We therefore make use of tyrosine kinase PWMs constructed from the data reported by Sugiyama *et al.*, 2019 (PMID: 31324866). We used these data in the original manuscript version to identify Y sites in the yeast proteome that are not phosphorylated and also unlikely to be phosphorylated by any human tyrosine kinase (TK).

In our new analysis, we confirm that the upregulated (WT-dead) phosphosites show a better match to the kinase motif compared to a sample of 10,000 randomised phosphosite sequences, and to native pY phosphosite sequences. This was determined using a normalised motif score between 0 and 1.

As can be seen, the upregulated psites (right) match the primary sequence specificities better than random (left) and native (middle) psite sequences, for all kinases except IRAK4. The difference for ABL1 is weak but still statistically significant ($p=1.4 \times 10^{-3}$, spurious-native, KS test; $p=3.4 \times 10^{-12}$, spurious-random, KS test).

We have now included this new plot in the manuscript as Appendix Figure S6.

4. Pages 6 and 10: Did expression of any of the TKs increase CDK1/Cdc28 pY14 levels and cause cell cycle arrest?

We have inspected our data and can confirm that TK expression does generally lead to an increase in phosphorylation of the inhibitory Cdc28 Y19 phosphorylation site. For example, we see that this site is strongly phosphorylated by v-SRC but to a lesser extent by human SRC. These data are represented by the following plot:

As stated in our response to a reviewer #1 comment, we have now inferred changes in kinase activity from the phosphorylation of known kinase substrates. This analysis indicates an increase in Cdc28 activity after the expression of v-SRC kinase and even more so after SRC expression. This may result from changes in CDC28 expression, changes in activation loop phosphorylation by CAK1, or changes in cyclin expression/binding. Indeed, from our analysis of activation loops we see increased phosphorylation of CDC28 T169 from the vSRC variants (Appendix figure S9).

We also note that there is some literature on the effect of v-SRC expression on the Cdc28 activity and cell cycle control. Boschelli et al., 1993 note for example that v-SRC expression causes cell cycle dysregulation and a strong increase in Cdc28 activity (PMID: 7691844). Xu and Lindquist (1993) also demonstrated that v-SRC expression causes cells to undergo cell cycle arrest (PMID: 7688470).

We followed up these observations experimentally by performing a DNA staining and flow cytometry test to test for changes in DNA content and to infer cell cycle arrest. The results are as follows for v-SRC (legend included for clarity):

Appendix figure S10: Effect of vSRC expression on ploidy level. vSRC WT or dead were expressed (100 nM estradiol) or not (0 nM estradiol) for 3 or 6 hours. Empty landing pad, haploid and diploid strains were grown in the same conditions. All strains were grown in triplicates. Cells were fixed in ethanol 70%. RNA was removed from the cells with an overnight incubation with RNase A. Finally, DNA content was stained overnight with Sytox Green. After staining, DNA content was registered for 5,000 cells in a Guava EasyCyte HT (Guava EasyCyte instrument, Cytex Bio). Figure was created with Biorender.com.

Compared to the dead mutant, there is an elevation in DNA content that is consistent with DNA synthesis but without mitosis or cytokinesis. This is in line with previous work from Boschelli et al., 1993 indicating an elevation in DNA content and loss of cell cycle control. These results are consistent with cell cycle arrest at the stage of mitosis.

Notably, the results for EPHB1 are different:

Appendix figure S11: Effect of EPHB1 expression on ploidy level. EPHB1 WT or dead were expressed (100 nM estradiol) or not (0 nM estradiol) for 3 or 6 hours. Empty landing pad, haploid and diploid strains were grown in the same conditions. All strains were grown in triplicates. Cells were fixed in ethanol 70%. RNA was removed from the cells with an overnight incubation with RNase A. Finally, DNA content was stained overnight with Sytox Green. After staining, DNA content was registered for 5,000 cells in a Guava EasyCyte HT (Guava EasyCyte instrument, Cytex Bio). Figure was created with Biorender.com.

The results for EPHB1 are more subtle and the interpretation is less clear. However, v-SRC and EPHB1 have markedly different effects on cell cycle regulation, perhaps implying different mechanisms for toxicity.

The results of the flow cytometry assays are now given in Appendix Figure S10 + Appendix Figure S11.

5. Did the authors test a pTyr-peptide specific enrichment protocol, e.g. with super-binder SH2 protein, to see whether additional pTyr sites could be identified? It is not clear from the methods whether phosphatase inhibitors were used in the lysis procedure, and, given the existence of PTPs in budding yeast, the inclusion of a PTP inhibitor, such as pervanadate, would be advisable.

For this study we chose IMAC for global phosphopeptide enrichment. While we have developed automated enrichment protocols with the SRC SH2 superbinder that increase the coverage of the pTyr phosphoproteome, these methods had not yet been established and validated when we conducted this study. Additionally, we were concerned that the use of the SRC SH2 superbinder could introduce sequence preference biases, as previously reported (PMID: 27642862). Finally, the number of pTyr sites observed in our study is quite impressive, suggesting that the IMAC enrichment methods are more than adequate to yield successful mapping of the pTyr phosphoproteome. Indeed, for the v-SRC kinase we report a number of pTyr sites comparable to what we have published for SH2 superbinder enrichment from pervanadate-stimulated HeLa cells (PMID: 37097255).

In the plots below, we show a comparison of pTyr sites identified in this study vs those from two superbinder enrichments of the spurious wt vSRC condition, measured in the Villen lab by DDA (not included in this study). It is crucial to highlight that there are significant differences in 1) MS measurement (DIA vs DDA), 2) data processing (DIA-NN vs Comet) and 3) phospho localization filtering (DIA-NN >75% vs Ascore >13). And still, it demonstrates that we are able to reach a good coverage of unbiased pY sites (up to 66% of a single superbinder run) while ensuring high pST coverage in the same MS run.

Phosphorylation site coverage of superbinder vs Fe-IMAC (this study). Identifications from this study were taken from conditions pY/pYd/vSRC as well as wt/mut and excluding the landing pad control. Conditions pST and kinase dead were excluded to highlight pTyr rich samples. This yields $n = 2$ for superbinder and $n = 186$ for this study. A) Unique number of pTyr sites identified per sample, with bar charts showing the total average. Superbinder data was filtered for Ascore ≥ 13 vs DIA-NN localization probability $\geq 75\%$ used in this study. B) Total unique number of pSer, pThr and pTyr sites. C) pTyr sites identified in this study were mapped onto 5615 from the two superbinder runs (blue), with highest observed superbinder log₁₀ MS1 intensity shown.

Superbinder pTyr sites not identified in this study (red) mostly map to low abundance regions.

With regards to phosphatase inhibitors, we agree that special care has to be taken to avoid dephosphorylation during lysis. We consciously did not use phosphatase inhibitors in the phospho-proteomics protocol used in this study, as 1) we previously showed that these inhibitors interfere with enrichment efficiency and thus phospho-peptide identifications and quantifications (PMID: 31885202), 2) we use 8M urea in our lysis buffer to prevent residual phosphatase activity and 3) lysate was kept on ice, at 4 °C, or stored at -80 °C in all steps except reduction, alkylation and digestion. We believe that this experimental approach yields best performance for sensitive and robust identification and quantification of phosphorylated peptides.

6. Page 7/Figure 2A: Solvent accessibility per se may not be a stringent enough criterion to define accessible Tyr sites for TK phosphorylation, since in general Ser/Thr/Tyr in alpha helices are not efficiently phosphorylated, if at all, by most ePKs. Another question is when do the buried pTyr sites become phosphorylated - are such Tyr sites phosphorylated during co-translationally when the protein is unfolded, or alternatively are they phosphorylated in a "denatured" subpopulation of the protein?

To answer this question we have looked into the secondary structure of the pY sites and also where they map along the protein length (i.e. in the middle or at the termini).

The results are as follows:

The secondary structure profiles were calculated from AF2 models. The secondary structure profile is almost identical between spuriously phosphorylated pY and tyrosine sites that are not phosphorylated in our dataset (for the same set of proteins). One possible interpretation is that spuriously phosphorylated sites are modified in a denatured/unfolded subpopulation as the modified sites show no (predicted) secondary structural preference. This plot has now been included as Appendix figure S7H

For the question about co-translational modification:

This histogram shows that pY, relative to non-phosphorylated tyrosines, are slightly but significantly enriched at the protein N-termini and C-termini. Since for co-translational phosphorylation we would only expect to see some enrichment at the N-terminus, these results do not strongly support the hypothesis of co-translational phosphorylation. This plot has now been included as EV Figure 2G.

7. Page 10: Did the authors try AlphaFold2-multimer to predict effects of pTyr phosphorylation on PPIs, in addition to InteractomeInsider? As noted, many PPIs involve SLiMs - were any of the novel pTyr sites in or near known SLiMs?

We should mention that, when predicting PPI perturbations, we also made use of a recent proteome-wide AlphaFold2 based PPI screen in *Saccharomyces cerevisiae* (PMID: 34762488). By the authors' own admission, their screen should work well for obligate PPIs in protein complexes but less well for transient interactions mediated by short linear motifs (SLiMs).

To address this limitation, we mined data from the ELM database (eukaryotic linear motif) using the ggGET ELM retrieval tool (<https://www.biorxiv.org/content/10.1101/2023.11.15.567056v1>). We did not see any of our spurious pY sites mapping to an experimentally verified SLiM. This could be explained by the fact that there is much less SLiM experimental data for *S. cerevisiae* than there is for humans, and that a large proportion of our pY sites (>80%) map to ordered regions.

We therefore used ggGET ELM again to predict SLiMs on the basis of regular expression matches (i.e. motif matches), while restricting our search area to regions that were solvent-exposed, accessible, and outside of annotated pfam domains. From this we cautiously predict around 5.2% of pY sites that map to at least one predicted SLiM.

AF2-multimer is not yet equipped to model phosphorylation sites, but we use it to model the interface for two putative SLiMs that overlap with spurious pY sites.

One for for a cyclin-docking motif (K/R-x-L):

Clb5 (cyclin)

And one for for a Cks1-docking motif (S/T-P):

These structural illustrations now form part of Appendix Figure S7F and Appendix Figure S7G, respectively. Our estimate of 5.2% pY sites matching at least one predicted SLiM has also now been incorporated into Figure 2F.

8. Page 10: How many endogenous yeast pTyr sites are known?

We define native pY sites as those that were identified in two recent phosphoproteome screens of budding yeast (PMID: 37845410, PMID: 33491328), giving n=169 high-confidence cases that were found in both screens. There may be more *bona fide* endogenous pTyr sites, but many of the pTyr reported in just one of the screens may represent mislocalised or lowly abundant sites (PMID: 37845410). We used this stringent filter as the inclusion of mislocalised sites would confound our analyses e.g. in the Figure 4 (now Figure 5) analysis of RSA and pY destabilisation differences between Y, native pY, and spurious pY.

9. Page 10: Did the authors attempt to measure the effect of spurious Tyr phosphorylation on the function of any of these yeast PKs or other proteins using in vitro phosphorylation by the TK in question of WT compared to a Tyr to Phe mutant form of the protein? Alternatively, they could use expanded genetic code technology to stoichiometrically install a pTyr at a spurious pTyr site of interest (PMID: 2804693) to determine its effect on protein folding and function.

We would like to thank the reviewer for this suggestion. As far as we know, it is not possible yet to use an expanded genetic code to incorporate a phosphotyrosine in a protein in yeast. Although it is possible to use *E.coli* to purify proteins containing unnatural amino acids and to convert them into

phosphotyrosines using an acid treatment, it is a challenging experimental approach (protein must be folded appropriately in *E. coli* and must resist the acid treatment). This approach has been performed with success with only a few proteins/domains, and thus would be an extensive investment of time and effort with no guarantee of success (PMID: 34929199). We argue that this is out of the scope for this research article.

However, in response to major point 5 by reviewer #1, we tested the effect of the Y → F and Y → E mutations on the interaction between the SH3 proteins Rvs167 and SLA1, which are predicted to be destabilised by Y phosphorylation at position 476 (position 53 in the SH3 domain). The Y53F mutant cannot be phosphorylated and the Y53E mutant mimics the negative charge on the pY phosphate. Through these experiments we experimentally validate our prediction that spurious phosphorylation destabilises this interaction. We refer the reviewer to major point 5 by reviewer #1 for a full response.

10. Figure 2H: Y32 is a known phosphorylation site in KRAS, and is proposed to inhibit its activity; was phosphorylation of the equivalent Y39 observed in the Ras1/2 proteins in any of the strains expressing active TKs?

This mirrors the same point made by reviewer #1 (major point 4) and so we refer reviewer #2 to that answer.

11. Page 14: The ultimate experiment would be to mutate a set of spurious pTyr sites in the proteins predicted to be the most important for a particular TK, although this would be beyond the scope of this paper.

We agree that this would be a very interesting set of experiments. We would have attempted something like this if we had found a small set of candidate sites that are responsible for toxicity. However, since there are over 1000 pY sites that we predict to be deleterious, we decided that this would not be a practical approach for us at this point. However, as noted in our response to reviewer #2 point 9 and reviewer #1 point 5 above, we have performed mutations on the phosphorylation to confirm our prediction that the pTyr site destabilises the interaction between Rvs167 and SLA1.

12. Page 16: Here, by comparing peak intensities of unphosphorylated and phosphorylated peptides from WT and dead kinase expressing cells, they authors assessed the stoichiometry of spurious pTyr site phosphorylation, an issue they really should have raised earlier. Can the authors confirm by another method that the 16% median stoichiometry they inferred for v-SRC phosphosite phosphorylation is an absolute stoichiometry.

We agree with the reviewer that assessing the stoichiometry of spurious phosphorylation sites is important for interpretations in this study. Since peptide intensities as measured by the mass spectrometer do not translate to peptide abundances, it has been historically difficult to measure stoichiometry on a large scale. Stoichiometry of individual sites can be calculated from targeted MS experiments using synthetic peptides and phosphopeptides as spike-in standards (PMID: 12771378), but these methods are only practical when a handful of sites are considered. They do not scale well to large-scale measurements. Two common large-scale approaches are the measurement of the full proteome before and after dephosphorylation, which allows inference of

stoichiometry on peptide- (not site-) level (PMID 21725298); and the calculation of stoichiometry from regulated non-phospho- and phospho-peptide pairs via ratios (PMID 20068231) or linear modeling (PMID 29183978; 29535314), of which we applied the latter in this study. The mathematical foundation has been presented by two independent research groups, including a benchmark on a ground truth dataset. We use “absolute” in the manuscript to refer not to a known absolute abundance of e.g. phosphorylated proteins, but to the fraction of the site that is phosphorylated (as opposed to the usual “relative” fold-change between conditions). The term “absolute stoichiometry” has been similarly used by others studying phosphosite phosphorylation (PMID 21725298). We note that our choice of this approach over the dephosphorylation approach (PMID 21725298) is based on pilot experiments in the lab in which we observed that pTyr sites are partially resistant to the “broad-spectrum” phosphatases used in this approach, and thus will be problematic in our study.

13. Figure 4/page 18: They showed that TK protein targets in yeast commonly have a pTyr protein orthologue in humans, and it is interesting that 20% of the pTyr sites are conserved in the human orthologue, but in human proteins do the mapped pTyr sites lie in a SLiM, or another kinase interacting surface, etc.?

Yes we agree that this is an interesting question. We find overall that quite a small number (n=118) of pY sites are site-conserved with a human pY site in an orthologue. However, this is a relatively large percentage (15.5%) of the set of sites mapping to a yeast protein with at least one tyrosine-phosphorylated human orthologue (new Figure 5D).

We briefly checked to see if there is a strong pattern in terms of the human domains that these conserved sites map to. The results are as follows:

We then checked to see if there is a strong pattern in terms of the SLiMs that the conserved pY sites map to in the human copy. Towards this end, we used an automated tool for the mining of the ELM database (eukaryotic linear motif:

<https://www.biorxiv.org/content/10.1101/2023.11.15.567056v1>). However, with this limited sample we did not see any conserved pY site mapping to an experimentally verified SLiM.

We then used predicted SLiM instances based upon regular expression matches, again using the *get elm* resource.

At least one SLiM is predicted in 32/118 of cases (27.1%). The predicted SLiM profile is as follows:

There are some matches to the GSK3 specificity motif and also to SH2 binding motif. However, it should also be mentioned that some of these SLiM classes (e.g. SH2 motifs and the endocytic AP motif Y..[LMVIF]) overlap with the TK phosphorylation motifs and so were bound to be predicted here.

These results are potentially interesting but because of the limited sample size and the lack of matches to experimentally verified SLiM instances, we do not plan on incorporating them into the revised manuscript.

14. It would be interesting to know how many more of the spurious, evolutionarily conserved pTyr sites in yeast orthologues would be detected in human cells treated with pervanadate to abolish PTP activity.

We agree with the reviewer that this would be interesting to check. If we assume that the proliferation of PTPs in humans (n=37 compared to n=3 in budding yeast) suppressed many functionless pY sites that *could* be phosphorylated in principle, then PTP inhibition in humans may increase the number of evolutionarily conserved pTyr sites.

We tested this using a mass spectrometry dataset that we published recently from HeLa cells treated with pervanadate (PMID: 37097255).

Compared to the original manuscript version, incorporating this data increases the set of unique pY sites from $n=10291$ -> $n=14176$. However, we were surprised to see that this only modestly increases the number of conserved pY sites at the site-specific level, from $n=118$ -> $n=128$.

If we attempt to recreate figure 5D (new version numbering) with this new data, the results look very similar:

For comparison, compared to Figure 5D (new manuscript version), the numbers are:

Yellow bar: 3.10% -> 3.41%

Blue bar: 8.20% -> 8.90%

Purple bar: 15.49% -> 15.01%

This analysis overall suggests that PTP inhibition in human cells does not lead to a significant increase in the number of site-conserved pY pairs when comparing this data with spuriously generated pY sites in yeast.

15. Page 22: In addition to undergoing a WGD, it has been suggested that budding yeast may also have lost a number of PK families during evolution, and it is possible that the reduced level of Tyr in the proteome may have occurred earlier during budding yeast evolution for another reason.

Yes in agreement our results suggest that the emergence of the tyrosine kinases was not the major driver of changes in proteomic Tyr content across species. We had not previously considered the role that the loss of S/T kinase families may have played in this. The following resource suggests that budding yeast lost 28 subfamilies relative to its ancestor: <http://kinase.com/evolution/>

One possibility is that the lost S/T kinase families had a strong preference for Y in their flanking regions. We therefore cross-referenced the 28 lost subfamilies with the recent Johnson et al., 2023 library of kinase PWMs where possible. We see a few 'lost' kinase subfamilies with a Ys in their primary specificity motif e.g.

MAST

E.g. DNAPK

However, for most other kinase subfamilies we observed either no Y preference or very weak Y preferences in the flanks. After examining this data, we think the lost subfamilies are unlikely to have contributed strongly to proteomic Y changes, or at least not directly. The biggest contributor may have been shifts in genomic GC content that underpin differences in proteomic AA content (PMID: 21596977).

16 Page 22: Since PTPs may have had to evolve before TKs to avoid the deleterious effects of spurious Tyr phosphorylation (PMID: 19269802), it is possible that the lack of correlation depended on the level of either specific or non-specific pTyr phosphatases expressed an early organism.

Yes we agree that the number of expressed tyrosine phosphatases is likely to be relevant, especially given the previous observation that the over-expression of a mammalian PTP can partially suppress v-SRC toxicity in yeast (Florio et al., 1994; PMID: 8049521).

We checked to see if there was any correlation between the copy number of PTP genes in the genome and proteomic Y content, achieved by searching a profile-HMM of the PTP family against

each proteome. As in the original manuscript, we correct for phylogenetic non-independence using the PIC method (Phylogenetic independence contrasts). The results are as follows:

We do not see a simple positive correlation between PTP copy number and proteomic Y content that may be expected if PTPs are reversing the toxicity of spurious Y phosphorylation. However, there are clear caveats to this analysis, such as the fact that we cannot easily account for dual-specificity phosphatases, and that the genomic PTP copy number may be a poor proxy for the total level of PTP activity in any given cell.

Nonetheless, we think this is important to consider and it has been noted previously that the evolution of kinase and phosphatase families tends to correlate (Chen et al., 2017, PMID: 28400531). For example, the early branching sponge (*Amphimedon queenslandica*) underwent a large PTP family expansion in parallel with the emergence of new TK families.

We have now added a sentence to the Discussion that raises the possibility that was suggested by the reviewer:

counter-selection for a smaller subset of spurious targets or phylogenetic lineages. The correlated evolution of tyrosine kinases and PTPs is also likely relevant for understanding the selective constraints on the pY phosphoproteome in any given species (Chen et al. 2017).

17. Page 27: Yeast does have a single SH2 domain protein, Spt6, which acts as histone chaperone, but may bind pSer rather than pTyr, and so would be considered atypical.

We would like to thank the reviewer for this helpful suggestion. Following this prompt, we took a deeper look into the Spt6 literature and were interested to find that recent papers indicate that the non-canonical tandem SH2 domain of Spt6 can bind to pS, pT, and pY with roughly equal preference. Relevant papers:

Sdano et al., 2017 eLife (PMID: 28826505)

Brazda et al., 2020, JMB (PMID: 32439331)

Connell et al., 2022 (PMID: 34967414)

We have updated the main text to reflect this recent research:

2015, 2022). We further note that yeast lacks conventional SH2 and PTB domains for the binding of phosphotyrosine (Kaneko *et al*, 2012), which can shield the pY residue from phosphatase activity (Jadwin *et al*, 2018; Hunter, 2009). While budding yeast encodes one SH2 domain protein, the histone chaperone Spt6, the tandem SH2 domain of Spt6 can bind non-canonically to pS, pT, or pY (Sdano et al. 2017; Brázda et al. 2020; Connell et al. 2022). Finally, we consider that a haploinsufficiency screen in yeast, with a theoretical reduction in protein abundance of 50%, generated a measurable fitness defect for only ~3% of the yeast genome (Deutschbauer *et al*, 2005). In principle, many proteins can therefore individually experience a reduction in abundance without a strong impact on fitness.

18. As an aside, Michael Snyder's group, which worked extensively on the yeast kinome a few years ago, reported in abstract form that 20% of recombinant yeast PKs can phosphorylate polyGluTyr. The physiological significance of this activity was not determined, but could be a reason why PTPs were evolved first.

We acknowledge the suggestion and would be an interesting hypothesis to explore in the future, when the results from the Snyder group become available.

Reviewer 3

Bradley et al are assessing artificial tyrosine phosphorylation in yeast by introducing tyrosine kinases. It has been shown that high tyrosine activity is deleterious to yeast cell growth, a fact which is not understood mechanistically. The authors describe various aspects of spurious (i.e. all the proteins which are phosphorylated) phosphorylation through computational analyses of the yeast substrates.

Spurious sites are recorded from recording phospho-proteomes from strains overexpressing GFP tagged non RTKs (13), kinase domains of RTKs (10), S/T kinases (7) and a series of vSRC (13) variants. In total, this is 4,082 upregulated pY sites mapping to 1,970 proteins and 9,014 up- and down-regulated pS/T sites mapping to 2,361 proteins, however the analyses deal with the fraction of pY sites only.

We thank the reviewer for their assessment of our paper. As explained in our response to reviewer #1, we have now devoted a new section and figure (new Figure 3) to the analysis of the pS/pT sites. Please refer to our reply to reviewer #1 for a full description of this analysis.

In the first part the authors analyze effects of all pY sites (as the union of all kinase overexpressing strains combined into one data set) on protein stability and PPIs based on computational structural models.

As explained in the response below, for much of the structural analysis we do not take the union of all phosphorylation but in fact have the results divided by individual kinases. This can also be seen in Tables EV4, EV5, EV6, and EV7.

In fact, the pY sites largely stem from the vSRC strains, and the other kinases contribute little if at all (the potential effects of individual experiments are completely lost, are there common sites etc. that relate to fitness?).

It is true that there are many kinases in our dataset that are inactive when expressed in yeast. However, we probably would not say that the pY sites largely stem from vSRC strains. In fact, EPHB1 actually generates more pY sites (WT-dead) than any other kinase and there are several other kinases that contribute a substantial fraction of pY sites (e.g. FYN, SRC, FGFR2, LYN, FRK, EPHB3, ABL1, and LCK).

We would also probably not say that the potential effects of individual experiments are completely lost. In fact, the results are separated by individual kinases in (revised manuscript numbering):

Figure 2D (upper): protein folding destabilisation

Figure 2D (lower): PPI destabilisation

EV figure 2B: pY RSA

EV figure 2D: pY disorder

EV figure 4: combined ddG (x axis - stability) and ddE (y axis - conservation)

In terms of the relative results between kinases – i.e. fraction of buried sites, fraction of ordered sites, fraction of destabilising pY – these results are very similar across kinases, with almost all kinases generating 20% of pY sites that are predicted to be destabilizing. The last part in particular is represented by a new appendix figure (Appendix figure S7B).

The final part about the common sites mirrors a point made by reviewer #1, and we have now made a new EV figure (EV Figure 1) that explicitly shows the extent of overlap between kinases, and this can be seen in our response to reviewer #1. These values are given in EV table 3. It is clear from this figure that there is substantial overlap between pY sites across kinases.

Also as stated in our response to reviewer #1, looking for a set of 'toxic only' sets that relate to fitness is not very discriminating as there are still n=2460 sites that meet this criteria.

Some analyses are recaps from the literature, some are novel.

The analysis of protein disorder overlaps with previous work in Corwin et al. 2017, and we think it is also valuable to recapitulate previous results with new data. However, to our knowledge, all of the other structural bioinformatics analysis is novel. If there are any missing references that we failed to add, we would be happy to include them in the revised version.

Figure 2f is key, summarizing all the findings such as sites predicted to destabilize a protein (using alpha fold models), a small fraction at interfaces predicted to destabilize PPIs (using interactome 3D). It is very hard to follow the conclusion that the analysis of spurious pY predicts the widespread destabilization of proteins and their PPIs.

We think maybe this depends on the interpretation of 'widespread' but for kinases like EPHB1 and vSRC we observe >400 pY sites that are predicted to destabilise the protein fold. Given caveats we note later about the phosphorylation stoichiometry, we have now toned down the relevant sentence in the manuscript to read as follows (changes are underlined):

'In summary, our proteome-wide structural analysis of spurious pY predicts the potential for widespread destabilisation of proteins and their PPIs'

We thank the reviewer for this useful suggestion.

Only a small fraction actually maps to interaction surfaces and an even smaller subset is predicted to change ddG. Actually, the data are presented in a descriptive manner using certain cut offs and no statistical assessment, e.g. through randomization etc. is given in Figure 2.

We have now re-written this section to make it less descriptive and therefore easier for the reader to follow. With respect to the statistical analysis, we actually give a comprehensive analysis of pY vs non-pY in Figure 5 (revised manuscript numbering) for *all* sites (without subsampling or randomisation) on the same proteins containing a tyrosine (either phosphorylated or not). These analyses are with respect to solvent accessibility, ordered/disordered content, and ddG (destabilization). We believe that we convincingly demonstrate strong differences between pY and non-pY for these terms. We have performed formal statistical tests on this data (Kruskal-Wallis tests) but the resulting p-values are extremely low and approximated as 0. This is now represented in the Figure 5 legend as $p < 2 \times 10^{-16}$ (Kruskal-Wallis test). We thank the reviewer for this suggestion.

I want to clearly say the analyses are interesting but the relevance is unclear. Moreover the fitness data do not support the analysis which is at some point also stated by the authors themselves: "Paradoxically, while we observe a strong negative correlation between the number of spurious pY sites and fitness (Figure 3e-f), it is also apparent that many such sites predicted to be deleterious have a negligible effect on fitness. "

This sentence refers to our analysis of the EPHB3 kinase, showing that it has a minimal impact on fitness in spite of generating around 800 spurious pY sites. One potential explanation is that the total number of observed sites is serving as a proxy for kinase activity, and the more biologically

relevant correlation is between kinase activity vs fitness (instead of # of spurious pY sites and fitness). We now explain this in the Discussion section.

The second part deals with fitness. The fitness effects of the above mentioned 44 kinases and were tested across 41 conditions known to induce various stresses (size of yeast colonies over time as a proxy for fitness), which again results an extensive body of data. There are some conditional fitness effects, with the underlying mechanism remaining unclear.

Figure 3f: fitness correlates with the number of pY sites! This is nice and two explanations are offered as possible explanation: spurious phospho-burden or the increase of stoichiometry of deleterious phospho-sites. However, except for the 3f correlation, no other clear result is obtained that connects the fitness effects to spurious phosphorylation / stoichiometry. Notably, higher stoichiometry, which stems from more active kinases (vSrcs), will also result in a higher number of yeast proteins being phosphorylated.

This is a legitimate concern and we will address this point in the discussion by adding a few sentences explaining the issues coming from the fact that stoichiometry and the number of sites both depend on the overall activity of the kinases, which makes it difficult to identify which one may be the most important contributor to fitness effects.

We have around ~1000 pY sites that we predict to be deleterious, so pinpointing the culprits for toxicity is a difficult task. However, after this revision we have a new analysis of pS/pT sites, indicating the upregulation (WT-dead) of critical activation loop pTyr sites (new Figure 3B-C), and new experimental flow cytometry analysis indicating cell cycle arrest (Appendix figure S10, Appendix figure S11). While these do not represent direct evidence linking particular phosphosites to fitness, they are candidates for further investigative research.

Evolutionary aspects: Comparisons of pY-proteins in yeast to human pY proteins, on a structural level reveals low conservation.

The Y counter selection hypothesis: Using new phylogenetic data/ new species, the authors do not find significant relationship between the number of predicted tyrosine kinases and proteome Y content, in contrast to the hypothesis by Linding et al. The analysis does contribute to the remaining controversy in the literature through novel sequence analysis, however is not based on the data presented in the manuscript (It is rechecked on spuriously phosphorylated proteins in the supplement).

It is true that the analysis in Figure 6A-D (new manuscript numbering) does not depend directly on the spurious pY sites that we measured by mass spectrometry. However, all of the Figure 6E-G evolutionary analysis relates directly to the spurious pY sites we generated in this study. As the reviewer states, some of the figure 6A conclusions are also checked on the data that we generated here in Appendix figure S24 (new manuscript numbering).

We have revised the manuscript to make it clearer to the reader that in fact the Figure 6 E-G analysis does indeed derive from the spurious pY sites recorded in this study.

They also investigate Y deserts in individual proteins again without significant results. Also evidence for counter selection on the basis of individual sites is not strong but cannot be rejected based on the presented analysis either.

We do not reject this idea completely and in the manuscript still reserve the possibility that counter-selection may be restricted to a smaller subset of sites and/or species. However, we would argue that for all of the evolutionary analyses considered together – proteome wide (figure 6A-6B), tyrosine deserts (figure 6C-6D), and site counter-selection (figure 6E-F) – there is very little evidence for widespread counter selection against spurious tyrosine phosphorylation. For example, we note in panel figure 6F, Y counter-selection for sites that are not phosphorylated is higher and much more significant (intermediate site, $p=1.2e-04$) than for exposed sites where one can see a weak but non-significant tendency for the counter-selection against pY sites (exposed sites, $p=7.7e-02$) that are spuriously phosphorylated in this dataset. In the revised manuscript, we will try to emphasise more clearly that we do not completely reject the possibility of counter-selection for some individual sites.

This analysis does not root on the data presented in the manuscript and therefore feels like an relatively loosely related appendix to the analyses.

We apologise for any confusion caused but we would like to reiterate that much of the analysis in this section does in fact root on the data presented in this manuscript. Specifically: figure 6E, figure 6F, and figure 6G.

While the earlier parts of the section do not root on the data presented in the manuscript, we felt it was necessary to test the counter-selection hypothesis at different levels (proteome, proteins, motifs, sites). We did this because the argument could have been made that selection may not be evident at the level of individual sites but may be detectable at a 'higher' level (e.g. whole proteome level as in Linding et al.).

Therefore, while we accept that for Figure 6A-D there is some disconnect with the rest of the paper, including this analysis allowed us to make much stronger biological conclusions than if we had only performed the site-based analysis shown in Figure 6E-F.

Also, we have reworded parts of this section to make it clear that Figure 6E-F draws on the data generated in this manuscript.

In summary, the topic is very interesting, the data body is substantial, however the analyses are very descriptive with no clear outcome of the many analyses. The authors do not develop a clear hypothesis (e.g. for further testing) from the analyses. A potential key finding put forward in the manuscript, namely that spurious phosphorylation impacts through protein interactions on fitness, is not sufficiently supported by the data/analyses.

We would like to thank the reviewer for their comments.

We apologise for any confusion caused, but we do not make the claim in the manuscript that 'spurious phosphorylation impacts through protein interactions on fitness'. We think there may be some confusion because in the Introduction, we refer to spurious interactions during evolution and here we are alluding to the fact that an interaction between the kinase and the substrate must occur for phosphorylation to take place. However, we do not claim that the disruption of native PPIs via spurious phosphorylation is specifically contributing to the fitness defects.

We think probably the way we communicated this idea was not very clear. We have now added a

new sentence to the Discussion to make it clear that, while we predict some PPIs to be destabilised by spurious pY, we do not necessarily believe that these disrupted interactions are responsible for toxicity.

We thank the reviewer again for alerting us about this potentially confusing aspect of our work.

Similarly, effects on protein stability can not be connected to the fitness phenotypes. However, I do understand that it is not a case of identifying individual sites ...

Yes we agree and we also state in the Discussion that the effects on stability may not necessarily be linked to fitness.

In general, this mirrors points raised by reviewer #2 about being able to connect the effects of individual sites to fitness. Therefore, we will refer reviewer #3 to our replies to reviewer #2. In brief, because we have so many candidate sites for destabilisation, testing them all experimentally is not feasible. However, we have now tested the effect of phosphorylation site Rvs167 pY476 on the Rvs167-Sla1 interaction (new EV figure 3), although we appreciate that we test for the loss of interaction stability but do not link this to toxicity. This assay is explained in more detail in our response to reviewer #1.

Dear Christian,

Thank you again for submitting your revised manuscript to The EMBO Journal. We have now received the reports of all three referees that were asked to re-assess your study, and I have already shared their comments with you (included again below). All referees are satisfied with the revision, acknowledge that the manuscript has been substantially and adequately revised, and support publication of the strengthened manuscript in The EMBO Journal. Referees #2 and #3 also have a few minor comments and suggestions for further improvement of the study and the manuscript, which you are kindly asked to address in a final version of your manuscript. Please include in your resubmission a detailed point-by-point response to the referee reports explaining how their remaining concerns are addressed in the revised manuscript.

From the editorial side, there are also some changes and corrections that we need from you before we can proceed with acceptance of the manuscript:

- Please provide a list of up to 5 keywords after the Abstract of your revised manuscript.
- Please change the heading of your Methods section to "Materials and Methods".
- Please change the heading "Code and data availability" to "Data availability".
- Please make sure that the deposited data will be publicly available at the time of publication. The reviewer access information can now be removed from the Data availability statement.
- Please change the heading of your conflict-of-interest statement to "Disclosure and competing interests statement".
- The author contributions statement should be removed from the manuscript file. Instead, we use CRediT to specify the contributions of each author in the journal submission system. Feel free to use the free text box to provide more detailed descriptions during submission. See also our guide to authors for more information:
<https://www.embopress.org/page/journal/14602075/authorguide#authorshipguidelines>.
- Please consider adding a callout for Fig. 5D to the Results section, where the other panels of the same Figure are mentioned, in alphabetical order.
- Please update your Author checklist: only the sections of the manuscript where the relevant information can be found should be mentioned in the last column; the information per se (e.g. regarding statistics) should be transferred to those sections of the manuscript file.
- EV figures should be renamed to Figure EV1-EV5 throughout the manuscript.
- Each table should be uploaded as an individual Table or Dataset (not zipped) as follows: Tables EV2-EV7, EV9-EV15 and EV18 should be renamed to Dataset EV1-EV14 and their legends should be included in a separate tab of each Excel file; the remaining tables (i.e. Tables EV1, EV8, EV16-17) should be renumbered to Table EV1-EV4. Please update all corresponding callouts throughout the manuscript.
- Please include the manuscript title on the first page of the Appendix PDF file.
- We noticed that the requested Source Data have not been uploaded yet. Please include in your resubmission all Source Data files previously requested by our Source Data coordinator along with a completed Source Data checklist.
- Please note that EMBO press papers are accompanied online by:
 - A) a short (2 sentences) summary of the findings and their significance,
 - B) 2-5 short bullet points highlighting the key results, and
 - C) a synopsis image in .jpg or .png format that is exactly 550 pixels wide and 300-600 pixels high (the height is variable within this range). You can either show a model or key data in the synopsis image. Please note that the text needs to be legible at the final size.Please upload this information along with your revised manuscript (the text for A and B should be provided in a separate Word file).
- Please define the annotated p values *** as well as provide the exact p-values for the same in the legend of figure 5a-b, d; as appropriate.
- Please note that the exact p values are not provided in the legends of figures 4b; 5e-f; EV 3a-b.

- Please indicate the statistical test used for data analysis in the legends of figures 4b-c; 5d.
- Please note that the box plots need to be defined in terms of minima, maxima, centre, bounds of box and whiskers, and percentile in the legends of figures 4b, g.
- Please note that information related to "n" is missing in the legends of figures 2b; 4b; 5d-e; 6f.
- Although "n" is provided, please describe the nature of entity for "n" in the legends of figures EV 2b; EV 5b-c, e.
- Please note that the error bars are not defined in the legends of figures 5e; 6f; EV 2b; EV 5b-c, e.
- Please make sure that Biorender.com is properly acknowledged in the legends of all those main, EV and Appendix figures where it was used, and make sure you have a license for the publication of these figures.
- Please change the order of the manuscript sections as follows: title page with complete author information, abstract, keywords, introduction, results, discussion, materials and methods, data availability, acknowledgements, disclosure and competing interests statement, references, main figure legends, tables, expanded view (EV) figure legends.

Please also note that as part of the EMBO publications' Transparent Editorial Process, The EMBO Journal publishes online a Peer Review File along with each accepted manuscript. This File will be published in conjunction with your paper and will include the referee reports, your point-by-point response and all pertinent correspondence relating to the manuscript. You can opt out of this by letting the editorial office know (contact@embojournal.org). If you do opt out, the Peer Review File link will point to the following statement: "No Peer Review File is available with this article, as the authors have chosen not to make the review process public in this case."

Best regards,

Ioannis

Referee #1:

The authors addressed all of the major comments from my original review of the manuscript, as well as many of the comments from the other two reviewers. The response and amendments to the paper were satisfactory, from my perspective, and they also reflect a substantial amount of work that yielded an improved paper. As noted in the original review, this manuscript is of general conceptual interest, even though it presents some speculative and very exploratory ideas (arguably that is why it is of broad interest!). It also provides large proteomics datasets that could be of interest both to the tyrosine kinase signaling field and the yeast biology/evolution field. As such, I feel that this manuscript, in its revised form, warrants publication.

Referee #2:

To address the extensive list of points raised by the reviewers, the authors have carried out and present the results a lot of data reanalysis, and also include a new validation experiment to analyze the effects of phosphorylation of the spurious Y53 site they identified in the Rvs67 SH3 domain on its interaction with Sla1

There is clearly more that could be done to determine how the toxic TKs induce cell growth defects, but the authors' arguments

why it would be difficult to define which spurious phosphorylation events are responsible provide a reasonable justification for not doing more prior to publication. In this connection, the authors discuss the issue of the stoichiometry of spurious Tyr/Ser/Thr phosphorylation events, which admittedly is hard to quantify accurately for large numbers of phosphosites, and that in some cases a low phosphorylation stoichiometry might limit adverse effects. In this regard, the authors have assumed that in general spurious phosphorylation events will be deleterious to protein function, in which case a high stoichiometry will be required. However, it is formally possible that the toxic phosphorylation is a result of activation of a "target" protein, in which case a low stoichiometry may not be an issue.

Points: 1. Page 22/EV Figure 3: The DHFR PCA data showing that the Y53E mutation in the Rvs167 SH3 domain reduced the strength of its interaction with Sla1 in vivo based on resistance to methotrexate DHFR inhibition are reasonable. However, a Tyr to Glu mutation is a poor pTyr mimic, since the strength of the Glu negative charge is significantly less than that of the phosphate on tyrosine and the geometries of the negative charges on Glu and pTyr with respect to the peptide backbone are very different. This might explain why the sensitivity to methotrexate of the Y53E mutant was not greater, when this mutation should represent "stoichiometric" phosphorylation, compared to the presumably substoichiometric Y53 phosphorylation observed upon EPHB1 TK induction. Also, it is not clear whether the Y53E mutant form of the Rvs167 SH3 domain folds properly - was this checked by FoldX? Finally, since one of the authors previously reported that phosphorylation of the Tyr equivalent to Y53 in an NCK SH3 domain decreased ligand binding, it would make sense to cite that paper in the text, and not just in the rebuttal.

2. Page 9/10: The authors claim that the increase in the MAPK activation loop pTyr signal when an exogenous TK is expressed is due to direct phosphorylation of the Tyr in the TXY activation loop motif by the exogenous TK - is there published biochemical evidence that other TKs, such as SFKs, can phosphorylate the MAPK TXY motif in other systems? An alternative possibility is that spurious pY events directly or indirectly reduce the activities of the cognate yeast MKPs/DUSPs that dephosphorylate these AL sites.

3. Appendix Figure S3: The effects of treatment with the phosphatase cocktail on the level of total pTyr detected by 4G10 immunoblotting was modest - can the authors estimate how much the signal was increased? The interpretation of this experiment is complicated by the fact that a combination of phosphatase inhibitors was used, and it would have been better to repeat this with pervanadate alone to inhibit PTPs

4. A minor point concerns the use of the term "destabilization". A spurious Tyr phosphorylation event at a buried Tyr might well destabilize the native fold, but for a PPI such as Rvs167/Sla1, destabilization may not be the cause of loss of the interaction, which could simply be a charge effect. Perhaps, the authors could consider using another term in such cases. Finally, in this connection, despite the authors' additional analysis on this issue that is described in their rebuttal, it remains unclear how the TKs in question gain access to phosphorylate buried Tyr residues, and the authors do not really come up with a plausible explanation (page 18).

Referee #3:

I very much acknowledge the substantial revisions of the manuscript and its improvements. Some comments for consideration.

Sentence in the introduction and the same in the final discussion paragraph:

"This comparison allows us to relate spurious interactions (i.e. phospho-sites) to fitness".

"The advantage of this system for the study of spurious interactions is that protein phosphorylation across the proteome can be assessed"

"In conclusion, even in spite of clear toxicity, this work suggests that many spurious interactions (i.e. phospho-sites) have a minimal impact upon fitness."

With the blue addition this is even more confusion: are you relating to hsYkinase-yeast protein interactions. If so, this is different to interactions later in the manuscript which refers to yeast protein-yeast protein interactions (they are not spurious interactions!) that may be affected by spurious phosphorylation. Interactions is not phospho-sites! And I do not know any more what spurious interactions are now. I suggest to remove the term spurious interaction.

New figure 3 is a useful functional addition! It shows widespread pathway activation on the basis of pS/T measurements, which is a novel and interesting addition towards mechanistic insight. Maybe CDC28 pathway deregulation contributes to toxicity...

The new reduced presentation of the analysis of human-yeast conservation (protein level, protein-pY level, and site level), is much clearer (Figure 5), however I think it too corroborates the findings in Corwin et al 2017 in a similar manner (5d) and should be referenced to in the text.

A comment on: The data therefore demonstrate significant structural differences between the spurious and native phospho-proteomes in yeast and human ... While this differences between spurious and human pY sites are worked out fine, I think it should be put into context and the reader should be reminded, that pY in the end is strongly preferential to domains (no matter

what) when compared to pS/T, the latter cluster strongly in IDPs (discussion?, "This implies a structural shift ...")
doi:10.1074/mcp.m115.051177; 10.1371/journal.pcbi.1004049; 10.1371/journal.pcbi.1002933; doi:10.1038/msb.2012.31;
10.1039/c3mb25514j (there should be tons of studies showing this)

I want to congratulate the authors to their work and suggest to go ahead.

Referee #1:

The authors addressed all of the major comments from my original review of the manuscript, as well as many of the comments from the other two reviewers. The response and amendments to the paper were satisfactory, from my perspective, and they also reflect a substantial amount of work that yielded an improved paper. As noted in the original review, this manuscript is of general conceptual interest, even though it presents some speculative and very exploratory ideas (arguably that is why it is of broad interest!). It also provides large proteomics datasets that could be of interest both to the tyrosine kinase signaling field and the yeast biology/evolution field. As such, I feel that this manuscript, in its revised form, warrants publication.

We would like to thank the referee again for their thoughtful review and comments.

Referee #2:

To address the extensive list of points raised by the reviewers, the authors have carried out and present the results a lot of data reanalysis, and also include a new validation experiment to analyze the effects of phosphorylation of the spurious Y53 site they identified in the Rvs67 SH3 domain on its interaction with Sla1

There is clearly more that could be done to determine how the toxic TKs induce cell growth defects, but the authors' arguments why it would be difficult to define which spurious phosphorylation events are responsible provide a reasonable justification for not doing more prior to publication. In this connection, the authors discuss the issue of the stoichiometry of spurious Tyr/Ser/Thr phosphorylation events, which admittedly is hard to quantify accurately for large numbers of phosphosites, and that in some cases a low phosphorylation stoichiometry might limit adverse effects. In this regard, the authors have assumed that in general spurious phosphorylation events will be deleterious to protein function, in which case a high stoichiometry will be required. However, it is formally possible that the toxic phosphorylation is a result of activation of a "target" protein, in which case a low stoichiometry may not be an issue.

We would like to thank the referee once again for their thorough and detailed comments, and for the time that they invested in the review.

With respect to the last part of the paragraph, we have added a sentence to the Discussion linking this possibility to our new analysis of yeast MAPKs, suggesting that low stoichiometry Y phosphorylation may lead to (partial) enzyme activation and consequently toxicity:

'Notably, for 'activatory' phosphorylations on enzymes such as MAPKs, low stoichiometry Y phosphorylation may be sufficient for toxicity.'

Points: 1. Page 22/EV Figure 3: The DHFR PCA data showing that the Y53E mutation in the Rvs167 SH3 domain reduced the strength of its interaction with Sla1 in vivo based on

resistance to methotrexate DHFR inhibition are reasonable. However, a Tyr to Glu mutation is a poor pTyr mimic, since the strength of the Glu negative charge is significantly less than that of the phosphate on tyrosine and the geometries of the negative charges on Glu and pTyr with respect to the peptide backbone are very different. This might explain why the sensitivity to methotrexate of the Y53E mutant was not greater, when this mutation should represent "stoichiometric" phosphorylation, compared to the presumably substoichiometric Y53 phosphorylation observed upon EPHB1 TK induction. Also, it is not clear whether the Y53E mutant form of the Rvs167 SH3 domain folds properly - was this checked by FoldX? Finally, since one of the authors previously reported that phosphorylation of the Tyr equivalent to Y53 in an NCK SH3 domain decreased ligand binding, it would make sense to cite that paper in the text, and not just in the rebuttal.

This is a very valid point to make regarding attempts to use an appropriate phospho-mimic for pY. In the legend of Figure EV3 we now add the caveat that E has a smaller negative charge than pY and is chemically distinct:

'Y53F prevents phosphorylation; Y53E partially mimics the negative charge of the phosphate, but we caveat that the phosphate group on tyrosine has a larger negative charge and pY is very distinct structurally from the glutamate side chain (Hunter 2012; Reinhardt and Leonard 2023).'

Following the reviewer suggestion, we checked whether the Y53E mutation would affect the stability of the SH3 fold using FoldX. While we do not predict tyrosine phosphorylation to affect folding stability ($\Delta\Delta G$ of -0.24 kcal/mol), we do predict a destabilising effect for Y53E on protein folding ($\Delta\Delta G$ of 2.03 kcal/mol) just above the threshold we use (2 kcal/mol) to call destabilising mutants. We therefore add another caveat to the figure legend that destabilisation of the protein fold itself could be contributing to the reduced growth during the PCA assays for Y53E.

'However, we caution that the Y53E mutation is predicted to have a destabilising effect on the SH3 domain itself ($\Delta\Delta G$ of 2.03 kcal/mol), which would also contribute to the reduced formation of the Rvs167-Sla1 interaction (in addition to destabilisation of the Rvs167-Sla1 interface).'

Finally, we confirm that the paper reporting the effect of tyrosine phosphorylation on NCK SH3 domain binding is cited in the Figure EV3 legend.

2. Page 9/10: The authors claim that the increase in the MAPK activation loop pTyr signal when an exogenous TK is expressed is due to direct phosphorylation of the Tyr in the TXY activation loop motif by the exogenous TK - is there published biochemical evidence that other TKs, such as SFKs, can phosphorylate the MAPK TXY motif in other systems? An alternative possibility is that spurious pY events directly or indirectly reduce the activities of the cognate yeast MKPs/DUSPs that dephosphorylate these AL sites.

This is an important point to raise. After searching the literature, we found only one reported instance of a tyrosine kinase directly phosphorylating a MAPK on the TxY activation loop, which was reported in *vitro*: PMID 16153436.

Interestingly, this was for a mutated form of the RET kinase and not the WT. It is difficult to interpret whether or not this lack of evidence implies that a highly active tyrosine kinase cannot directly phosphorylate a MAPK in a non-native system, and especially considering we observe pervasive spurious phosphorylation in this system (~4000 unique sites across ~1/3rd of the proteome).

Nonetheless, this is an important consideration and we have now added a sentence to the relevant Results section stating that we cannot exclude the possibility of a more indirect mechanism for the increased phosphorylation of MAPK activation loops:

'However, we cannot exclude a more complicated mechanism (e.g. MAPK phosphatase inhibition) for these observations.'

We thank the referee again for their comments.

3. Appendix Figure S3: The effects of treatment with the phosphatase cocktail on the level of total pTyr detected by 4G10 immunoblotting was modest - can the authors estimate how much the signal was increased? The interpretation of this experiment is complicated by the fact that a combination of phosphatase inhibitors was used, and it would have been better to repeat this with pervanadate alone to inhibit PTPs.

We estimate the signal increase to at least 15%. The estimation was done using the ImageStudio software from Licor. To do so we draw a box on each lane from 130kDa to 40kDa. We then compare the signal with treatment to no treatment. We agree that pervanadate would probably have been better, but since our fitness experiment was done using the combination of phosphatase inhibitors we thought it would be more significant to test with the same product.

4. A minor point concerns the use of the term "destabilization". A spurious Tyr phosphorylation event at a buried Tyr might well destabilize the native fold, but for a PPI such as Rvs167/Sla1, destabilization may not be the cause of loss of the interaction, which could simply be a charge effect. Perhaps, the authors could consider using another term in such cases. Finally, in this connection, despite the authors' additional analysis on this issue that is described in their rebuttal, it remains unclear how the TKs in question gain access to phosphorylate buried Tyr residues, and the authors do not really come up with a plausible explanation (page 18).

Yes there is potential confusion between destabilisation of the protein fold and destabilisation of the protein-protein interface. We confirm that when we use FoldX to predict inter-molecular destabilisation, we are predicting destabilisation of the protein-protein interface independently of whether the pY site is destabilising the protein fold. We have added a sentence to the Materials and Methods to clarify this.

We have also been back through the manuscript to make it clearer that, when we refer to PPI destabilisation, we are considering only the destabilisation of the protein-protein interface. We want to thank the referee for pointing this out and improving the manuscript.

Finally, it is true that we simply do not know how the TKs are accessing the buried Y residues for phosphorylation. We have reworded a sentence on page 18 to make it clear that this remains an open question from our analysis. Also, following the analysis in the previous rebuttal, we have replaced (in the Discussion) the suggestion of co-translational modification with the idea of a partly unfolded subpopulation for each substrate, which we believe is more plausible given our data.

Referee #3:

I very much acknowledge the substantial revisions of the manuscript and its improvements. Some comments for consideration.

We thank the reviewer for the review of the original manuscript and the revision, and for their suggestions.

Sentence in the introduction and the same in the final discussion paragraph:

"This comparison allows us to relate spurious interactions (i.e. phospho-sites) to fitness".

"The advantage of this system for the study of spurious interactions is that protein phosphorylation across the proteome can be assessed"

"In conclusion, even in spite of clear toxicity, this work suggests that many spurious interactions (i.e. phospho-sites) have a minimal impact upon fitness."

With the blue addition this is even more confusion: are you relating to hsYkinase-yeast protein interactions. If so, this is different to interactions later in the manuscript which refers to yeast protein-yeast protein interactions (they are not spurious interactions!) that may be affected by spurious phosphorylation. Interactions is not phospho-sites! And I do not know any more what spurious interactions are now. I suggest to remove the term spurious interaction.

Indeed in this manuscript when we use the word 'interaction' we can be referring to one of two things:

1) The transient interaction between the human kinase and yeast substrate, which may lead to tyrosine phosphorylation. This is what we mean by 'spurious interaction'.

2) Native protein-protein interactions between yeast proteins, which may be perturbed by spurious Y phosphorylation sites.

We admit that this distinction should have been made clearer in the original manuscript, and we apologise for the continued confusion.

The term 'spurious interaction' is difficult for us to avoid because in the Introduction and Discussion we frame our biological question very generally in terms of any off-target physical interaction in the proteome and only later do we focus on phosphosites, which can be products of a spurious interaction between the kinase and substrate i.e. 'a PPI that leaves a trace'.

With some careful rewording we now make clear the connection between the spurious phosphosite and the spurious interaction between the kinase and substrate:

The spurious physical interaction between the human kinase and yeast substrate leaves a trace in the form of the phosphorylated residue, which serves as a chemical tag that can be assessed and quantified using mass spectrometry-based phosphoproteomics (Olsen *et al*, 2006; Villén *et al*, 2007; Rikova *et al*, 2007; Leutert *et al*, 2019) ’

and:

This comparison allows us to relate spurious interactions to fitness, as each phosphosite is a remnant of the spurious physical interaction between the kinase and substrate.’

and:

‘In conclusion, even in spite of clear toxicity, this work suggests that many spurious interactions – represented here by phosphosites generated from non-native kinase-substrate interactions – have a minimal impact upon fitness.’

Finally, we also take steps to ensure in our structural bioinformatics analysis that we are referring to native protein-protein interactions that may be perturbed by the spurious phosphosite. For example, in the opening sentence of paragraph 4 of the structural bioinformatics section:

‘The spurious phosphosites generated may also perturb native protein-protein interactions in yeast.’

Also here:

‘In summary, our proteome-wide structural analysis of spurious pY predicts a potential for the widespread destabilisation of proteins and native protein-protein interactions.’

Also here:

‘Overall, while we predict the destabilisation of proteins and native protein-protein interfaces (Figure 2D-F), it is not clear if these effects are contributing directly to toxicity.’

New figure 3 is a useful functional addition! It shows widespread pathway activation on the basis of pS/T measurements, which is a novel and interesting addition towards mechanistic insight. Maybe CDC28 pathway deregulation contributes to toxicity...

The new reduced presentation of the analysis of human-yeast conservation (protein level, protein-pY level, and site level), is much clearer (Figure 5), however I think it too corroborates

the findings in Corwin et al 2017 in a similar manner (5d) and should be referenced to in the text.

We would like to thank the reviewer for their comments. We agree that the Figure 5D analysis supports the evolutionary findings in Corwin et al., 2017 (Corwin et al, Figure 2B) and have added a new sentence to this section that makes this point, with a citation to Corwin et al., 2017:

'This supports the previous finding that spurious pY sites in yeast are significantly more tyrosine-conserved in animals and their unicellular relatives than yeast non-pY sites (Corwin et al. 2017).'

A comment on: The data therefore demonstrate significant structural differences between the spurious and native phospho-proteomes in yeast and human ... While this differences between spurious and human pY sites are worked out fine, I think it should be put into context and the reader should be reminded, that pY in the end is strongly preferential to domains (no matter what) when compared to pS/T, the latter cluster strongly in IDPs (discussion?, "This implies a structural shift ...")

doi:10.1074/mcp.m115.051177; 10.1371/journal.pcbi.1004049; 10.1371/journal.pcbi.1002933; doi:10.1038/msb.2012.31; 10.1039/c3mb25514j (there should be tons of studies showing this)

In line with the reviewer suggestion, we have added a new sentence for context on this in the Discussion and given some citations:

'This is especially important given the general tendency of phosphotyrosine to map to ordered and domain regions compared to pS/T (Figure 2C) (Corwin et al. 2017; Ramasamy et al. 2022).'

Dr. Judit Villen
University of Washington
Genome Sciences
3720 15th Ave NE
Foege Bldg, S133C
Seattle, WA 98195

24th Jul 2024

Re: EMBOJ-2023-115986R1
The fitness cost of spurious phosphorylation

Dear Dr. Villen and Dr. Landry,

Thank you for submitting your final revised manuscript for our consideration. I am pleased to inform you that we have now accepted it for publication in The EMBO Journal.

With kind regards,

Hartmut
